# Robust Data Valuation with Weighted Banzhaf Values

**Weida Li**
vidaslee@gmail.com

**Yaoliang Yu**
School of Computer Science
University of Waterloo
Vector Institute
yaoliang.yu@uwaterloo.ca

## Abstract

Data valuation, a principled way to rank the importance of each training datum, has become increasingly important. However, existing value-based approaches (e.g., Shapley) are known to suffer from the stochasticity inherent in utility functions that render consistent and reliable ranking difficult. Recently, Wang and Jia (2023) proposed the noise-structure-agnostic framework to advocate the Banzhaf value for its robustness against such stochasticity as it achieves the largest safe margin among many alternatives. Surprisingly, our empirical study shows that the Banzhaf value is not always the most robust when compared with a broader family: weighted Banzhaf values. To analyze this scenario, we introduce the concept of Kronecker noise to parameterize stochasticity, through which we prove that the uniquely robust semi-value, which can be analytically derived from the underlying Kronecker noise, lies in the family of weighted Banzhaf values while minimizing the worst-case entropy. In addition, we adopt the maximum sample reuse principle to design an estimator to efficiently approximate weighted Banzhaf values, and show that it enjoys the best time complexity in terms of achieving an $(\epsilon, \delta)$-approximation. Our theory is verified under both synthetic and authentic noises. For the latter, we fit a Kronecker noise to the inherent stochasticity, which is then plugged in to generate the predicted most robust semi-value. Our study suggests that weighted Banzhaf values are promising when facing undue noises in data valuation.

## 1 Introduction

In machine learning, data curation plays a crucial role in training powerful models, and one big difficulty facing practitioners is how to quantify the extent of usefulness each datum possesses (under a specific application). In practice, large datasets are typically noisy and often contain mislabeled data, which could harm the downstream performance of models trained on top. Sources that affect data quality include mechanical or human labelling errors (Frénay and Verleysen 2013), and poisoning attacks from malicious adversaries (Steinhardt et al. 2017). Therefore, it would be extremely useful if the potential importance of each datum can be efficiently and reliably determined. Data valuation aims to lay out a principled way to filter out noisy data and hunt down data that are valuable to retain.

In the emerging field of data valuation, the concept of *value*, originally developed in cooperative game theory as an axiomatic and principled way to assign contributions to all players in a game, has become increasingly popular in machine learning (e.g., Ghorbani and Zou 2019; Jia et al. 2019a,b; Kwon and Zou 2022; Lin et al. 2022; Wang and Jia 2023). Typically, the procedures of training and testing are treated as games that constitute utility functions while each datum serves as a player. The contribution imputed to each player is then equivalent to the importance determined for each datum.

Recently, Wang and Jia (2023) pointed out that the inherent stochasticity in utility functions, which stems from the procedure of training models, is substantial enough to incur unexpected inconsistency in ranking data. To solve this issue, they introduced the noise-structure-agnostic framework to

37th Conference on Neural Information Processing Systems (NeurIPS 2023).

demonstrate that the Banzhaf value (Banzhaf 1965) is the most robust against such stochasticity compared with other previously employed values. Surprisingly, our extensive experiments on various datasets, see Figure 1 for some examples, reveal that the Banzhaf value is not always the most robust compared with weighted Banzhaf values, to which the Banzhaf value belongs.

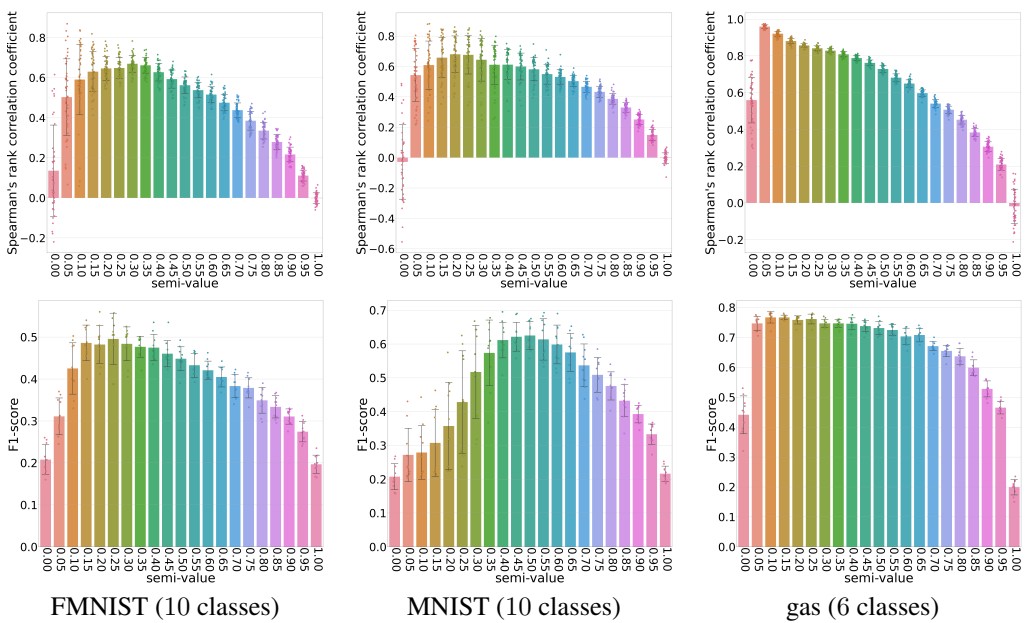

FMNIST (10 classes)  MNIST (10 classes)  gas (6 classes)

Figure 1: The first row contains results from the experiment of ranking consistency, while the second row is from the experiment of noisy label detection. For both metrics, the higher the better. The error bars represent the standard deviation over 10 random seeds. Specifically, each Spearman's rank correlation coefficient is averaged over $\binom{10}{2} = 45$ pairs of independent rankings. Each parameter along the horizontal axis defines a weighted Banzhaf value, and $0.50$ corresponds to the Banzhaf value.

In this work, our goal is to shed further light on the above-mentioned phenomenon both theoretically and empirically. As one might expect, our extensive experiments confirm that overall there is no *single* universally the best value, in terms of either noisy label detection or ranking consistency, across a variety of experiment settings. Nevertheless, we show that it is possible to *adaptively* estimate a weighted Banzhaf value that achieves competitive robustness compared with the existing baselines. We summarize our contributions as follows:

1. First, we introduce the concept of Kronecker noise to parameterize the inherent stochasticity in practice, and prove that the uniquely most robust semi-value lies in the family of weighted Banzhaf values in terms of minimizing the worst-case entropy. Specifically, the parameter for the most robust semi-value can be directly derived from the underlying Kronecker noise.

2. Second, it is known that evaluating each semi-value is exponentially expensive w.r.t. the number of data being valuated, and thus we adopt the maximum sample reuse principle to design an efficient estimator for approximating weighted Banzhaf values. Particularly, we show that it enjoys the best time complexity in terms of achieving an $(\epsilon, \delta)$-approximation.

3. Third, both synthetic and authentic noises are employed in our experiments to verify the robustness assertion in Theorem 1. For the latter, we fit a Kronecker noise to the inherent stochasticity by minimizing the Kullback-Leibler divergence between Gaussian distributions.

4. Last, our empirical study shows that it is often that some member of weighted Banzhaf values achieves the best performance in noisy label detection as well as ranking consistency when compared with the existing baselines. The empirical evidence highlights that there is no *single* universally the best value across various experiment settings.

Our code is available at `https://github.com/watml/weighted-Banzhaf`.

## 2 Background

### 2.1 Notations

Let $\mathcal{D}_{tr}$ and $\mathcal{D}_{val}$ be training and validation datasets, respectively. We write $n = |\mathcal{D}_{tr}|$, and identify $\mathcal{D}_{tr}$ with $[n] = \{1, 2, \ldots, n\}$. The definition of utility function $v : 2^{[n]} \to \mathbb{R}$ is pivotal for employing valued-based data valuation methods. Typically, for each subset $S \subseteq [n]$, $v(S)$ is set to be the performance of a chosen model trained on $S$. For example, $v(S)$ is usually the accuracy measured on $\mathcal{D}_{val}$ for classification tasks. Specifically, $v(\emptyset)$ is set to be the performance of randomly initialized models in this work. Let $\mathcal{G}_n$ be the set that contains all possible utility functions with domain $2^{[n]}$, and define $\mathcal{G} = \bigcup_{n \geq 1} \mathcal{G}_n$. Particularly, the subscript of $v_n \in \mathcal{G}$ is to indicate the number of data being valuated. For convenience, we write $S \cup i$ and $S \setminus i$ instead of $S \cup \{i\}$ and $S \setminus \{i\}$, and use $s = |S|$ when $S$ represents a set.

### 2.2 Axioms for Values

Value-based data valuation methods take advantage of the axiomatic approach developed in cooperative game theory (Dubey et al. 1981; Shapley 1953; Weber 1977). A value is a function $\phi$ that maps each utility function $v_n \in \mathcal{G}$ to a contribution vector $\mathbf{c} \in \mathbb{R}^n$. In other words, $\phi_i(v) = c_i$ is the contribution of the $i$-th datum in $\mathcal{D}_{tr}$ for the model performance achieved by training on $\mathcal{D}_{tr}$. There are four axioms we expect a value to have, which are

1. **Linearity**: $\phi(a \cdot u_n + v_n) = a \cdot \phi(u_n) + \phi(v_n)$ for every $u_n, v_n \in \mathcal{G}$ and every $a \in \mathbb{R}$;
2. **Dummy**: $\phi_i(v_n) = C$ if $v_n(S \cup i) = v_n(S) + C$ for every $S \subseteq [n] \setminus i$;
3. **Monotonicity**: $\phi_i(v_n) \geq 0$ if $v_n(S \cup i) \geq v_n(S)$ for every $S \subseteq [n] \setminus i$;
4. **Symmetry**:[1] $\phi(v_n \circ \pi) = \phi(v_n) \circ \pi$ for every permutation $\pi$ of $[n]$.

As proved by Weber (1977, Theorems 5 and 10),[2] a value $\phi$ satisfies the above four axioms if and only if there exists a list of non-negative vectors $\mathbf{\Xi} = (\mathbf{p}^n \in \mathbb{R}^n)_{n \geq 1}$ such that, for each $n \geq 1$, $\sum_{s=1}^{n} \binom{n-1}{s-1} p_s^n = 1$ and

$$\phi_i(v_n; \mathbf{\Xi}) = \sum_{S \subseteq [n] \setminus i} p_{s+1}^n \cdot (v_n(S \cup i) - v_n(S)) \text{ for every } v_n \in \mathcal{G} \text{ and every } i \in [n]. \tag{1}$$

All such values are called probabilistic values.

There is one more regularization implicitly used by Dubey et al. (1981) to filter out some undesired probabilistic values. The eventual regularized probabilistic values are called semi-values.[3][4] We explicitly rephrase the regularization as an extra axiom in this paper. Let $z$ be an extra datum labelled by $n + 1$, and extend each utility function $v_n \in \mathcal{G}$ into $\bar{v}_{n+1}$ defined by $\bar{v}_{n+1}(S) = v_n(S \cap [n])$ for every $S \subseteq [n + 1]$. The extra axiom reads as follows:

5. **Consistency**: $\phi_i(v_n; \mathbf{\Xi}) = \phi_i(\bar{v}_{n+1}; \mathbf{\Xi})$ for every $v_n \in \mathcal{G}$ and every $i \in [n]$.

The axiom of consistency states that a value should not change the previously assigned values of importance if the added data do not contribute anything. Note that the proposed axiom of consistency is the discrete version of the dummy consistency proposed by Friedman (2003, Definition 7) in cost-sharing problems. We have the following characterization for semi-values:

**Proposition 1** (Dubey et al. 1981, Theorem 1(a)). *A probabilistic value $\phi$ is a semi-value if and only if there exists a probability distribution $P$ on the interval $[0, 1]$ such that*

$$p_s^n = \int_{[0,1]} t^{s-1}(1-t)^{n-s} \mathrm{d}P(t) \text{ for every } 1 \leq s \leq n. \tag{2}$$

---

[1]It can be equivalently replaced by the axiom of equal treatment in this context, which is not generally equal to the axiom of symmetry. We refer the reader to Appendix A for more details.

[2]Their assumption that $v_n(\emptyset) = 0$ can be quickly removed by showing that $\phi(v_n) = \mathbf{0}$ if $v_n$ is constant based on the axioms of linearity and monotonicity.

[3]In some references (e.g., Wang and Jia 2023), probabilistic value is referred to as semi-value, but semi-value may refer to a distinct entity (e.g., Dragan 2006), i.e., what we call semi-value in this work.

[4]The family of probabilistic values is strictly larger than that of semi-values. For example, for $\mathbf{p}^3 = (0, 1/2, 0)$, there is no non-negative $\mathbf{p}^4$ satisfying that $p_i^3 = p_i^4 + p_{i+1}^4$ for every $1 \leq i \leq 3$.

Table 1: The underlying CDFs on the interval $[0, 1]$ for semi-values of interest. $x$ denotes the variable of CDFs. $\chi_{[0.5,1]}$ is defined by $\chi_{[0.5,1]}(x) = 1$ if $x \in [0.5, 1]$ and 0 otherwise. Beta$(\alpha, \beta)$ is a parameterized notation for the Beta Shapley values.

| Method | G-Shapley | Beta$(\alpha, \beta)$ | Data Banzhaf |
|--------|-----------|-----------------------|--------------|
| CDF | $x$ | $\propto \int_0^x t^{\beta-1}(1-t)^{\alpha-1}\mathrm{d}t$ | $\chi_{[0.5,1]}$ |

*Moreover, the map from all such probability distributions to semi-values is one-to-one.*

The axiom of consistency basically imposes that

$$p_s^n = p_s^{n+1} + p_{s+1}^{n+1} \text{ for every } 1 \le s \le n,$$

which is the key for connecting all components in $\Xi$. All in all, combing Eqs. (1) and (2) yields the formula for each semi-value,

$$\phi_i(v_n; P) = \int_0^1 \sum_{S \subseteq [n] \setminus i} t^s (1-t)^{n-1-s} (v_n(S \cup i) - v_n(S)) \, \mathrm{d}P(t) \tag{3}$$

for every $v_n \in \mathcal{G}$ and every $i \in [n]$. It suggests that each semi-value is just the integral of weighted Banzhaf values w.r.t. the underlying probability distribution $P$. Particularly, substituting the uniform distribution for $P$ recovers the formula for the Shapley value (Shapley 1953) derived from the multilinear extension of utility functions (Owen 1972, Theorem 5).

## 2.3 Semi-values for Data Valuation

As a pioneering work, Ghorbani and Zou (2019) proposed Data Shapley that includes G-Shapley and TMC-Shapley for data valuation, and demonstrated its effectiveness in applications. Later, Beta Shapley was developed to improve the overall performance of value-based data valuation (Kwon and Zou 2022). Recently, Wang and Jia (2023) set up a noise-structure-agnostic framework to demonstrate that their proposed Data Banzhaf is the most robust compared with the previous works. Basically, the distinction between these methods is the use of semi-values. The underlying cumulative density functions (CDFs) of $P$ in Eq. (2) for these value-based data valuation methods are listed in Table 1. Specifically, the TMC-Shapley (if the indices for truncation are the same for all sampled permutations of $[n]$) is some probabilistic value up to some scalar.

In cooperative game theory references, weighted Banzhaf values (which are semi-values) were discovered by Ding et al. (2010) and Marichal and Mathonet (2011, Theorem 10) based on the approximation of utility functions (Hammer and Holzman 1992).[5] To obey the axiom of symmetry, the family of symmetric weighted Banzhaf values is taken herein, which can be parameterized by $w \in [0, 1]$. Precisely, for $w$-weighted Banzhaf value, the corresponding coefficients in Eq. (1) are

$$p_s^n = w^{s-1}(1-w)^{n-s} \text{ for every } 1 \le s \le n.$$

Equivalently, its underlying CDF is $\chi_{[w,1]}$, while its underlying probability distribution is the Dirac delta distribution $\delta_w$ defined by

$$\delta_w(S) = \begin{cases} 1 & w \in S \\ 0 & w \notin S \end{cases} \text{ for every Borel measurable subset } S \text{ of } [0, 1].$$

Particularly, 0.5-weighted Banzhaf value is exactly the Banzhaf value employed in Data Banzhaf, while 1.0-weighted Banzhaf value is just the leave-one-out method, which is

$$\phi_i(v_n; \delta_1) = v_n([n]) - v_n([n] \setminus i) \text{ for every } v_n \in \mathcal{G} \text{ and } i \in [n].$$

---

[5]In some references (e.g., Domenech et al. 2016), weighted Banzhaf values are referred to as multinomial probabilistic values.

# 3 Main Work

In this section, we first introduce how to use the maximum sample reuse (MSR) principle to design an efficient estimator for weighted Banzhaf values, and also present its time complexity in terms of $(\epsilon, \delta)$-approximation. Then, we introduce the concept of Kronecker noise to parameterize the underlying stochasticity in utility functions, through which we show that the uniquely most roust semi-value always lies in the family of weighted Banzhaf values while minimizing the worst-case entropy.

## 3.1 Efficient Estimator for Weighted Banzhaf Values

To exactly compute each probabilistic value shown in Eq. (1), we have to evaluate the provided utility function $2^n$ times in total, which is intractable when $n$ is large, not to mention the expensive cost of training models. Therefore, in general, each probabilistic value can only be approximated using estimators. Specifically, the sampling lift estimator can be used to approximate every probabilistic value (Moehle et al. 2022), and its weighted version was employed by Kwon and Zou (2022) for approximating their proposed Beta Shapley values. Meanwhile, there are efficient estimators designed specifically for the Shapley value (e.g., Castro et al. 2009; Covert and Lee 2021; Jia et al. 2019b; Lundberg and Lee 2017). Recently, Wang and Jia (2023) demonstrated how to use the MSR principle to design an efficient estimator for the Banzhaf value, but they also presented that such a principle can not extend to many other probabilistic values (e.g., the Beta Shapley values). Nevertheless, we show that MSR estimator exists for weighted Banzhaf values.

For approximating $w$-weighted Banzhaf value based on some $v_n \in \mathcal{G}$, we first independently sample a sequence of Bernoulli vectors $\{\mathbf{b}_1, \mathbf{b}_2, \ldots, \mathbf{b}_m\}$ where the entries of each $\mathbf{b}_j \in \mathbb{R}^n$ are drawn from $n$ independent Bernoulli distribution $X$ with $P(X = 1) = w$; then, the sequence of Bernoulli vectors are transformed into a sequence of subsets $\{S_1, S_2, \ldots, S_m\}$ of $[n]$ by defining $i \in S_j$ if and only if the $i$-th entry of $\mathbf{b}_j$ is 1; finally, the approximate value is computed by

$$\hat{\phi}_i(v_n; \delta_w) = \frac{1}{|\mathcal{S}_{\ni i}|} \sum_{S \in \mathcal{S}_{\ni i}} v_n(S) - \frac{1}{|\mathcal{S}_{\not\ni i}|} \sum_{S \in \mathcal{S}_{\not\ni i}} v_n(S) \text{ for every } i \in [n] \quad (4)$$

where $\mathcal{S}_{\ni i} = \{S_j \mid i \in S_j\}$ and $\mathcal{S}_{\not\ni i} = \{S_j \mid i \notin S_j\}$.

As shown in Eq. (4), each utility evaluation is employed for estimating the importance of every datum in $[n]$, while for the (reweighted) sampling lift it is two utility evaluations for only one datum; see Appendix D.2 for more details. Therefore, it could be expected that the MSR estimators are much more efficient, which is certainly confirmed by Figure 2. Meanwhile, such an MSR estimator enjoys the currently best time complexity bound demonstrated by Wang and Jia (2023, Theorem 4.9).

**Proposition 2.** *Assume $\|v_n\|_\infty \le r$ for every $v_n \in \mathcal{G}$. The estimator based on Eq.(4) requires $O(\frac{n}{\epsilon^2} \log \frac{n}{\delta})$ utility evaluations to achieve $P(\|\hat{\phi}(v_n; \delta_w) - \phi(v_n; \delta_w)\|_2 \ge \epsilon) \le \delta$ for every utility function $v_n \in \mathcal{G}$.*

## 3.2 Robustness under Kronecker Noises

If $2^{[n]}$ is ordered, each stochastic utility function $v_n \in \mathcal{G}$ can be treated as a random vector in $\mathbb{R}^{2^n}$, and thus its randomness can be summarized by its covariance matrix $\mathbf{D} \in \mathbb{R}^{2^n \times 2^n}$. However, there are $\Theta(2^n)$ parameters to characterize. To make the analysis tractable, our approach is to parameterize $\mathbf{D}$ as simply as possible so that the most robust semi-value can be analyzed accordingly. Before proceeding, we introduce the binary ordering for $2^{[n]}$, so that we can legally have the covariance matrix $\mathbf{D}$.

**Ordering of** $2^{[n]}$   Define a binary sequence $[0]_2, [1]_2, \ldots, [2^n - 1]_2$ by letting $[k]_2$ be the binary representation of $k$ in $n$-digit format. We order $\{S\}_{S \subseteq [n]}$ along this binary sequence by assigning to each binary representation $[k]_2$ a subset $S_k \subseteq [n]$ defined by $i \in S_k$ if and only if the $(n+1-i)$-th digit of $[k]_2$ is 1. For example, if $n = 3$, the binary sequence is $000, 001, 010, 011, 100, 101, 110, 111$, and the corresponding ordering for the subsets of $[3]$ is $\emptyset, \{1\}, \{2\}, \{1, 2\}, \{3\}, \{1, 3\}, \{2, 3\}, \{1, 2, 3\}$. We refer to this ordering as binary ordering. In the rest of the paper, we treat each stochastic utility function $v_n \in \mathcal{G}$ as a random vector in $\mathbb{R}^{2^n}$ w.r.t. the binary ordering. The advantage of this ordering

is that it aligns well with the Kronecker product so that we can parameterize the covariance matrix of each stochastic utility function using just one simple covariance matrix $\boldsymbol{\Sigma} \in \mathbb{R}^{2 \times 2}$.

**Assumption 1** (Kronecker noises). *Let $2^{[n]}$ be ordered w.r.t. the binary ordering. For each random vector $v_n \in \mathbb{R}^{2^n}$ (equivalently, a stochastic utility function in $\mathcal{G}$), we assume $v_n - \mathbb{E}[v_n] = \boldsymbol{\epsilon}_n \otimes \boldsymbol{\epsilon}_{n-1} \otimes \cdots \otimes \boldsymbol{\epsilon}_1$ where $(\boldsymbol{\epsilon}_i)_{1 \leq i \leq n}$ are $n$ independent continuous random vectors in $\mathbb{R}^2$ that have the same positive-definite covariance matrix $\boldsymbol{\Sigma}$.*

**Proposition 3.** *Under Assumption 1, there is*

$$v_n(S) = \mathbb{E}[v_n(S)] + \prod_{i \in S} \epsilon_{i,2} \prod_{i \notin S} \epsilon_{i,1}, \text{ for every } S \subseteq [n],$$

$$\text{Cov}(v_n) = \boldsymbol{\Sigma} \otimes \boldsymbol{\Sigma} \otimes \cdots \otimes \boldsymbol{\Sigma} \text{ (n repetitions)},$$

$$\text{Cov}(v_n(S), v_n(T)) = \sigma_{11}^{|[n] \setminus (S \cup T)|} \cdot \sigma_{12}^{|(S \setminus T) \cup (T \setminus S)|} \cdot \sigma_{22}^{|S \cap T|} \text{ for every } S, T \subseteq [n]$$

**Remark 1.** *According to Proposition 3, Assumption 1 provides a simple parameterization for the covariance matrix of the stochasticity inherent in $v_n$. Though such a simple modeling is yet ideal, it is sufficient for arguing theoretically that there is no single universally the most robust semi-value across different stochastic utility functions. Notably, as shown in Table 2, weighted Banzhaf values tend to be the most consistent in ranking data, and the most empirically robust one varies across different experiment settings. Overall, our theory emphasizes that a noise-structure-specific framework is able to provide a finer conclusion for robustness compared with the previous noise-structure-agnostic framework (Wang and Jia 2023).*

Next, we propose to use the entropy of continuous random vectors to define the worst-case uncertainty brought by the parameterized covariance matrix, which is

$$\sup_{v_n \in \mathcal{G}_n : \text{Cov}(v_n) = \boldsymbol{\Sigma}^{[n]}} h(\phi(v_n; P)) \tag{5}$$

where $P$ is a probability distribution that defines a semi-value Eq. (3), $h$ is the entropy that measures the uncertainty of continuous random vectors, and $\boldsymbol{\Sigma}^{[n]} = \boldsymbol{\Sigma} \otimes \boldsymbol{\Sigma} \otimes \cdots \otimes \boldsymbol{\Sigma}$ ($n$ repetitions). Intuitively, $\phi(v_n; P)$ tends to be deterministic as $h(\phi(v_n; P))$ decreases to zero. Since Eq. (5) defines the least upper bound on $h(\phi(v_n; P))$, the lower Eq. (5) is, the more robust (equivalently, more deterministic) the corresponding semi-value is. Therefore, the most robust semi-value is defined to be the one that minimizes

$$\underset{P \in \mathcal{P}}{\text{argmin}} \sup_{v_n \in \mathcal{G}_n : \text{Cov}(v_n) = \boldsymbol{\Sigma}^{[n]}} h(\phi(v_n; P)) \tag{6}$$

where $\mathcal{P}$ is the set that contains all (Borel-measurable) probability distributions on the interval $[0, 1]$. In other words, we try to determine a semi-value that can best tolerate the largest uncertainty brought by any noises having the covariance matrix $\boldsymbol{\Sigma}^{[n]}$. As one may expect, the most robust semi-value depends on the choice of $\boldsymbol{\Sigma} \in \mathbb{R}^{2 \times 2}$.

Since $\phi(v_n; P)$ is linear in $v_n \in \mathcal{G}_n$, the result of $\text{Cov}(\phi(v_n; P))$ subject to $\text{Cov}(v_n) = \boldsymbol{\Sigma}^{[n]}$ only depends on $\boldsymbol{\Sigma}^{[n]}$ and $P$. Consequently, $\sup_{v_n \in \mathcal{G}_n : \text{Cov}(v_n) = \boldsymbol{\Sigma}^{[n]}} h(\phi(v_n; P)) = \sup_{\boldsymbol{Y} : \text{Cov}(\boldsymbol{Y}) = \boldsymbol{\Psi}} h(\boldsymbol{Y})$ where $\boldsymbol{\Psi} = \text{Cov}(\phi(v_n; P))$. It is known that $\sup_{\boldsymbol{Y} : \text{Cov}(\boldsymbol{Y}) = \boldsymbol{\Psi}} h(\boldsymbol{Y}) = \frac{n}{2}(1 + \log 2\pi) + \frac{1}{2} \log \det \boldsymbol{\Psi}$, and the maximum is achieved if $\boldsymbol{Y}$ is a Gaussian random vector. Besides, $\boldsymbol{Y}$ is Gaussian if $v_n$ is Gaussian because $\phi(v_n; P)$ is linear in $v_n \in \mathcal{G}_n$. All in all, the optimization problem (6) equivalently reduces to be

$$\underset{P \in \mathcal{P}}{\text{argmin}} \det(\text{Cov}(\phi(v_n; P)))$$
$$\text{s.t. } \text{Cov}(v_n) = \boldsymbol{\Sigma}^{[n]}. \tag{7}$$

**Theorem 1.** *Under Assumption 1, and further assume $\sigma_{12} < \min(\sigma_{11}, \sigma_{22})$, the Dirac delta distribution $\delta_{w^*}$ with*

$$w^* = \frac{\sigma_{11} - \sigma_{12}}{\sigma_{11} + \sigma_{22} - 2\sigma_{12}} \tag{8}$$

*is optimal to the problem (7) for every $n \geq 1$. In addition, the optimal solution is unique if $n > 3$.*

**Remark 2.** *Theorem 1 suggests that $0.5$-weighted Banzhaf value (i.e., the Banzhaf value) is the most robust provided that $\sigma_{11} = \sigma_{22}$, which coincides with (Wang and Jia 2023, Theorem 4.6). Specifically,*

*they simulated some isotropic Gaussian noises added to deterministic utility functions to verify that the Banzhaf value is the most robust in ranking data, which is also supported by Theorem 1. However, the first column of Figure 3 shows that the Banzhaf value is not always the most robust if the added noises follow Kronecker noises with $\sigma_{11} \neq \sigma_{22}$. In practice, Table 2 also refutes that the Banzhaf value is universally the most robust.*

## 4 Experiments

Our experiments contain three aspects: 1) approximation efficiency of the MSR estimator for weighted Banzhaf values; 2) verifying Theorem 1 using synthetic and authentic noises; 3) empirical study to show that the most robust/effective semi-values tend to be some weighted Banzhaf values.

All datasets used are from open sources, and are classification tasks. Except for MNIST and FMNIST, each $\mathcal{D}_{tr}$ or $\mathcal{D}_{val}$ is balanced between different classes. Without explicitly stated, we set $|\mathcal{D}_{val}| = 200$. All utility functions are set to be the accuracy reported on $\mathcal{D}_{val}$ with logistic regression models being trained on $\mathcal{D}_{tr}$, except that we implement LeNet (LeCun et al. 1998) for MNIST and FMNIST. To have the merit of efficiency, we adopt one-epoch one-mini-batch learning for training models in all types of experiments (Ghorbani and Zou 2019). More experiment details and results are included in Appendices D and E.

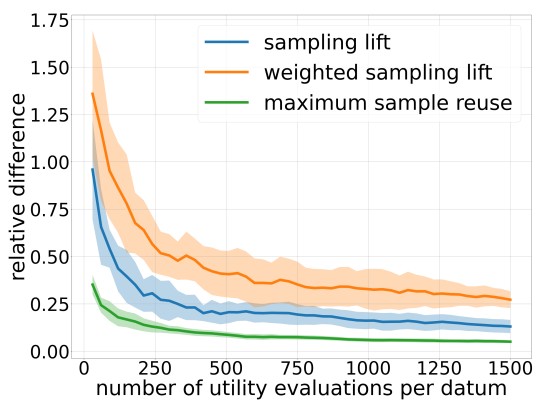

Figure 2: Comparison of three estimators for 0.8-Banzhaf value on 2dplanes dataset. The shaded region represents the standard deviation over 10 different random seeds.

### 4.1 Approximation Efficiency

The MSR estimator is compared with reweighted sampling lift and sampling lift estimators. To calculate the exact values, we set $|\mathcal{D}_{tr}| = 16$. Besides, the learning rate is set to be 1.0. To have fair comparison, we plot relative difference $\|\hat{\phi} - \phi\|_2 / \|\phi\|_2$ along the number of average utility evaluations, i.e. $\frac{\#\text{utility evaluations}}{n}$. Figure 2 is plotted with 10 different random seeds used by estimators. It demonstrates that the MSR estimator is significantly the most efficient.

### 4.2 Verification for the Theorem 1

Experiments based on synthetic and authentic noises are designed to verify Theorem 1. We set $|\mathcal{D}_{tr}| = 10$ so that the exact values can be cheaply computed.

For synthetic experiments, we simulate Kronecker noises with each $\epsilon_i$ independently and identically following a Gaussian distribution $\mathcal{N}(0, \boldsymbol{\Sigma})$, which are employed as added noises to deterministic utility functions. Besides, the learning rate is set to be 0.05. Specifically, we fix $\sigma_{11}$ and $\sigma_{22}$ while varying $\sigma_{12}$ along the horizontal axis. For each $\sigma_{12}$, the corresponding added noises are simulated using 10 different seeds. The Spearman's rank correlation coefficient is used to measure the similarity between the rankings of the ground-truth value and any noisy one. The results are reported in the first column of Figure 3, where the one tagged as "robust semi-value" refers to the corresponding weighted Banzhaf values predicted by Eq. (8). It is clear that Theorem 1 predicts the most robust semi-value well, and that the Banzhaf value is not universally the most robust under these synthetic noises.

Next, we examine Theorem 1 by fitting a Kronecker noise to the stochasticity inherent in the given utility function, and then compute the predicted most robust semi-value using Eq. (8). To produce simple utility functions, we set $|\mathcal{D}_{val}| = 10$. Besides, the learning rate for utility functions is set to be 1.0. The underlying covariance matrix is empirically approximated by $\hat{\mathbf{D}}$, and then we fit a Kronecker

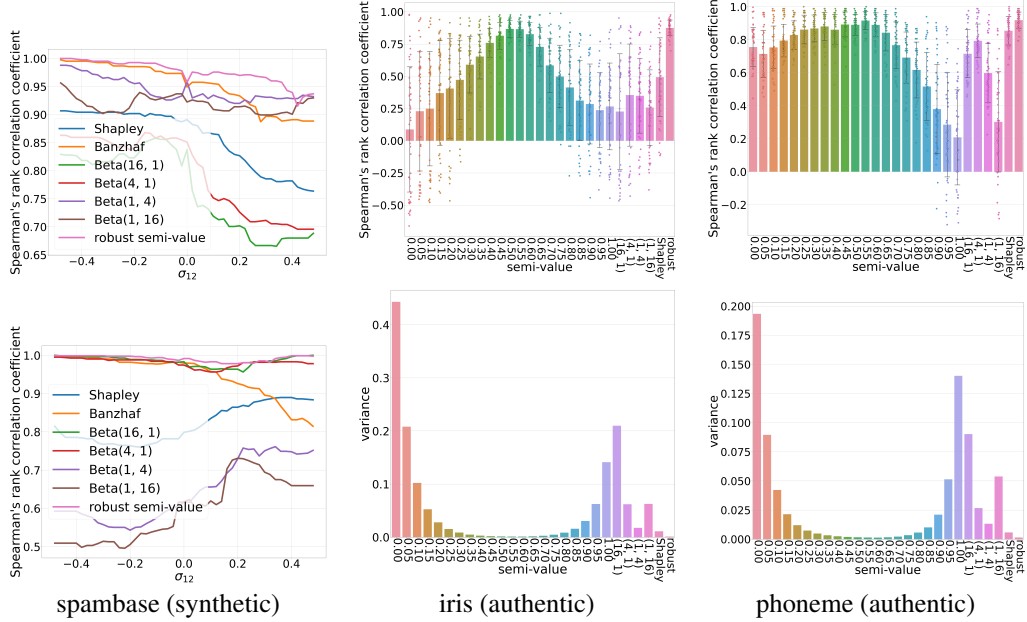

| spambase (synthetic) | iris (authentic) | phoneme (authentic) |

Figure 3: (a) Synthetic noises: Each point represents the Spearman's rank correlation coefficient averaged over 10 random seeds. For each $\sigma_{12}$, the robust semi-value is the weighted Banzhaf value parameterized by $(\sigma_{11} - \sigma_{12})/(\sigma_{11} + \sigma_{22} - 2\sigma_{12})$, as predicted by Theorem 1. For the first row, we set $\sigma_{11} = 1.5$ and $\sigma_{22} = 0.5$, while it is $\sigma_{11} = 0.5$ and $\sigma_{22} = 1.5$ for the other. (b) Authentic noises: Ranking consistency is measured by the mean and standard deviation of the Spearman's rank correlation coefficients between all pairs of 10 random seeds. The variance refers to $\sum_{i=1}^{10} \|\hat{\phi}^i - \mathbf{m}\|_2^2/10$ where $\mathbf{m} = \frac{1}{10} \sum_{i=1}^{10} \hat{\phi}^i$. The computed "robust" parameters are $0.5255$ and $0.5341$ for iris and phoneme datasets, respectively.

noise by optimizing the Kullback-Leibler divergence between Gaussian distributions, which is

$$\underset{\mathbf{\Sigma} \in \mathbb{R}^{2 \times 2}}{\operatorname{argmin}} \ \mathsf{KL}\Big(\mathcal{N}(0, \hat{\mathbf{D}}) \mid \mathcal{N}(0, \overbrace{\mathbf{\Sigma} \otimes \mathbf{\Sigma} \otimes \cdots \otimes \mathbf{\Sigma}}^{10 \text{ repetitions}})\Big).$$

The final results averaged over 10 random seeds are presented in the last two columns of Figure 3. Each parameter in $\{0.00, 0.05, 0.10, \ldots, 0.95, 1.00\}$ defines a specific weighted Banzhaf value, while $\{(16, 1), (4, 1), (1, 4), (1, 16)\}$ are parameters for the Beta Shapley values. The one tagged as "robust" is generated from the approximate Kronecker noise using Eq. (8). As demonstrated, the predicted "robust" semi-values do achieve the best in both ranking consistency and variance.

### 4.3 Empirical Study

Following the previous study (Wang and Jia 2023), we implement an exploratory study on 23 datasets, including two types of experiment: 1) noisy label detection and 2) ranking consistency.

We fix $|\mathcal{D}_{tr}| = 1,000$ for all datasets except that it is $|\mathcal{D}_{tr}| = 2,000$ for MNIST and FMNIST, which means the induced stochasticity inherent in utility functions is more complicated than Assumption 1. For each dataset, the chosen model is individually fine-tuned for the learning rate. The randomness in each utility function stems from the underlying random seed that determines the order of feeding training data as well as the initialization of models. All results are reported using mean and standard deviation over 10 random seeds. For each estimator, the total number of utility evaluations is set to be $400,000$. Permutation estimator (Castro et al. 2009) is used for Shapley value, reweighted sampling lift estimator for Beta Shapley, and MSR estimator for weighted Banzhaf values. Specifically, 0-weighted Banzhaf value and 1-weighted Banzhaf value require only $n + 1$ utility evaluations to compute exactly, and thus we do not use any estimators for them.

Table 2: Ranking consistency. For weighted Banzhaf values, we only report the best one from the parameter list $\{0.00, 0.05, 0.10, \ldots, 0.95, 1.00\}$. The first value in the column of "Weighted Banzhaf" is the parameter that achieves the highest ranking consistency. Boldface numbers mark the best. The number following each dataset refers to the number of classes used in the experiments.

| Dataset | Weighted Banzhaf | | Banzhaf | Beta(16, 1) | Beta(4, 1) | Beta(1, 4) | Beta(1, 16) | Shapley |
|---|---|---|---|---|---|---|---|---|
| MNIST (10) | 0.20 | **0.680** ± 0.120 | 0.581 ± 0.075 | 0.073 ± 0.082 | 0.020 ± 0.043 | −0.004 ± 0.039 | −0.009 ± 0.037 | 0.011 ± 0.019 |
| FMNIST (10) | 0.30 | **0.670** ± 0.042 | 0.562 ± 0.040 | 0.017 ± 0.092 | 0.007 ± 0.046 | −0.001 ± 0.035 | 0.009 ± 0.029 | 0.002 ± 0.021 |
| 2dplanes (2) | 0.80 | **0.950** ± 0.012 | 0.839 ± 0.091 | 0.424 ± 0.155 | 0.358 ± 0.133 | 0.046 ± 0.087 | 0.001 ± 0.130 | 0.256 ± 0.083 |
| bank-marketing (2) | 0.75 | **0.895** ± 0.028 | 0.814 ± 0.092 | 0.599 ± 0.096 | 0.445 ± 0.093 | 0.015 ± 0.094 | 0.041 ± 0.128 | 0.226 ± 0.033 |
| bioresponse (2) | 0.00 | **0.997** ± 0.000 | 0.948 ± 0.003 | 0.193 ± 0.032 | 0.169 ± 0.030 | 0.016 ± 0.037 | 0.004 ± 0.034 | 0.122 ± 0.021 |
| covertype (7) | 0.35 | **0.840** ± 0.033 | 0.832 ± 0.032 | 0.618 ± 0.066 | 0.490 ± 0.051 | 0.035 ± 0.134 | −0.028 ± 0.227 | 0.234 ± 0.026 |
| cpu (2) | 0.20 | **0.926** ± 0.011 | 0.874 ± 0.008 | 0.549 ± 0.096 | 0.298 ± 0.084 | 0.003 ± 0.132 | −0.004 ± 0.158 | 0.142 ± 0.028 |
| default (2) | 0.30 | **0.953** ± 0.011 | 0.941 ± 0.007 | 0.542 ± 0.078 | 0.369 ± 0.059 | 0.022 ± 0.090 | 0.004 ± 0.131 | 0.170 ± 0.034 |
| gas (6) | 0.05 | **0.959** ± 0.009 | 0.729 ± 0.019 | 0.204 ± 0.076 | 0.105 ± 0.057 | 0.003 ± 0.068 | −0.017 ± 0.094 | 0.063 ± 0.032 |
| har (6) | 0.05 | **0.956** ± 0.003 | 0.741 ± 0.012 | 0.133 ± 0.063 | 0.074 ± 0.057 | −0.004 ± 0.050 | −0.001 ± 0.058 | 0.051 ± 0.031 |
| letter (26) | 0.50 | **0.966** ± 0.004 | **0.966** ± 0.004 | 0.530 ± 0.048 | 0.346 ± 0.065 | 0.021 ± 0.110 | −0.015 ± 0.148 | 0.197 ± 0.031 |
| optdigits (10) | 0.25 | **0.923** ± 0.010 | 0.898 ± 0.009 | 0.687 ± 0.039 | 0.485 ± 0.049 | 0.024 ± 0.076 | 0.004 ± 0.145 | 0.278 ± 0.030 |
| pendigits (10) | 0.05 | **0.947** ± 0.014 | 0.806 ± 0.010 | 0.384 ± 0.100 | 0.259 ± 0.085 | 0.031 ± 0.096 | 0.024 ± 0.142 | 0.113 ± 0.024 |
| phoneme (2) | 0.85 | **0.783** ± 0.159 | 0.588 ± 0.309 | 0.601 ± 0.141 | 0.499 ± 0.148 | 0.028 ± 0.125 | 0.008 ± 0.153 | 0.325 ± 0.105 |
| pol (2) | 0.25 | **0.970** ± 0.003 | 0.938 ± 0.003 | 0.479 ± 0.108 | 0.320 ± 0.105 | 0.066 ± 0.144 | 0.033 ± 0.158 | 0.186 ± 0.026 |
| satimage (6) | 0.30 | **0.863** ± 0.024 | 0.814 ± 0.028 | 0.341 ± 0.135 | 0.189 ± 0.087 | 0.004 ± 0.086 | 0.006 ± 0.087 | 0.100 ± 0.033 |
| segment (7) | 0.25 | **0.941** ± 0.012 | 0.883 ± 0.006 | 0.361 ± 0.118 | 0.213 ± 0.069 | 0.001 ± 0.102 | 0.005 ± 0.149 | 0.113 ± 0.030 |
| spambase (2) | 0.20 | **0.939** ± 0.007 | 0.876 ± 0.007 | 0.466 ± 0.073 | 0.336 ± 0.070 | −0.001 ± 0.094 | 0.004 ± 0.148 | 0.194 ± 0.030 |
| texture (11) | 0.05 | **0.961** ± 0.011 | 0.790 ± 0.009 | 0.217 ± 0.073 | 0.109 ± 0.050 | 0.003 ± 0.072 | −0.020 ± 0.080 | 0.049 ± 0.032 |
| wind (2) | 0.00 | **0.793** ± 0.111 | 0.720 ± 0.036 | 0.398 ± 0.091 | 0.215 ± 0.078 | 0.004 ± 0.079 | 0.012 ± 0.114 | 0.116 ± 0.028 |

Table 3: Noisy label detection. For weighted Banzhaf values, we only report the best one from the parameter list $\{0.00, 0.05, 0.10, \ldots, 0.95, 1.00\}$. The first value in the column of "Weighted Banzhaf" is the parameter that achieves the highest F1-score. Boldface numbers mark the best. The number following each dataset refers to the number of classes used in the experiments.

| Dataset | Weighted Banzhaf | | Banzhaf | Beta(16, 1) | Beta(4, 1) | Beta(1, 16) | Beta(1, 4) | Shapley |
|---|---|---|---|---|---|---|---|---|
| MNIST (10) | 0.50 | **0.625** ± 0.040 | **0.625** ± 0.040 | 0.228 ± 0.023 | 0.226 ± 0.017 | 0.210 ± 0.013 | 0.207 ± 0.013 | 0.229 ± 0.020 |
| FMNIST (10) | 0.25 | **0.496** ± 0.058 | 0.448 ± 0.027 | 0.217 ± 0.019 | 0.223 ± 0.017 | 0.208 ± 0.015 | 0.209 ± 0.016 | 0.212 ± 0.016 |
| 2dplanes (2) | 0.05 | **0.609** ± 0.059 | 0.346 ± 0.014 | 0.572 ± 0.052 | 0.532 ± 0.055 | 0.219 ± 0.035 | 0.210 ± 0.030 | 0.412 ± 0.053 |
| bank-marketing (2) | 0.05 | **0.528** ± 0.015 | 0.460 ± 0.014 | 0.487 ± 0.025 | 0.452 ± 0.026 | 0.247 ± 0.024 | 0.226 ± 0.020 | 0.368 ± 0.025 |
| bioresponse (2) | 0.05 | **0.502** ± 0.012 | 0.481 ± 0.011 | 0.323 ± 0.031 | 0.316 ± 0.030 | 0.221 ± 0.022 | 0.204 ± 0.027 | 0.293 ± 0.025 |
| covertype (7) | 0.05 | **0.518** ± 0.026 | 0.318 ± 0.012 | 0.481 ± 0.025 | 0.398 ± 0.026 | 0.217 ± 0.014 | 0.212 ± 0.023 | 0.318 ± 0.032 |
| cpu (2) | 0.05 | **0.772** ± 0.017 | 0.648 ± 0.015 | 0.682 ± 0.031 | 0.508 ± 0.045 | 0.242 ± 0.028 | 0.212 ± 0.031 | 0.390 ± 0.027 |
| default (2) | 0.05 | **0.398** ± 0.010 | 0.346 ± 0.011 | 0.355 ± 0.019 | 0.316 ± 0.019 | 0.203 ± 0.022 | 0.211 ± 0.023 | 0.286 ± 0.029 |
| gas (6) | 0.10 | **0.767** ± 0.018 | 0.731 ± 0.021 | 0.402 ± 0.030 | 0.358 ± 0.020 | 0.220 ± 0.018 | 0.209 ± 0.025 | 0.321 ± 0.026 |
| har (6) | 0.10 | **0.869** ± 0.008 | 0.809 ± 0.013 | 0.388 ± 0.028 | 0.355 ± 0.025 | 0.222 ± 0.023 | 0.210 ± 0.022 | 0.328 ± 0.021 |
| letter (26) | 0.05 | **0.552** ± 0.022 | 0.467 ± 0.009 | 0.444 ± 0.038 | 0.384 ± 0.022 | 0.238 ± 0.030 | 0.226 ± 0.030 | 0.336 ± 0.022 |
| optdigits (10) | 0.05 | **0.852** ± 0.011 | 0.701 ± 0.016 | 0.739 ± 0.025 | 0.652 ± 0.023 | 0.248 ± 0.030 | 0.218 ± 0.033 | 0.513 ± 0.033 |
| pendigits (10) | 0.35 | **0.822** ± 0.006 | 0.814 ± 0.008 | 0.590 ± 0.056 | 0.489 ± 0.037 | 0.240 ± 0.027 | 0.224 ± 0.026 | 0.390 ± 0.027 |
| phoneme (2) | 0.05 | **0.492** ± 0.021 | 0.382 ± 0.023 | 0.482 ± 0.030 | 0.450 ± 0.034 | 0.242 ± 0.048 | 0.225 ± 0.033 | 0.376 ± 0.047 |
| pol (2) | 0.05 | **0.646** ± 0.011 | 0.554 ± 0.009 | 0.498 ± 0.035 | 0.476 ± 0.037 | 0.275 ± 0.046 | 0.248 ± 0.045 | 0.388 ± 0.030 |
| satimage (6) | 0.05 | **0.604** ± 0.082 | 0.476 ± 0.037 | 0.472 ± 0.055 | 0.398 ± 0.055 | 0.225 ± 0.021 | 0.209 ± 0.022 | 0.342 ± 0.031 |
| segment (7) | 0.15 | **0.763** ± 0.015 | 0.712 ± 0.014 | 0.502 ± 0.042 | 0.452 ± 0.042 | 0.249 ± 0.027 | 0.228 ± 0.031 | 0.351 ± 0.021 |
| spambase (2) | 0.05 | **0.757** ± 0.012 | 0.717 ± 0.008 | 0.586 ± 0.057 | 0.511 ± 0.033 | 0.243 ± 0.036 | 0.223 ± 0.038 | 0.412 ± 0.027 |
| texture (11) | 0.15 | **0.834** ± 0.009 | 0.762 ± 0.013 | 0.458 ± 0.041 | 0.388 ± 0.036 | 0.210 ± 0.018 | 0.194 ± 0.034 | 0.315 ± 0.028 |
| wind (2) | 0.05 | **0.686** ± 0.016 | 0.618 ± 0.019 | 0.520 ± 0.026 | 0.428 ± 0.044 | 0.227 ± 0.025 | 0.222 ± 0.031 | 0.359 ± 0.020 |

**Ranking Consistency**     In data valuation, it could be the ranking of data that matters the most, but the randomness of utility functions may produce inconsistent rankings. For each semi-value, the averaged Spearman's rank correlation coefficients over all possible pairs, which is $\binom{10}{2} = 45$ in total, are calculated to measure ranking consistency. The results are summarized in Table 2, while figures with more details are included in Appendix E. As presented, it is mostly some member of weighted Banzhaf values, not always the Banzhaf value, that achieves the highest ranking consistency compared with the baselines.

**Noisy Label Detection**     Notice that a more robust semi-value does not necessarily produce a higher quality of data valuation. This experiment is to present how well weighted Banzhaf values would achieve in noisy label detection. For each dataset, we randomly flip the labels of 20 percent of data in $\mathcal{D}_{tr}$ to be any of the rest in a uniform manner. Those flipped data are treated as having noisy/incorrect labels. Particularly, such a flipped $\mathcal{D}_{tr}$ is also used for reporting ranking consistency in Table 2. Then, each estimated semi-value is used to detect noisy labels by marking data assigned with values of importance less than or equal to 20 percent percentile as mislabeled. The results are summarized in Table 3, and figures that contain more details are included in Appendix E. As observed, the best performance is mostly achieved by weighted Banzhaf values, and most of them are not the Banzhaf value.

In both types of experiments, it manifests itself that the semi-value that achieves the best in ranking consistency does not coincide with the one that achieves the highest F1-score on each dataset. Nevertheless, the detailed figures in Appendix E show that weighted Banzhaf values tend to be more robust than the Beta Shapley values, which include the Shapley value. Moreover, we can conclude that there is no single universally the best semi-value across various experiment settings in terms of either ranking consistency or noisy label detection. For each utility function, it would be useful if the best semi-value can be effectively determined in advance.

## 5    Conclusion

We introduce a noise-structure-specific framework to theoretically assert that it is always a member of weighted Banzhaf values that is deemed the most robust in terms of minimizing the worst-case entropy. Theorem 1 shows that the most robust semi-value would vary across different experiment settings. Our extensive empirical study demonstrates that weighted Banzhaf values are promising when dealing with inevitable noises inherent in utility functions. Particularly, all experiments suggest that there is no universally the best or the most robust semi-value for all experiment settings. Therefore, future investigation is to set up principled approaches that can pin down the most robust or the most effective value in advance.

## 6    Limitations

The concept of Kronecker noise is still ideal for fitting the stochasticity in practice. The verification of Theorem 1 using fitted Kronecker noises can only be done based on simple utility functions, as one may expect. Nevertheless, the theoretical and empirical evidence is sufficient to argue that there is no single universally the most robust or the most effective semi-value across various experiment settings, and that weighted Banzhaf values are promising in practice.

## Acknowledgments and Disclosure of Funding

We thank the reviewers and the area chair for thoughtful comments that have improved our final draft. YY gratefully acknowledges NSERC and CIFAR for funding support.

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

# Table of Contents

## A  The Axiom of Equal Treatment

In the existing references, the axiom of symmetry is sometimes used to refer to the axiom of equal treatment, but they are not always interchangeable.

> **Equal Treatment**: $\phi_i(v_n) = \phi_j(v_n)$ if $v_n(S \cup i) = v_n(S \cup j)$ for every $S \subseteq [n]\backslash\{i, j\}$.

The axiom of equal treatment is not always equal to the axiom of symmetry. As a counterexample, pick $n = 2$, and consider a value $\phi$ defined by $\phi_1(v) = \frac{1}{2}v(1) + \frac{1}{2}v(2)$ and $\phi_2(v) = \frac{1}{3}v(1) + \frac{2}{3}v(2)$. The reader can verify that it satisfies the axiom of equal treatment but violates the axiom of symmetry (e.g., take $v(1) = 3$ and $v(2) = 6$).

Nevertheless, they are equivalent in some contexts. Recall that a weaker version of the axiom of dummy is

> **Null**: $\phi_i(v_n) = 0$ if $v_n(S \cup i) = v_n(S)$ for every $S \subseteq [n]\backslash i$.

**Proposition 4.** *Suppose a value $\phi$ verifies the axioms of linearity and null, then $\phi$ verifies the axiom of symmetry if and only if it verifies the axiom of equal treatment.*

*Proof.* The only if part is trivial as the axiom of symmetry always implies the axiom of equal treatment, and thus we only prove the if part. Let $n \geq 1$ be fixed, since $\phi$ verifies the axioms of linearity and null, by (Weber 1977, Theorem 2),[6] we have

$$\phi_i(v_n) = \sum_{S \subseteq [n]\backslash i} p_S^i \left( v_n(S \cup i) - v_n(S) \right) \text{ for every } v_n \in \mathcal{G}_n \text{ and } i \in [n].$$

For every $S \subseteq N$, we define $u_S \in \mathcal{G}_n$ by letting $u_S(T) = 1$ if $T = S$ and 0 otherwise. For every $i \neq j$ and $S \subseteq [n]\backslash ij$, by the axiom of equal treatment, there is

$$p_S^i = \phi_i(u_{S \cup i} + u_{S \cup j} + u_{S \cup ij}) = \phi_j(u_{S \cup i} + u_{S \cup j} + u_{S \cup ij}) = p_S^j. \tag{9}$$

On the other hand, for every $S$ with $s > 1$, and every $i, j \in S$, there is

$$p_{S\backslash i}^i = \phi_i(u_S) = \phi_j(u_S) = p_{S\backslash j}^j. \tag{10}$$

---

[6]By using the axiom of null instead of the axiom of dummy, the only difference is that there is no $\sum_{S \subseteq [n]\backslash i} p_S^i = 1$.

Combing Eqs. (9) and (10), we can get $p_S^i = p_s$ for every $i \in [n]$ and every $S \subseteq [n]\backslash i$. To demonstrate, take $p_{\{2,3\}}^1$ and $p_{\{5,6\}}^4$ as example, there is

$$p_{\{2,3\}}^1 = p_{\{2,3\}}^4 = p_{\{4,3\}}^2 = p_{\{4,3\}}^5 = p_{\{4,5\}}^3 = p_{\{4,5\}}^6 = p_{\{5,6\}}^4.$$

Therefore, the equality $p_S^i = p_s$ for every $i \in [n]$ and every $S \subseteq [n]\backslash i$ implies the axiom of symmetry. □

## B Proofs for the Propositions

**Proposition 2.** *Assume $\|v_n\|_\infty \leq r$ for every $v_n \in \mathcal{G}$. The estimator based on Eq.(4) requires $O(\frac{n}{\epsilon^2} \log \frac{n}{\delta})$ utility evaluations to achieve $P(\|\hat{\phi}(v_n; \delta_w) - \phi(v_n; \delta_w)\|_2 \geq \epsilon) \leq \delta$ for every utility function $v_n \in \mathcal{G}$.*

*Proof.* Note that it only requires $n + 1$ utility evaluations for exactly yielding 0 and 1-weighted Banzhaf values. Therefore, the MSR estimator is designed for $0 < w < 1$. Our proof is adapted from (Wang and Jia 2023, Theorem 4.9).

Let $v_n \in \mathcal{G}$ and $i \in [n]$ be fixed, and by assumption $\|v_n\|_\infty \leq r$. According to the underlying sampling for $\{S_j\}$, there is $|\mathcal{S}_{\ni i}| \sim \text{binomial}(w)$. Let a bridge estimator defined by

$$\overline{\phi}_i(v_n; \delta_w) = \frac{1}{wm} \sum_{S \in \mathcal{S}_{\ni i}} v_n(S) - \frac{1}{(1-w)m} \sum_{S \in \mathcal{S}_{\not\ni i}} v_n(S).$$

For simplicity, we write $\overline{\phi}_i$, $\hat{\phi}_i$ and $\phi_i$ instead of $\overline{\phi}_i(v_n; \delta_w)$, $\hat{\phi}_i(v_n; \delta_w)$ and $\phi_i(v_n; \delta_w)$. Plus, we also have $s_{\ni i} = |\mathcal{S}_{\ni i}|$ and $s_{\not\ni i} = |\mathcal{S}_{\not\ni i}|$.

First of all, we prove that $\mathbb{E}[\overline{\phi}_i] = \phi_i$. Suffice it to prove that $\mathbb{E}[c_i(S)v_n(S)] = \phi_i$ where $c_i(S) = \frac{1}{w}$ if $i \in S$ and $-\frac{1}{(1-w)}$ otherwise. For every $S \in [n]\backslash i$,

$$\frac{P(S \cup i)}{w} = \frac{w^{s+1}(1-w)^{n-1-s}}{w} = w^s(1-w)^{n-1-s},$$

$$-\frac{P(S)}{1-w} = -\frac{w^s(1-w)^{n-s}}{1-w} = -w^s(1-w)^{n-1-s}.$$

On the other hand, $\phi_i = \sum_{S \subseteq [n]\backslash i} w^s(1-w)^{n-1-s}(v_n(S \cup i) - v_n(S))$.

Next, we prove the convergence for the MSR estimator. Since $\mathbb{E}[\overline{\phi}_i] = \phi_i$, by the Hoeffding's inequality,

$$P(|\overline{\phi}_i - \phi_i| \geq \epsilon) \leq 2\exp\left(-\frac{m\epsilon^2}{2r^2T^2}\right)$$

where $T = \max(\frac{1}{w}, \frac{1}{1-w})$. Assume $s_{\ni i} > 0$ and $s_{\not\ni i} > 0$, note that $s_{\ni i} + s_{\not\ni i} = m$, there is

$$|\hat{\phi}_i - \overline{\phi}_i| = \left|\left(\frac{1}{s_{\ni i}} - \frac{1}{wm}\right)\sum_{S \in \mathcal{S}_{\ni i}} v_n(S) + \left(\frac{1}{s_{\not\ni i}} - \frac{1}{(1-w)m}\right)\sum_{S \in \mathcal{S}_{\not\ni i}} v_n(S)\right|$$

$$\leq \frac{r}{wm}|s_{\ni i} - wm| + \frac{r}{(1-w)m}|s_{\not\ni i} - (1-w)m|$$

$$= \frac{r}{wm}|s_{\ni i} - wm| + \frac{r}{(1-w)m}|s_{\ni i} - wm| = \frac{r}{w(1-w)m}|s_{\ni i} - wm|.$$

Note that the first inequality in the above still holds if $s_{\ni i} = 0$ or $s_{\not\ni i} = 0$. Since $s_{\ni i} \sim \text{binomial}(w)$, by the Hoeffding's inequality, there is

$$P(|s_{\ni i} - wm| \geq \Delta) \leq 2\exp\left(-\frac{2\Delta^2}{m}\right).$$

Therefore, $|\hat{\phi}_i - \overline{\phi}_i| < \frac{r\Delta}{w(1-w)m}$ provided that $|s_{\ni i} - wm| < \Delta$. Then,

$$P(|\hat{\phi}_i - \phi_i| \geq \epsilon) = P(|\hat{\phi}_i - \phi_i| \geq \epsilon \cap |s_{\ni i} - wm| < \Delta) + P(|\hat{\phi}_i - \phi_i| \geq \epsilon \cap |s_{\ni i} - wm| \geq \Delta)$$

$$\leq P(|\hat{\phi}_i - \phi_i| \geq \epsilon \mid |s_{\ni i} - wm| < \Delta) + 2\exp\left(-\frac{2\Delta^2}{m}\right)$$

$$\leq P\left(|\overline{\phi}_i - \phi_i| \geq \epsilon - \frac{r\Delta}{w(1-w)m} \mid |s_{\ni i} - wm| < \Delta\right) + 2\exp\left(-\frac{2\Delta^2}{m}\right)$$

$$\leq \frac{P(|\overline{\phi}_i - \phi_i| \geq \epsilon - \frac{r\Delta}{w(1-w)m})}{1 - 2\exp\left(-\frac{2\Delta^2}{m}\right)} + 2\exp\left(-\frac{2\Delta^2}{m}\right)$$

$$\leq \frac{2\exp\left(-\frac{m\left(\epsilon - \frac{r\Delta}{w(1-w)m}\right)^2}{2r^2T^2}\right)}{1 - 2\exp\left(-\frac{2\Delta^2}{m}\right)} + 2\exp\left(-\frac{2\Delta^2}{m}\right)$$

$$\leq 3\exp\left(-\frac{m\left(\epsilon - \frac{r\Delta}{w(1-w)m}\right)^2}{2r^2T^2}\right) + 2\exp\left(-\frac{2\Delta^2}{m}\right)$$

where the last inequality is due to that $1 - 2\exp\left(-\frac{2\Delta^2}{m}\right) \geq \frac{2}{3}$ if $m$ is sufficiently large. Solving the equation $\frac{m\left(\epsilon - \frac{r\Delta}{w(1-w)m}\right)^2}{2r^2T^2} = \frac{2\Delta^2}{m}$ yields $\Delta = \frac{w(1-w)m\epsilon}{r(2Tw - 2Tw^2 + 1)}$. Specifically, $2Tw - 2Tw^2 + 1 = 2|w - 0.5| + 2$, and thus its range is $[2, 3)$ provided that $w \in (0, 1)$, which indicates $\epsilon - \frac{r\Delta}{w(1-w)m} \geq \frac{\epsilon}{2}$. So, we have $\frac{2\Delta^2}{m} = Cm\epsilon^2$ where $C = \frac{2w^2(1-w)^2}{r^2(2|w-0.5|+2)^2}$. Eventually,

$$P(|\hat{\phi}_i - \phi_i| \geq \epsilon) \leq 5\exp\left(-Cm\epsilon^2\right)$$

provided that $1 - 2\exp\left(-\frac{2\Delta^2}{m}\right) \geq \frac{2}{3}$, or equivalently $m \geq \frac{\log 6}{C\epsilon^2}$. Then,

$$P(\|\hat{\phi} - \phi\|_2 \geq \epsilon) \leq P\left(\bigcup_{i \in [n]} |\hat{\phi}_i - \phi_i| \geq \frac{\epsilon}{\sqrt{n}}\right) \leq 5n\exp\left(-\frac{Cm\epsilon^2}{n}\right).$$

Solving $5n\exp\left(-\frac{Cm\epsilon^2}{n}\right) \leq \delta$ leads to $m \geq \frac{n}{C\epsilon^2}\log\frac{5n}{\delta}$. Eventually, we conclude that the MSR estimator requires

$$\max\left(\frac{\log 6}{C\epsilon^2}, \frac{n}{C\epsilon^2}\log\frac{5n}{\delta}\right) = O\left(\frac{n}{\epsilon^2}\log\frac{n}{\delta}\right)$$

samples (equivalently, utility evaluations) to achieve $P(\|\hat{\phi} - \phi\|_2 \geq \epsilon) \leq \delta$.

$\square$

**Proposition 3.** *Under Assumption 1, there is*

$$v_n(S) = \mathbb{E}[v_n(S)] + \prod_{i \in S} \epsilon_{i,2} \prod_{i \notin S} \epsilon_{i,1}, \text{ for every } S \subseteq [n],$$

$$\mathrm{Cov}(v_n) = \Sigma \otimes \Sigma \otimes \cdots \otimes \Sigma \ (n \text{ repetitions}),$$

$$\mathrm{Cov}(v_n(S), v_n(T)) = \sigma_{11}^{|[n]\setminus(S\cup T)|} \cdot \sigma_{12}^{|(S\setminus T)\cup(T\setminus S)|} \cdot \sigma_{22}^{|S\cap T|} \text{ for every } S, T \subseteq [n]$$

*Proof.* By Assumption 1, $v_n - \mathbb{E}[v_n] = \epsilon_n \otimes \epsilon_{n-1} \otimes \cdots \otimes \epsilon_1$ such that $(\epsilon_i)_{1 \leq i \leq n}$ are independent continuous random vectors in $\mathbb{R}^2$ that have the same covariance matrix $\Sigma$. For simplicity, we write $\epsilon^{[n]} = \epsilon_n \otimes \epsilon_{n-1} \otimes \cdots \otimes \epsilon_1$.

Since $(A \otimes B)(C \otimes D) = (AC) \otimes (BD)$, there is

$$\epsilon^{[n]}(\epsilon^{[n]})^\top = (\epsilon_n\epsilon_n^\top) \otimes (\epsilon_{n-1}\epsilon_{n-1}^\top) \otimes \cdots \otimes (\epsilon_1\epsilon_1^\top).$$

By the independence, we have

$$\text{Cov}(v_n) = \text{Cov}(\boldsymbol{\epsilon}^{[n]}) = \overbrace{\boldsymbol{\Sigma} \otimes \boldsymbol{\Sigma} \otimes \cdots \otimes \boldsymbol{\Sigma}}^{n \text{ times}}. \tag{11}$$

Next, we prove the rest by induction. It is clear that it holds for $n = 1$. Suppose it holds for $n = k > 0$, i.e., ,

$$v_k(S) = \mathbb{E}[v_k(S)] + \prod_{i \in S} \epsilon_{i,2} \prod_{i \notin S} \epsilon_{i,1} \text{ for every } S \subseteq [k],$$

$$\text{Cov}(v_k(S), v_k(T)) = \sigma_{11}^{|[k] \setminus (S \cup T)|} \cdot \sigma_{12}^{|(S \setminus T) \cup (T \setminus S)|} \cdot \sigma_{22}^{|S \cap T|} \text{ for every } S, T \subseteq [k] \tag{12}$$

and we proceed for $n = k + 1$. Suppose the binary ordering for $\{S\}_{S \subseteq [k]}$ is

$$S_1, S_2, \cdots, S_{2^k},$$

then, the binary ordering for $\{S\}_{S \subseteq [k+1]}$ is

$$S_1, S_2, \dots, S_{2^k}, S_1 \cup \{k+1\}, S_2 \cup \{k+1\}, \dots, S_{2^k} \cup \{k+1\}, \tag{13}$$

which aligns well with $\boldsymbol{\epsilon}_{k+1} \otimes (\boldsymbol{\epsilon}_k \otimes \boldsymbol{\epsilon}_{k-1} \otimes \cdots \otimes \boldsymbol{\epsilon}_1)$. Thus, we have $v_{k+1}(S) = \mathbb{E}[v_{k+1}(S)] + \prod_{i \in S} \epsilon_{i,2} \prod_{i \notin S} \epsilon_{i,1}$ for every $S \subseteq [k+1]$. On the other hand, write $\mathbf{C}^{[k]} = \text{Cov}(v_k)$, by Eq. (11), we have

$$\mathbf{C}^{[k+1]} = \begin{pmatrix} \sigma_{11}\mathbf{C}^{[k]} & \sigma_{12}\mathbf{C}^{[k]} \\ \sigma_{12}\mathbf{C}^{[k]} & \sigma_{22}\mathbf{C}^{[k]}. \end{pmatrix} \tag{14}$$

Let $S, T \subseteq [k+1]$ be given, there are four cases in total: i) $S, T \subseteq [k]$; ii) $S, T \nsubseteq [k]$; iii) $S \subseteq [k]$, $T \nsubseteq [k]$; and iv) $S \nsubseteq [k]$, $T \subseteq [k]$. For the first case, combing Eqs. (13), (14) and (12), there is

$$\mathbf{C}^{[k+1]}(S, T) = \sigma_{11}\mathbf{C}^{[k]}(S, T) = \sigma_{11}^{|[k+1] \setminus (S \cup T)|} \cdot \sigma_{12}^{|(S \setminus T) \cup (T \setminus S)|} \cdot \sigma_{22}^{|S \cap T|}.$$

The rest of the cases can be done similarly. □

## C  Prerequisites and Proof for Theorem 1

To prove Theorem 1, our first step is to lift each $\phi(v_n; P)$ into $\mathbb{R}^{2^n}$ w.r.t. the binary ordering. Precisely, we extend each semi-value $\phi_P$, a shorthand for $\phi(\cdot; P)$, to be a semi-index $\mathcal{I}_P$ that maps each $v \in \mathcal{G}_n$ (for every $n \geq 1$) into another utility function in $\mathcal{G}_n$, whose matrix representations on all $\mathcal{G}_n$ are inductive. Then, we prove a theorem of robustness for semi-indices, which is then used to prove Theorem 1.

### C.1  Extend Semi-values to Be Semi-indices

Index is another concept developed in cooperative game theory that serves as an extension for value (Fujimoto et al. 2006; Grabisch and Roubens 1999), which has also been employed in the community of machine learning (Sundararajan et al. 2020; Zhang et al. 2021). Specifically, an index $\mathcal{I}$ is a function that maps each utility function $v \in \mathcal{G}_n$ (for every $n \geq 1$) into another utility function in $\mathcal{G}_n$. Note that each utility function $v \in \mathcal{G}_n$ can be viewed as a vector indexed by all subsets of $[n]$. In other words, $\mathcal{I}$ is to impute contributions to all possible subsets of $[n]$, not just to each datum in $[n]$, so as to account for the contributions of interactions among all data $[n]$. For convenience, we write $\mathcal{I}(v_n, S)$ for the value of $\mathcal{I}(v_n)$ indexed at $S$.

Notably, the axiom of recursion was proposed by Grabisch and Roubens (1999) to extend the Banzhaf and Shapley values into the Banzhaf and Shapley indices, respectively. To state the axiom of recursion, we need further definitions. Suppose $v \in \mathcal{G}_n$ and $S \subseteq [n]$ are given. The restriction of $v$ on $S$, denoted by $v_S$, is defined by

$$v_S(T) = v(T), \forall T \subseteq S. \tag{15}$$

In other words, the domain $2^{[n]}$ of $v$ is restricted to be $2^S$.

Plus, let $j \in [n] \setminus S$ be given, the restriction of $v$ on $S$ in the presence of $j$, denoted by $v_S^{\cup j}$, is defined by

$$v_S^{\cup j}(T) = v(T \cup j), \forall T \subseteq S. \tag{16}$$

**Remark 3.** *The definition of $v_S^{\cup j}$ is not the same as the one defined by Grabisch and Roubens (1999), which is instead $v_S^{\cup j}(T) = v(T \cup j) - v(j), \forall T \subseteq S$. The term $-v(j)$ is only to make $v_S^{\cup j}(\emptyset) = 0$, and we comment that it is not necessary for all the results therein. Moreover, this extra term will make Eq.* (17) *false.*

1. **Recursion**: For every $v \in \mathcal{G}_n$, $S \subseteq [n]$ with $s > 1$, and $j \in S$, there is

$$\mathcal{I}(v, S) = \mathcal{I}(v_{[n]\setminus j}^{\cup j}, S \setminus j) - \mathcal{I}(v_{[n]\setminus j}, S \setminus j).$$

The axiom of recursion dictates that the contribution imputed to the subset $S$ should be the difference of the contributions imputed to $S \setminus j$ in the presence and absence of any player $j \in S$. So, starting from $\mathcal{I}_P(v_n, i) = \phi_i(v_n; P)$ for every $v_n \in \mathcal{G}$ and $i \in [n]$, the axiom of recursion can be used to define $\mathcal{I}_P(v_n, S)$ recursively for every *non-empty* subsets $S \subseteq [n]$. Particularly, the defined value of $\mathcal{I}(v_n, S)$ is independent of the choice of $j \in S$.

One thing that remains is to determine the contribution for the empty subset. For each semi-value $\phi_P$, we argue that it should be what is stated as the axiom of empty.

2. **Empty**: To extend a semi-value $\phi_P$ to be a semi-index $\mathcal{I}_P$, there should be $\mathcal{I}_P(v_n, \emptyset) = \sum_{S \subseteq [n]} p_{s+1}^{n+1} \cdot v(S)$ for every $v_n \in \mathcal{G}$. Recall that $p_{s+1}^{n+1} = \int_0^1 t^s (1-t)^{n-s} dP(t)$ for every $0 \le s \le n$.

Observe that $\sum_{S \subseteq [n]} p_{s+1}^{n+1} = \sum_{s=0}^{n} \binom{n}{s} p_{s+1}^{n+1} = 1$, the imposed value for $\mathcal{I}_P(v_n, \emptyset)$ is just the expected value of $v_n$ w.r.t. the probability distribution $(p_{s+1}^{n+1})_{S \subseteq [n]}$ induced by the underlying $P \in \mathcal{P}$. One can verify the axiom of empty consistently extends the axiom of recursion, i.e.,

$$\mathcal{I}_P(v, i) = \mathcal{I}_P(v_{[n]\setminus i}^{\cup i}, \emptyset) - \mathcal{I}_P(v_{[n]\setminus i}, \emptyset) \text{ for every } v \in \mathcal{G}_n \text{ and } i \in [n]. \tag{17}$$

**Definition 1** (Semi-indices). *A semi-index $\mathcal{I}_P$ is extended from a semi-value $\phi_P$ using the axioms of recursion and empty.*

**Remark 4.** *We can therefore extend each $w$-weighted Banzhaf value to be $w$-weighted Banzhaf index. Particularly, 0-weighted Banzhaf index is exactly the Möbius transform Eq.* (18)*, while 1-weighted Banzhaf index is instead the co-Möbius transform (Grabisch et al. 2000, Eq. (12)). In addition, extending the Shapley value and the Banzhaf value in this way will yield the Shapley interaction index and the Banzhaf interaction index presented in (Grabisch and Roubens 1999, Theorem 3).*

**Proposition 5** (Formulas for Semi-indices). *For each semi-index $\mathcal{I}_P$, there is*

$$\mathcal{I}_P(v_n, S) = \sum_{R \subseteq [n]\setminus S} p_{r+1}^{n+1-s} \cdot \Delta_S v_n(R) \text{ for every } v_n \in \mathcal{G} \text{ and } S \subseteq [n]$$

$$\text{where } p_j^m = \int_0^1 t^{j-1}(1-t)^{m-j} dP(t) \text{ for } 1 \le j \le m$$

$$\text{and } \Delta_S v_n(R) = \sum_{T \subseteq S} (-1)^{s-t} v_n(R \cup T).$$

*Proof.* For simplicity, we write $v$ instead of $v_n$. It is clear that it holds when $s = 0$ and $s = 1$. Assume it holds for all $S$ such that $|S| = k$. Let $S \subseteq [n]$ that satisfies $|S| = k + 1$, and some $i \in S$ be given, there is

$\mathcal{I}_P(v_n, S)$

$= \mathcal{I}_P(v_{[n]\setminus i}^{\cup i}, S\setminus i) - \mathcal{I}_P(v_{[n]\setminus i}, S\setminus i)$          the axiom of recursion

$= \sum_{R \subseteq [n]\setminus S} p_{r+1}^{n-k} \cdot \Delta_{S\setminus i} v_{[n]\setminus i}^{\cup i}(R) - \sum_{R \subseteq [n]\setminus S} p_{r+1}^{n-k} \cdot \Delta_{S\setminus i} v_{[n]\setminus i}(R)$    by inductive assumption

$= \sum_{R \subseteq [n]\setminus S} p_{r+1}^{n-k} \cdot (\Delta_{S\setminus i} v(R \cup i) - \Delta_{S\setminus i} v(R))$

$= \sum_{R \subseteq [n]\setminus S} p_{r+1}^{n+1-s} \cdot \Delta_S v(R).$

Note the above reasoning is independent of the choice of $i \in S$. $\qquad\qquad\square$

The reader can also verify that each semi-index $\mathcal{I}_P$ satisfies the following axioms, the first three of which are often laid out for how indices should be in the fields of cooperative game theory and machine learning (Fujimoto et al. 2006; Grabisch and Roubens 1999; Sundararajan et al. 2020).

3. **Linearity**: $\mathcal{I}_P(a \cdot u_n + v_n, S) = a \cdot \mathcal{I}_P(u_n, S) + \mathcal{I}_P(v_n, S)$ for every $v_n, u_n \in \mathcal{G}$, $a \in \mathbb{R}$ and $S \subseteq [n]$.

4. **Dummy Interaction**: If $S \subseteq [n]$ with $s > 1$ contains a datum $i$ such that for some constant $C$, $v_n(T \cup i) = v_n(T) + C$ for all $T \subseteq [n] \backslash i$, then $\mathcal{I}_P(v_n, S) = 0$.

5. **Symmetry**: $\mathcal{I}_P(v_n \circ \pi, S) = \mathcal{I}_P(v_n, \pi(S))$ for every utility function $v_n \in \mathcal{G}$, every permutation $\pi$ of $[n]$ and every $S \subseteq [n]$.

6. **Consistency**: For each utility function $v_n \in \mathcal{G}$, its extension $\overline{v}_{n+1}$ is defined by $\overline{v}_{n+1}(S) = v_n(S \cap [n])$ for every $S \subseteq [n+1]$. Then, $\mathcal{I}_P(\overline{v}_{n+1}, S) = \mathcal{I}_P(v_n, S)$ for every $S \subseteq [n]$.

## C.2 Interaction Transform

The concept of interaction transform is crucial to establish well-structured matrix representations for each semi-index $\mathcal{I}_P$. Specifically, Definitions 3 and 4 are inspired by (Denneberg and Grabisch 1999).

**Definition 2** (Möbius Transform). *The Möbius transform $\mu$ is a function that maps each utility function $v \in \mathcal{G}_n$ (for every $n \geq 1$) into another utility function in $\mathcal{G}_n$. For each $v_n \in \mathcal{G}$, we write $\mu(v_n, S)$ for the value of $\mu(v_n)$ indexed at $S \subseteq [n]$. The definition is*

$$\mu(v_n, S) = \sum_{R \subseteq S} (-1)^{s-r} v_n(R) \ \text{for every } S \subseteq [n]. \tag{18}$$

**Definition 3** (Convolution). *The convolution $\iota \star u$ between each $\iota \in \mathbb{R}^{\mathbb{N}}$ and each $u \in \mathcal{G}_n$ is defined by*

$$(\iota \star u)(S) = \sum_{[n] \supseteq T \supseteq S} \iota(t-s) u(T) \ \text{for every } S \subseteq [n]. \tag{19}$$

*In other words, each $\iota \in \mathbb{R}^{\mathbb{N}}$ may be seen as a function that maps each utility function $v \in \mathcal{G}_n$ (for every $n \geq 1$) into another utility function in $\mathcal{G}_n$, and thus we may write $\iota(u)$ instead of $\iota \star u$.*

**Definition 4** (Interaction Transform). *For each semi-index $\mathcal{I}_P$, its interaction transform is defined to be a vector $\iota_P \in \mathbb{R}^{\mathbb{N}}$ that satisfies*

$$\mathcal{I}_P = \iota_P \circ \mu.$$

**Proposition 6** (Interaction Transforms for Semi-indices). *For each semi-index $\mathcal{I}_P$, its interaction transform $\iota_P \in \mathbb{R}^{\mathbb{N}}$ is*

$$\iota_P(i) = \int_0^1 t^i \mathrm{d}P(t) \ \text{for every } i \in \mathbb{N}. \tag{20}$$

*Here, $\iota_P(i)$ refers to the $i$-th entry of $\iota_P$.*

*Proof.* Let $v \in \mathcal{G}_n$ be given, and for simplicity, we write $\mu$ instead for $\mu(v)$. For every $S \subseteq [n]$, there is

$$\mathcal{I}_P(v, S) = \sum_{T \subseteq [n] \backslash S} p_{t+1}^{n+1-s} \Delta_S v(T) = \sum_{T \subseteq [n] \backslash S} p_{t+1}^{n+1-s} \sum_{R \subseteq T} \mu(S \cup R)$$

$$= \sum_{R \subseteq [n] \backslash S} \sum_{R \subseteq T \subseteq [n] \backslash S} p_{t+1}^{n+1-s} \mu(S \cup R) = \sum_{[n] \supseteq W \supseteq S} \sum_{W \backslash S \subseteq T \subseteq [n] \backslash S} p_{t+1}^{n+1-s} \mu(W)$$

$$= \sum_{[n] \supseteq W \supseteq S} \left( \sum_{i=0}^{n-w} \binom{n-w}{i} p_{w-s+i+1}^{n+1-s} \right) \mu(W).$$

We refer the reader to (Grabisch et al. 2000, Eq. (10)) for the equality $\Delta_S v(T) = \sum_{R \subseteq T} \mu(S \cup R)$ used in the second equality.

Since $p_k^m = \int_0^1 t^{k-1}(1-t)^{m-k}$ for $1 \le k \le m$, there is

$$\sum_{i=0}^{n-w} \binom{n-w}{i} p_{w-s+i+1}^{n+1-s} = \int_0^1 t^{w-s} \mathrm{d}P(t).$$

Therefore, we have

$$\mathcal{I}_P(v, S) = \sum_{[n] \supseteq W \supseteq S} \left( \int_0^1 t^{w-s} \mathrm{d}P(t) \right) \mu(W) = \sum_{[n] \supseteq W \supseteq S} \iota_P(w-s)\mu(W) = [\iota_P \star \mu(v)](S).$$

$\square$

### C.3 Matrix Representation for Semi-indices

From now on, let $2^{[n]}$ be ordered w.r.t. the binary ordering. Then, $v_n \in \mathcal{G}$ is a vector in $\mathbb{R}^{2^n}$, and so is $\mathcal{I}_P(v_n)$. Since $\mathcal{I}_P$ is linear in $v_n \in \mathcal{G}_n$, $\mathcal{I}_P$ has a matrix representation on each $\mathcal{G}_n$.

**Proposition 7** (Matrix Representations for Semi-indices). *Let $2^{[n]}$ be ordered using the binary ordering, the matrix representation of semi-index $\mathcal{I}_P$ on $\mathcal{G}_n$ w.r.t. the standard basis is*

$$\mathbf{M}_P^{[n]} = \int_0^1 \mathbf{B}^{[n]}(t)\mathrm{d}P(t)$$

*where*

$$\mathbf{B}^{[n]}(t) = \overbrace{\mathbf{B}^{[1]}(t) \otimes \mathbf{B}^{[1]}(t) \otimes \cdots \otimes \mathbf{B}^{[1]}(t)}^{n \text{ repetitions}} \text{ and } \mathbf{B}^{[1]}(t) = \begin{pmatrix} 1-t & t \\ -1 & 1 \end{pmatrix}.$$

*Here, $\otimes$ is the Kronecker product.*

*Proof.* Since $\mathcal{I}_P = \iota_P \circ \mu$, suffice it to find the matrix representations for both $\mu$ and $\iota_P$ w.r.t. the standard basis in $\mathcal{G}_n$.

As provided by Grabisch et al. (2000, Eq. (45)), which can be verified by induction using Eq. (13), the matrix representation for the Möbius transform $\mu$ on $\mathcal{G}_n$ w.r.t. the standard basis is

$$\mathbf{W}^{[n]} = \overbrace{\mathbf{W}^{[1]} \otimes \mathbf{W}^{[1]} \otimes \cdots \otimes \mathbf{W}^{[1]}}^{n \text{ repetitions}} \text{ where } \mathbf{W}^{[1]} = \begin{pmatrix} 1 & 0 \\ -1 & 1 \end{pmatrix}$$

We first prove by induction that the matrix representation of $\iota_P$ on $\mathcal{G}_n$ w.r.t. the standard basis is

$$\mathbf{T}_P^{[n]} = \int_0^1 \mathbf{A}^{[n]}(t)\mathrm{d}P(t) \tag{21}$$

where

$$\mathbf{A}^{[n]}(t) = \overbrace{\mathbf{A}^{[1]}(t) \otimes \mathbf{A}^{[1]}(t) \otimes \cdots \otimes \mathbf{A}^{[1]}(t)}^{n \text{ repetitions}} \text{ and } \mathbf{A}^{[1]}(t) = \begin{pmatrix} 1 & t \\ 0 & 1 \end{pmatrix}$$

It is clear that it holds for $n = 1$. Assume it holds for some $k > 0$, i.e., for every $u^{[k]} \in \mathcal{G}_k$

$$\iota_P \star u^{[k]} = \mathbf{T}_P^{[k]} u^{[k]}, \tag{22}$$

and we proceed with $k + 1$. Let $u^{[k+1]} \in \mathcal{G}_{k+1}$ and $S \subseteq [k + 1]$ be given, there are two cases: i) $S \subseteq [k]$ and ii) $S \nsubseteq [k]$. If $S \subseteq [k]$, by Eq. (19),

$$(\iota_P \star u^{[k+1]})(S)$$

$$= \sum_{[k+1] \supseteq T \supseteq S} \iota_P(t - s) u^{[k+1]}(T)$$

$$= \sum_{[k] \supseteq T \supseteq S} \iota_P(t - s) u^{[k+1]}(T) + \sum_{[k] \nsupseteq T \supseteq S} \iota_P(t - s) u^{[k+1]}(T)$$

$$= \sum_{[k] \supseteq T \supseteq S} \iota_P(t - s) u^{[k+1]}_{[k]}(T) + \sum_{[k] \nsupseteq T \supseteq S} \iota_P(t - s) u^{[k+1]}(T) \qquad \text{by Eq. (15)}$$

$$= [\mathbf{T}_P^{[k]} u^{[k+1]}_{[k]}](S) + \sum_{[k] \nsupseteq T \supseteq S} \iota_P(t - s) u^{[k+1]}(T) \qquad \text{by Eq. (22)}$$

$$= [\mathbf{T}_P^{[k]} u^{[k+1]}_{[k]}](S) + \sum_{[k] \supseteq R \supseteq S} \iota_P(r + 1 - s) u^{[k+1], \cup \{k+1\}}_{[k]}(R) \qquad \text{by } R = T \backslash \{k + 1\} \text{ and Eq. (16)}$$

$$= [\mathbf{T}_P^{[k]} u^{[k+1]}_{[k]}](S) + \left[ \left( \int_0^1 t \cdot \mathbf{A}^{[k]}(t) \mathrm{d}P(t) \right) u^{[k+1], \cup \{k+1\}}_{[k]} \right](S) \qquad \text{by Eqs. (20), (21) and (22)}$$

$$= [\mathbf{T}_P^{[k+1]} u^{[k+1]}](S) \qquad \text{by Eq. (13).}$$

The remaining case can be done in a similar fashion. After all, the matrix representation of $\iota_P$ on $\mathcal{G}_n$ w.r.t. the standard basis is

$$\mathbf{M}_P^{[n]} = \mathbf{T}_P^{[n]} \times \mathbf{W}^{[n]}$$

$$= \int_0^1 \overbrace{\mathbf{A}^{[1]}(t) \otimes \mathbf{A}^{[1]}(t) \otimes \cdots \otimes \mathbf{A}^{[1]}(t)}^{n \text{ repetitions}} \mathrm{d}P(t) \times (\overbrace{\mathbf{W}^{[1]} \otimes \mathbf{W}^{[1]} \otimes \cdots \otimes \mathbf{W}^{[1]}}^{n \text{ repetitions}})$$

$$= \int_0^1 (\mathbf{A}^{[1]}(t) \otimes \mathbf{A}^{[1]}(t) \otimes \cdots \otimes \mathbf{A}^{[1]}(t)) \times (\mathbf{W}^{[1]} \otimes \mathbf{W}^{[1]} \otimes \cdots \otimes \mathbf{W}^{[1]}) \mathrm{d}P(t)$$

$$= \int_0^1 \mathbf{B}^{[1]}(t) \otimes \mathbf{B}^{[1]}(t) \otimes \cdots \otimes \mathbf{B}^{[1]}(t) \mathrm{d}P(t)$$

where

$$\mathbf{B}^{[1]}(t) = \mathbf{A}^{[1]}(t) \times \mathbf{W}^{[1]}.$$

$\square$

## C.4   A Theorem of Robustness for Semi-indices

**Definition 5** (Truncated Semi-indices). *Given a semi-index $\mathcal{I}_P$ and a non-empty collection of subsets $\mathcal{S} \subseteq 2^{[n]}$, the truncated semi-index $\mathcal{I}_{P,\mathcal{S}}$ is obtained by only retaining the values of $\mathcal{I}_P$ indexed by $\mathcal{S}$.*

**Theorem 2.** *Under Assumption 1, and further assume $\sigma_{12} < \min(\sigma_{11}, \sigma_{22})$, for every non-empty collection of subsets $\mathcal{S} \subseteq 2^{[n]}$, the Dirac delta distribution $\delta_{w^*}$ with*

$$w^* = \frac{\sigma_{11} - \sigma_{12}}{\sigma_{11} + \sigma_{22} - 2\sigma_{12}}$$

*is optimal to the optimization problem*

$$\underset{P \in \mathcal{P}}{\arg\min} \det \left( \mathrm{Cov} \left( \mathcal{I}_{P,\mathcal{S}}(v_n) \right) \right)$$

$$\text{s.t. } \mathrm{Cov}(v_n) = \boldsymbol{\Sigma} \otimes \boldsymbol{\Sigma} \otimes \cdots \otimes \boldsymbol{\Sigma} \text{ (n repetitions).}$$

*Proof.* The matrix representation of $\mathcal{I}_P$ on $\mathcal{G}_n$ w.r.t. the standard basis is $\mathbf{M}_P^{[n]} = \mathbf{T}_P^{[n]} \times \mathbf{W}^{[n]}$, as presented in Proposition 7. For simplicity, we write $\mathbf{V}_{[n],P} = \mathrm{Cov}(\mathcal{I}_P(v_n))$. Since $\mathcal{I}_P$ is linear in $v_n$, there is

$$\mathbf{V}_{[n],P} = \mathbf{M}_P^{[n]} \mathbf{C}^{[n]} (\mathbf{M}_P^{[n]})^\top. \tag{23}$$

Note that $\det(\mathbf{T}_P^{[n]}) = \det(\mathbf{W}^{[n]}) = 1$ as they are triangular matrices with all diagonal entries being one, and thus we have

$$\det(\mathbf{M}_P^{[n]}) = 1, \tag{24}$$

$$\det(\mathbf{V}_{[n],P}) = \det(\mathbf{C}^{[n]}). \tag{25}$$

Observe that $\mathrm{Cov}(\mathcal{I}_{P,\mathcal{S}}(v_n))$ is just a sub-matrix of $\mathbf{V}_{[n],P}$. To represent such sub-matrices, for every non-empty $\alpha, \beta \subseteq [2^n]$, $\mathbf{V}_{[n],P}^{\alpha,\beta}$ is defined to be the sub-matrix of $\mathbf{V}_{[n],P}$ whose rows and columns are indexed by $\alpha$ and $\beta$, respectively. Plus, we write $\mathbf{V}_{[n],P}^{\alpha}$ and $\mathbf{V}_{[n],P}^{:,\alpha}$ instead for $\mathbf{V}_{[n],P}^{\alpha,\alpha}$ and $\mathbf{V}_{[n],P}^{[2^n],\alpha}$, respectively. Therefore, for each $\mathcal{I}_{P,\mathcal{S}}(v_n)$, there exists a non-empty $\alpha \subseteq [2^n]$ such that $\mathbf{V}_{[n],P}^{\alpha} = \mathrm{Cov}(\mathcal{I}_{P,\mathcal{S}}(v_n))$.

We prove by induction that for every $n > 0$, it holds that for every non-empty $\alpha \subseteq [2^n]$ and for every $P \in \mathcal{P}$, $\det(\mathbf{V}_{[n],P}^{\alpha}) \geq \det(\mathbf{V}_{[n],\delta_{w^*}}^{\alpha})$.

For $n = 1$, let $s = \int_0^1 t \mathrm{d}P(t)$, there is

$$\mathbf{V}_{[1],P} = \begin{pmatrix} \sigma_{11}(1-s)^2 + \sigma_{22}s^2 - 2\sigma_{12}s(1-s) & (\sigma_{11} + \sigma_{22} - 2\sigma_{12})s + \sigma_{12} - \sigma_{11} \\ (\sigma_{11} + \sigma_{22} - 2\sigma_{12})s + \sigma_{12} - \sigma_{11} & \sigma_{11} + \sigma_{22} - 2\sigma_{12} \end{pmatrix},$$

by which one can verify the statement by noticing that the function $s \mapsto \sigma_{11}(1-s)^2 + \sigma_{22}s^2 - 2\sigma_{12}s(1-s)$ achieves its minimum at point $s = w^*$. Suppose the statement holds for some $k > 0$, and we proceed for $k + 1$.

Before proceeding with the induction, we have to explore the Kronecker structure between $\mathbf{V}_{[k+1],P}$ and $\mathbf{V}_{[k],P}$. By Eq. (23), and Proposition 7,

$$\mathbf{V}_{[k+1],P} = \left( \int_0^1 \begin{pmatrix} (1-t)\mathbf{B}^{[k]}(t) & t\mathbf{B}^{[k]}(t) \\ -\mathbf{B}^{[k]}(t) & \mathbf{B}^{[k]}(t) \end{pmatrix} \mathrm{d}P(t) \right) \times \begin{pmatrix} \sigma_{11}\mathbf{C}^{[k]} & \sigma_{12}\mathbf{C}^{[k]} \\ \sigma_{12}\mathbf{C}^{[k]} & \sigma_{22}\mathbf{C}^{[k]} \end{pmatrix}$$
$$\times \left( \int_0^1 \begin{pmatrix} (1-t)\mathbf{B}^{[k]}(t) & t\mathbf{B}^{[k]}(t) \\ -\mathbf{B}^{[k]}(t) & \mathbf{B}^{[k]}(t) \end{pmatrix} \mathrm{d}P(t) \right)^{\top}. \tag{26}$$

Denote $\mathbf{L}_P^{[k]} = \int_0^1 t\mathbf{B}^{[k]}(t)\mathrm{d}P(t)$, Eq. (26) can be rewritten as

$$\mathbf{V}_{[k+1],P} = \begin{pmatrix} \mathbf{M}_P^{[k]} - \mathbf{L}_P^{[k]} & \mathbf{L}_P^{[k]} \\ -\mathbf{M}_P^{[k]} & \mathbf{M}_P^{[k]} \end{pmatrix} \begin{pmatrix} \sigma_{11}\mathbf{C}^{[k]} & \sigma_{12}\mathbf{C}^{[k]} \\ \sigma_{12}\mathbf{C}^{[k]} & \sigma_{22}\mathbf{C}^{[k]} \end{pmatrix} \begin{pmatrix} \mathbf{M}_P^{[k]} - \mathbf{L}_P^{[k]} & \mathbf{L}_P^{[k]} \\ -\mathbf{M}_P^{[k]} & \mathbf{M}_P^{[k]} \end{pmatrix}^{\top}$$
$$= \begin{pmatrix} \mathbf{U} & \mathbf{F}^{\top} \\ \mathbf{F} & \mathbf{E} \end{pmatrix} \tag{27}$$

where

$$\mathbf{U} = \sigma_{11}(\mathbf{M}_P^{[k]} - \mathbf{L}_P^{[k]})\mathbf{C}^{[k]}(\mathbf{M}_P^{[k]} - \mathbf{L}_P^{[k]})^{\top} + \sigma_{22}\mathbf{L}_P^{[k]}\mathbf{C}^{[k]}(\mathbf{L}_P^{[k]})^{\top}$$
$$+ \sigma_{12}(\mathbf{M}_P^{[k]} - \mathbf{L}_P^{[k]})\mathbf{C}^{[k]}(\mathbf{L}_P^{[k]})^{\top} + \sigma_{12}\mathbf{L}_P^{[k]}\mathbf{C}^{[k]}(\mathbf{M}_P^{[k]} - \mathbf{L}_P^{[k]})^{\top}, \tag{28}$$

$$\mathbf{F} = \mathbf{M}_P^{[k]}\mathbf{C}^{[k]}\mathbf{K}, \tag{29}$$

$$\mathbf{K} = (\sigma_{12} - \sigma_{11})(\mathbf{M}_P^{[k]} - \mathbf{L}_P^{[k]})^{\top} + (\sigma_{22} - \sigma_{12})(\mathbf{L}_P^{[k]})^{\top}, \tag{30}$$

$$\mathbf{E} = (\sigma_{11} + \sigma_{22} - 2\sigma_{12})\mathbf{V}_{[k],P}. \tag{31}$$

Our goal is to simplify the expression

$$\mathbf{S} = \mathbf{U} - \mathbf{F}^{\top}\mathbf{E}^{-1}\mathbf{F}.$$

Note that $\mathbf{E}$ and $\mathbf{M}_P^{[k]}$ are invertible by Eqs. (24) and (25) and that $\boldsymbol{\Sigma}$ is assumed to be positive definite (and thus $\det(\mathbf{C}^{[n]}) > 0$). Observe that $\sigma_{11} + \sigma_{22} - 2\sigma_{12} > 0$ by assumption, using Eqs. (23) and (31), Eq. (29) is equal to

$$\mathbf{F} = \mathbf{E}\left( \frac{1}{\sigma_{11} + \sigma_{22} - 2\sigma_{12}}((\mathbf{M}_P^{[k]})^{\top})^{-1}\mathbf{K} \right),$$

which indicates that

$$\mathbf{F}^\top \mathbf{E}^{-1} \mathbf{F}$$

$$= \left( \frac{1}{\sigma_{11} + \sigma_{22} - 2\sigma_{12}} \mathbf{K}^\top (\mathbf{M}_P^{[k]})^{-1} \right) \mathbf{E}\mathbf{E}^{-1}\mathbf{E} \left( \frac{1}{\sigma_{11} + \sigma_{22} - 2\sigma_{12}} ((\mathbf{M}_P^{[k]})^\top)^{-1} \mathbf{K} \right)$$

$$= \frac{1}{(\sigma_{11} + \sigma_{22} - 2\sigma_{12})^2} \mathbf{K}^\top (\mathbf{M}_P^{[k]})^{-1} \mathbf{E} ((\mathbf{M}_P^{[k]})^\top)^{-1} \mathbf{K}$$

$$= \frac{1}{\sigma_{11} + \sigma_{22} - 2\sigma_{12}} \mathbf{K}^\top \mathbf{C}^{[k]} \mathbf{K}.$$

The last equality is obtained by using Eqs. (23) and (31) again. Write $\mathbf{R} = \mathbf{M}_P^{[k]} - \mathbf{L}_P^{[k]}$. From Eqs. (28) and (30), there is

$$\mathbf{U} = \sigma_{11} \mathbf{R}\mathbf{C}^{[k]}\mathbf{R}^\top + \sigma_{22}\mathbf{L}_P^{[k]}\mathbf{C}^{[k]}(\mathbf{L}_P^{[k]})^\top + \sigma_{12}(\mathbf{R}\mathbf{C}^{[k]}(\mathbf{L}_P^{[k]})^\top + \mathbf{L}_P^{[k]}\mathbf{C}^{[k]}\mathbf{R}^\top),$$

$$\mathbf{K}^\top \mathbf{C}^{[k]} \mathbf{K} = (\sigma_{12} - \sigma_{11})^2 \mathbf{R}\mathbf{C}^{[k]}\mathbf{R}^\top + (\sigma_{22} - \sigma_{12})^2 \mathbf{L}_P^{[k]}\mathbf{C}^{[k]}(\mathbf{L}_P^{[k]})^\top$$

$$+ (\sigma_{12} - \sigma_{11})(\sigma_{22} - \sigma_{12})(\mathbf{R}\mathbf{C}^{[k]}(\mathbf{L}_P^{[k]})^\top + \mathbf{L}_P^{[k]}\mathbf{C}^{[k]}\mathbf{R}^\top).$$

Notice that

$$(\sigma_{11} + \sigma_{22} - 2\sigma_{12})\mathbf{U} - \mathbf{K}^\top \mathbf{C}^{[k]} \mathbf{K}$$

$$= (\sigma_{11}\sigma_{22} - \sigma_{12}^2) \left( \mathbf{R}\mathbf{C}^{[k]}\mathbf{R}^\top + \mathbf{L}_P^{[k]}\mathbf{C}^{[k]}(\mathbf{L}_P^{[k]})^\top + \mathbf{R}\mathbf{C}^{[k]}(\mathbf{L}_P^{[k]})^\top + \mathbf{L}_P^{[k]}\mathbf{C}^{[k]}\mathbf{R}^\top \right)$$

$$= (\sigma_{11}\sigma_{22} - \sigma_{12}^2) \left( (\mathbf{R} + \mathbf{L}_P^{[k]})\mathbf{C}^{[k]}(\mathbf{R}^\top + (\mathbf{L}_P^{[k]})^\top) \right) \tag{32}$$

$$= (\sigma_{11}\sigma_{22} - \sigma_{12}^2)\mathbf{M}_P^{[k]}\mathbf{C}^{[k]}(\mathbf{M}_P^{[k]})^\top.$$

Combining Eqs. (32) and (23), we eventually get the simplified formula for $\mathbf{S}$,

$$\mathbf{S} = \frac{\sigma_{11}\sigma_{22} - \sigma_{12}^2}{\sigma_{11} + \sigma_{22} - 2\sigma_{12}} \mathbf{V}_{[k],P}. \tag{33}$$

Particularly, $\sigma_{11}\sigma_{22} - \sigma_{12}^2 > 0$ as $\mathbf{\Sigma}$ is assumed to be positive definite. So far so good, we are ready to complete the induction by cases while looking into Eq. (27).

Case 1: If $[2^{k+1}]\backslash[2^k] \subseteq \alpha \subseteq [2^{k+1}]$, let $\beta = \alpha\backslash([2^{k+1}]\backslash[2^k])$, which could be empty. Then, there is

$$\mathbf{V}_{[k+1],P}^\alpha = \begin{pmatrix} \mathbf{U}^\beta & (\mathbf{F}^{:,\beta})^\top \\ \mathbf{F}^{:,\beta} & (\sigma_{11} + \sigma_{22} - 2\sigma_{12})\mathbf{V}_{[k],P} \end{pmatrix}.$$

Note that we can write

$$\mathbf{F}^{:,\beta} = \mathbf{F}\mathbf{D}_\beta \tag{34}$$

where $\mathbf{D}_\beta$ is the corresponding column-selecting matrix. It also gives that

$$\mathbf{D}_\beta^\top \mathbf{U}\mathbf{D}_\beta = \mathbf{U}^\beta. \tag{35}$$

Using Eqs. (33), (34) and (35), according to Lemma 2, there is

$$\det(\mathbf{V}_{[k+1],P}^\alpha)$$

$$= \det\left((\sigma_{11} + \sigma_{22} - 2\sigma_{12})\mathbf{V}_{[k],P}\right) \det\left(\mathbf{U}^\beta - (\mathbf{F}^{:,\beta})^\top ((\sigma_{11} + \sigma_{22} - 2\sigma_{12})\mathbf{V}_{[k],P})^{-1}\mathbf{F}^{:,\beta}\right)$$

$$= \det\left((\sigma_{11} + \sigma_{22} - 2\sigma_{12})\mathbf{V}_{[k],P}\right) \det(\mathbf{S}^\beta) = L\det(\mathbf{S}^\beta)$$

$$= L\det\left( \frac{\sigma_{11}\sigma_{22} - \sigma_{12}^2}{\sigma_{11} + \sigma_{22} - 2\sigma_{12}} \mathbf{V}_{[k],P}^\beta \right)$$

where $L = \det\left((\sigma_{11} + \sigma_{22} - 2\sigma_{12})\mathbf{V}_{[k],P}\right) = (\sigma_{11} + \sigma_{22} - 2\sigma_{12})^{2^k} \det(\mathbf{C}^{[k]})$ by Eq. (25).

Therefore, by inductive assumption, for every $P \in \mathcal{P}$,

$$\det(\mathbf{V}_{[k+1],P}^\alpha) = L\det\left( \frac{\sigma_{11}\sigma_{22} - \sigma_{12}^2}{\sigma_{11} + \sigma_{22} - 2\sigma_{12}} \mathbf{V}_{[k],P}^\beta \right)$$

$$\geq L\det\left( \frac{\sigma_{11}\sigma_{22} - \sigma_{12}^2}{\sigma_{11} + \sigma_{22} - 2\sigma_{12}} \mathbf{V}_{[k],\delta_{w^*}}^\beta \right) = \det(\mathbf{V}_{[k+1],\delta_{w^*}}^\alpha). \tag{36}$$

Case 2: If $[2^{k+1}]\backslash[2^k] \not\subseteq \alpha \subseteq [2^{k+1}]$ and $\alpha \neq \emptyset$, let $\beta = [2^{k+1}]\backslash[2^k]$ and $\gamma = \alpha \cap \beta$. Note that $\gamma$ could be empty. By Lemma 5, there is

$$\det((\sigma_{11} + \sigma_{22} - 2\sigma_{12})\mathbf{V}_{[k],P}^{\gamma-2^k}) \det(\mathbf{V}_{[k+1],P}^{\beta\cup\alpha}) \leq \det(\mathbf{V}_{[k+1],P}^{\alpha}) \det((\sigma_{11} + \sigma_{22} - 2\sigma_{12})\mathbf{V}_{[k],P}),$$

which implies, for every $P \in \mathcal{P}$,

$$
\begin{aligned}
\det(\mathbf{V}_{[k+1],P}^{\alpha}) &\geq \frac{\det((\sigma_{11} + \sigma_{22} - 2\sigma_{12})\mathbf{V}_{[k],P}^{\gamma-2^k}) \det(\mathbf{V}_{[k+1],P}^{\beta\cup\alpha})}{(\sigma_{11} + \sigma_{22} - 2\sigma_{12})^{2^k} \det(\mathbf{C}^{[k]})} \\
&\geq \frac{\det((\sigma_{11} + \sigma_{22} - 2\sigma_{12})\mathbf{V}_{[k],\delta_{w^*}}^{\gamma-2^k}) \det(\mathbf{V}_{[k+1],\delta_{w^*}}^{\beta\cup\alpha})}{(\sigma_{11} + \sigma_{22} - 2\sigma_{12})^{2^k} \det(\mathbf{C}^{[k]})} \\
&= \det(\mathbf{V}_{[k+1],\delta_{w^*}}^{\alpha}).
\end{aligned}
\tag{37}
$$

The second inequality comes from the inductive assumption as well as case 1, while the last equality is due to that $\mathbf{V}_{[k+1],\delta_{w^*}}$ is diagonal according to Lemma 7 $\qquad\square$

## C.5 Proof for Theorem 1

**Theorem 1.** *Under Assumption 1, and further assume $\sigma_{12} < \min(\sigma_{11}, \sigma_{22})$, the Dirac delta distribution $\delta_{w^*}$ with*

$$w^* = \frac{\sigma_{11} - \sigma_{12}}{\sigma_{11} + \sigma_{22} - 2\sigma_{12}} \tag{8}$$

*is optimal to the problem* (7) *for every $n \geq 1$. In addition, the optimal solution is unique if $n > 3$.*

*Proof.* Let $\mathcal{S} = \{\{1\}, \{2\}, \dots, \{n\}\}$, then

$$\mathcal{I}_{P,\mathcal{S}}(v_n) = \phi(v_n; P).$$

Therefore, by Theorem 2, $\delta_{w^*}$ is an optimal solution. For the uniqueness, it does not hold if $n \leq 2$. For example, let

$$\mathcal{O} = \{P \in \mathcal{P} \mid \int_0^1 t dP(t) = w^*\},$$

we point out that every $P \in \mathcal{O}$ is optimal if $n = 1$ or $n = 2$. To see $|\mathcal{O}| > 1$, for example, if $\sigma_{11} = \sigma_{22}$ and $\sigma_{12} = 0$, the probability distribution for the Shapley value lies in $\mathcal{O}$. Nevertheless, we prove the uniqueness as follows: 1) first show that if $n = k$ possesses such a uniqueness, so does $n = k + 1$, and 2) then demonstrate that it holds for $n = 4$.

For the inductive part, it can be completed by reusing the induction in the proof for Theorem 2. Specifically, for case 1, let $\alpha = ([2^{k+1}]\backslash[2^k]) \cup \beta$ where $\beta$ contains all indices pointing to subsets $\{1\}, \{2\}, \dots,$ and $\{k\}$, then the inequality in Eq. (36) is strict if $P \neq \delta_{w^*}$. On the other hand, for case 2, let $\alpha$ contain all indices pointing to subsets $\{1\}, \{2\}, \dots,$ and $\{k+1\}$, the second inequality in Eq. (37) is also strict if $P \neq \delta_{w^*}$ by using the aforementioned strict inequality in case 1.

To show the uniqueness for $n = 4$, it can also be done by looking into the proof for Theorem 2. Specifically, in the induction, we focus on case 2 for $k = 3$ and $\alpha = \{2, 3, 5, 9\}$ that points to subsets $\{1\}, \{2\}, \{3\}$ and $\{4\}$. Note that $\beta = [2^4]\backslash[2^3] = \{9, 10, \dots, 16\}$ in case 2. By Lemma 5, a necessary condition for the first inequality in Eq. (37) to be equal is that each column of $\mathbf{V}_{[4],P}^{\{2,3,5\},\{10,11,\dots,16\}}$ lies in $\mathrm{span}(\mathbf{V}_{[4],P}^{\{2,3,5\},\{9\}})$. We prove that this condition can only be met by $\delta_{w^*}$, and thus the first inequality in Eq. (36) is strict if $P \neq \delta_{w^*}$.

Regarding Eq. (23), using the fact that

$$
\begin{aligned}
\mathbf{V}_{[n],P} &= \left(\int_0^1 \mathbf{B}^{[n]}(t)\mathrm{d}P(t)\right) \times \mathbf{C}^{[n]} \times \left(\int_0^1 \mathbf{B}^{[n]}(t)\mathrm{d}P(t)\right)^\top \\
&= \int_0^1 \mathrm{d}P(t) \int_0^1 \mathbf{B}^{[n]}(t)\mathbf{C}^{[n]}(\mathbf{B}^{[n]}(s))^\top \mathrm{d}P(s),
\end{aligned}
$$

one can verify that $\mathbf{V}_{[4],P}^{\{2,3,5\},\{9\}} = c_P \mathbf{1}$ for some $c_P \in \mathbb{R}$. Therefore, to meet the necessary condition, all entries in each column of $\mathbf{V}_{[4],P}^{\{2,3,5\},\{10,11,\dots,16\}}$ should be the same. Particularly, there is

$$\mathbf{V}_{[4],P}^{\{3,5\},\{12\}}$$

$$= \begin{pmatrix} \int_0^1 \mathrm{d}P(t) \int_0^1 (\sigma_{11} + \sigma_{22} - 2\sigma_{12})p(t)^2(p(s \cdot t) + (\sigma_{12} - \sigma_{11})(s + t - 1) + \sigma_{11})\mathrm{d}P(s) \\ \int_0^1 \mathrm{d}P(t) \int_0^1 p(s)p(t)^3 \mathrm{d}P(s) \end{pmatrix}$$

where $p(w) = (\sigma_{11} + \sigma_{22} - 2\sigma_{12})w + \sigma_{12} - \sigma_{11}$.

The equation $\mathbf{V}_{[4],P}^{\{3\},\{12\}} - \mathbf{V}_{[4],P}^{\{5\},\{12\}} = 0$ leads to

$$\int_0^1 (\sigma_{11}\sigma_{22} - \sigma_{12}^2)p(t)^2 \mathrm{d}P(t) = 0.$$

Set $g(w) = (\sigma_{11}\sigma_{22} - \sigma_{12}^2)p(w)^2$, since $g(w)$ attains its minimum $0$ only at the point $w^*$, by Lemma 6, the necessary condition is met if and only if $P = \delta_{w^*}$. □

For the ease of the reader to verify the calculation in Theorem 1, we include below a Matlab code to help compute.

```
syms sigma11 sigma12 sigma22 s t real;
Sigma1 = [sigma11, sigma12; sigma12, sigma22];
B1 = [1-t, t; -1, 1];
Bt = B1;
Sigma = Sigma1;
for i = 1:3
    Sigma = kron(Sigma1, Sigma);
    Bt = kron(B1, Bt);
end
Bs = subs(Bt, t, s);
V = Bt*Sigma*Bs';
disp(simplify(V([2,3,5],9))); % V^{{2,3,5},{9}}
Vp = simplify(V([3,5],12));
disp(Vp); % V^{{3,5},{12}}
disp(simplify(Vp(1)-Vp(2))); % V^{{3},{12}}-V^{{5},{12}}
```

### C.6 Lemmas

In this section, we provide all lemmas necessary to prove Theorems 2 and 1.

**Lemma 1.** *Suppose $\mathbf{A}$ is positive definite and $\mathbf{B}$ is positive semi-definite, there is*

$$\det(\mathbf{A} + \mathbf{B}) \geq \det(\mathbf{A})$$

*where the equality holds if and only if $\mathbf{B} = \mathbf{0}$.*

*Proof.* Since $\mathbf{A} + \mathbf{B} = \mathbf{A}^{\frac{1}{2}}(\mathbf{I} + \mathbf{A}^{-\frac{1}{2}}\mathbf{B}\mathbf{A}^{-\frac{1}{2}})\mathbf{A}^{\frac{1}{2}}$, there is $\det(\mathbf{A} + \mathbf{B}) = \det(\mathbf{A})\det(\mathbf{I} + \mathbf{A}^{-\frac{1}{2}}\mathbf{B}\mathbf{A}^{-\frac{1}{2}})$. By spectral decomposition, $\mathbf{A}^{-\frac{1}{2}}\mathbf{B}\mathbf{A}^{-\frac{1}{2}} = \mathbf{Q}\mathbf{D}\mathbf{Q}^\top$ where $\mathbf{Q}\mathbf{Q}^\top = \mathbf{I}$ and $\mathbf{D}$ is diagonal with all entries being non-negative. Then, $\det(\mathbf{I} + \mathbf{A}^{-\frac{1}{2}}\mathbf{B}\mathbf{A}^{-\frac{1}{2}}) = \det(\mathbf{I} + \mathbf{D}) \geq 1$, and thus $\det(\mathbf{A} + \mathbf{B}) \geq \det(\mathbf{A})$.

For the claim on the equality, suppose $\det(\mathbf{I} + \mathbf{A}^{-\frac{1}{2}}\mathbf{B}\mathbf{A}^{-\frac{1}{2}}) = 1$, it leads to $\mathbf{D} = \mathbf{0}$, and thus $\mathbf{B} = \mathbf{0}$. □

**Lemma 2.** *Given a positive definite matrix $\mathbf{M} \in S_{++}$ that is partitioned as*

$$\mathbf{M} = \begin{pmatrix} \mathbf{A}_{m \times m} & \mathbf{B}_{m \times k} \\ \mathbf{B}_{k \times m}^\top & \mathbf{C}_{k \times k} \end{pmatrix},$$

*there is $\det(\mathbf{M}) = \det(\mathbf{A} - \mathbf{B}\mathbf{C}^{-1}\mathbf{B}^\top)\det(\mathbf{C})$.*

*Proof.*

$$\mathbf{Q} \begin{pmatrix} \mathbf{A}_{m \times m} & \mathbf{B}_{m \times k} \\ \mathbf{B}_{k \times m}^\top & \mathbf{C}_{k \times k} \end{pmatrix} \mathbf{Q}^\top = \begin{pmatrix} \mathbf{A} - \mathbf{B} \mathbf{C}^{-1} \mathbf{B}^\top & \mathbf{0} \\ \mathbf{0} & \mathbf{C} \end{pmatrix} \text{ where } \mathbf{Q} = \begin{pmatrix} \mathbf{I}_{m \times m} & -\mathbf{B}\mathbf{C}^{-1} \\ \mathbf{0}_{k \times m} & \mathbf{I}_{k \times k} \end{pmatrix}.$$

□

**Lemma 3.** *Let* $\mathbf{M}$ *be the matrix given in Lemma 2, then*

$$\det(\mathbf{M}) \le \det(\mathbf{A}) \det(\mathbf{C}).$$

*Moreover, the equality holds if and only if* $\mathbf{B} = \mathbf{0}$.

*Proof.* By Lemmas 2 and 1, $\det(\mathbf{M}) = \det(\mathbf{A} - \mathbf{B}\mathbf{C}^{-1}\mathbf{B}^\top) \det(\mathbf{C}) \le \det(\mathbf{A}) \det(\mathbf{C})$. Notice that the proof for Lemma 2 ensures that $\mathbf{A} - \mathbf{B}\mathbf{C}^{-1}\mathbf{B}^\top$ is positive definite.

By Lemma 1, the equality holds if and only if $\mathbf{B}\mathbf{C}^{-1}\mathbf{B}^\top = \mathbf{0}$. Note that for a positive definite matrix $\mathbf{C}^{-1}$, $\mathbf{x}^\top \mathbf{C}^{-1} \mathbf{x} = 0$ if and only if $\mathbf{x} = \mathbf{0}$. Therefore, $\mathbf{B}\mathbf{C}^{-1}\mathbf{B}^\top = \mathbf{0}$ if and only if $\mathbf{B} = \mathbf{0}$. □

**Notation** For a square matrix $\mathbf{A} \in \mathbb{R}^{m \times m}$, and $\alpha, \beta \subseteq \{1, 2, \ldots, m\}$, $\mathbf{A}[\alpha, \beta]$ is defined to be the sub-matrix of $\mathbf{A}$ by only keeping the rows and columns indexed by $\alpha$ and $\beta$, respectively. Plus, $\mathbf{A}[\alpha]$ is abused for $\mathbf{A}[\alpha, \alpha]$.

**Lemma 4.** *Suppose a positive definite matrix* $\mathbf{M} \in \mathbb{R}^{m \times m}$ *is given, for every* $\alpha \subseteq \{1, 2, \ldots, m\}$, *there is*

$$\det(\mathbf{M}^{-1}[\alpha^c]) = \frac{\det(\mathbf{M}[\alpha])}{\det(\mathbf{M})}.$$

*The convention is* $\det(\mathbf{M}[\emptyset]) = 1$.

*Proof.* Without loss of generality, we write $\mathbf{M} = \begin{pmatrix} \mathbf{A} & \mathbf{B} \\ \mathbf{B}^\top & \mathbf{C} \end{pmatrix}$ such that $\mathbf{M}[\alpha] = \mathbf{C}$ and $\mathbf{M}[\alpha^c] = \mathbf{A}$.

By Lemma 2, $\det(\mathbf{M}) = \det(\mathbf{S}) \det(\mathbf{C})$ where $\mathbf{S} = \mathbf{A} - \mathbf{B}\mathbf{C}^{-1}\mathbf{B}^\top$. On the other hand,

$$\mathbf{M}^{-1} = \begin{pmatrix} \mathbf{S}^{-1} & -\mathbf{S}^{-1}\mathbf{B}\mathbf{C}^{-1} \\ -\mathbf{C}^{-1}\mathbf{B}^\top\mathbf{S}^{-1} & \mathbf{C}^{-1} + \mathbf{C}^{-1}\mathbf{B}^\top\mathbf{S}^{-1}\mathbf{B}\mathbf{C}^{-1}. \end{pmatrix}$$

Therefore, we have

$$\frac{\det(\mathbf{M}[\alpha])}{\det(\mathbf{M})} = \frac{\det(\mathbf{C})}{\det(\mathbf{M})} = \det(\mathbf{S}^{-1}) = \det(\mathbf{M}^{-1}[\alpha^c]).$$

□

**Lemma 5.** *For every positive definite matrix* $\mathbf{M}_{m \times m} \in \mathcal{S}_{++}$, *and every* $\alpha, \beta \subseteq \{1, 2, \ldots m\}$,

$$\det(\mathbf{M}[\alpha \cup \beta]) \det(\mathbf{M}[\alpha \cap \beta]) \le \det(\mathbf{M}[\alpha]) \det(\mathbf{M}[\beta])$$

*The convention here is* $\det(\mathbf{M}[\emptyset]) = 1$. *Moreover, provided that* $(\alpha \cup \beta) \backslash \alpha \ne \emptyset$ *and* $(\alpha \cup \beta) \backslash \beta \ne \emptyset$, *and w.l.o.g., write*

$$\mathbf{M} = \begin{pmatrix} \mathbf{A} & \mathbf{B} & \mathbf{C} \\ \mathbf{B}^\top & \mathbf{D} & \mathbf{E} \\ \mathbf{C}^\top & \mathbf{E}^\top & \mathbf{F} \end{pmatrix} \tag{38}$$

*such that* $\mathbf{M}[\alpha] = \begin{pmatrix} \mathbf{A} & \mathbf{B} \\ \mathbf{B}^\top & \mathbf{D} \end{pmatrix}$ *and* $\mathbf{M}[\beta] = \begin{pmatrix} \mathbf{D} & \mathbf{E} \\ \mathbf{E}^\top & \mathbf{F} \end{pmatrix}$, *A necessary condition to have the equality is that each column of* $\mathbf{C}$ *lies in* $\mathrm{span}(\mathbf{B})$.

*Proof.* Our proof is adapted from (Horn and Johnson 2012, Theorem 7.8.9), by which we add a necessary condition for the equality.

By Lemmas 4 and 3, there is

$$\frac{\det(\mathbf{M}[\alpha \cap \beta])}{\det(\mathbf{M}[\alpha \cup \beta])} = \det(\mathbf{M}^{-1}[((\alpha \cup \beta)\backslash\alpha) \cup ((\alpha \cup \beta)\backslash\beta)])$$

$$\leq \det(\mathbf{M}^{-1}[(\alpha \cup \beta)\backslash\alpha]) \det(\mathbf{M}^{-1}[(\alpha \cup \beta)\backslash\beta])$$

$$= \frac{\det(\mathbf{M}[\alpha])}{\det(\mathbf{M}[\alpha \cup \beta])} \cdot \frac{\det(\mathbf{M}[\beta])}{\det(\mathbf{M}[\alpha \cup \beta])}.$$

Suppose $(\alpha \cup \beta)\backslash\alpha \neq \emptyset$ and $(\alpha \cup \beta)\backslash\beta \neq \emptyset$, by Lemma 3, the equality holds if and only if $\mathbf{M}^{-1}[(\alpha \cup \beta)\backslash\beta, (\alpha \cup \beta)\backslash\alpha] = \mathbf{0}$. Using Eq. (38), one can verify that

$$\mathbf{M}^{-1}[(\alpha \cup \beta)\backslash\beta, (\alpha \cup \beta)\backslash\alpha] = (\mathbf{A}^{-1}\mathbf{B}\mathbf{G}^{-1}\mathbf{H} - \mathbf{A}^{-1}\mathbf{C})(\mathbf{J} - \mathbf{H}^\top\mathbf{G}^{-1}\mathbf{H})^{-1}$$

$$\text{where } \mathbf{G} = \mathbf{D} - \mathbf{B}^\top\mathbf{A}^{-1}\mathbf{B},$$

$$\mathbf{H} = \mathbf{E} - \mathbf{B}^\top\mathbf{A}^{-1}\mathbf{C},$$

$$\mathbf{J} = \mathbf{F} - \mathbf{C}^\top\mathbf{A}^{-1}\mathbf{C},$$

and thus the equality holds if and only if $\mathbf{C} = \mathbf{B}\mathbf{G}^{-1}\mathbf{H}$. $\qquad \square$

**Lemma 6.** *Given a continuous non-negative function $g : [0, 1] \to \mathbb{R}_+$ that achieves its minimum $0$ only at one point $w^* \in [0, 1]$, $\delta_{w^*}$ is the unique probability measure on $[0, 1]$ that minimizes*

$$\underset{P \in \mathcal{P}}{\operatorname{argmin}}\, L(P) := \int_0^1 g(t)\mathrm{d}P(t).$$

*Proof.* It is clear that $L(\delta_{w^*}) = 0 = \inf_{P \in \mathcal{P}} L(P)$. Suffice it to prove that $L(P) = 0$ implies $P = \delta_{w^*}$.

Suppose $P \in \mathcal{P}$ is given such that $L(P) = 0$. A point $w \in [0, 1]$ is said to be irrelevant w.r.t. $P$ if there exists $\epsilon > 0$, $P((w - \epsilon, w + \epsilon) \cap [0, 1]) = 0$. For the sake of contradiction, suppose a distinct point $w \in [0, 1]$ ($w \neq w^*$) is relevant. Since $g(w) > g(w^*) = 0$ and $g$ is continuous, there exists $\epsilon > 0$ such that $g(s) > 0$ for all $s \in [w - \epsilon, w + \epsilon] \cap [0, 1]$. Since $m = \inf_{s \in [w-\epsilon, w+\epsilon] \cap [0,1]} g(s) > 0$ and $w$ is relevant, we have

$$L(P) \geq \int_{[w-\epsilon, w+\epsilon] \cap [0,1]} m\mathrm{d}P(t) = m \cdot P([w - \epsilon, w + \epsilon] \cap [0, 1]) > 0, \text{'}$$

a contradiction. So the only possible relevant point is $w^*$.

Next, we prove that $P = \delta_{w^*}$. For each $\delta > 0$, construct $S_\delta = [0, w^* - \delta] \cup [w^* + \delta, 1]$. For each $w \in S_\delta$, since $w$ is irrelevant, there exists $\epsilon(w) > 0$ such that $P(O_w) = 0$ where $O_w = (w - \epsilon(w), w + \epsilon(w)) \cap [0, 1]$. Then, $\{O_w : w \in S_\delta\}$ forms an open (w.r.t. $[0, 1]$) cover of $S_\delta$. Since $S_\delta$ is compact, there is a finite subcover $S_\delta \subseteq \bigcup_{1 \leq i \leq k} O_{w_i}$. Therefore, $P(S_\delta) \leq \sum_{i=1}^k P(O_{w_i}) = 0$, which implies $P(S_\delta) = 0$. By letting $\delta \to 0$, $P(S_\delta) \to P([0, 1]\backslash\{w^*\}) = 0$. Using the $\pi - \lambda$ theorem, e.g., (Durrett 2019, Theorem A.1.4), one can get $P = \delta_{w^*}$. $\qquad \square$

**Lemma 7.** *Suppose $\mathbf{V}_{[n],P} = \mathbf{M}_P^{[n]}\mathbf{C}^{[n]}(\mathbf{M}_P^{[n]})^\top$ where $\mathbf{M}_P^{[n]}$ is defined in Proposition 7 and $\mathbf{C}^{[n]} = \boldsymbol{\Sigma} \otimes \boldsymbol{\Sigma} \otimes \cdots \otimes \boldsymbol{\Sigma}$ (n repetitions), and assume $\sigma_{12} < \min(\sigma_{11}, \sigma_{22})$. Then, the Dirac delta distribution $\delta_{w^*}$ where $w^* = \dfrac{\sigma_{11} - \sigma_{12}}{\sigma_{11} + \sigma_{22} - 2\sigma_{12}}$ makes the matrix $\mathbf{V}_{[n],P}$ diagonal. In addition, $\delta_{w^*}$ is the unique probability distribution on $[0, 1]$ that makes $\mathbf{V}_{[n],P}$ diagonal if $n > 1$.*

*Proof.* Since $(\mathbf{A} \otimes \mathbf{B})(\mathbf{C} \otimes \mathbf{D}) = \mathbf{AC} \otimes \mathbf{BD}$, according to Proposition 7, for each $w \in [0, 1]$,

$$\mathbf{V}_{[n],\delta_w} = \mathbf{B}^{[n]}(w)\mathbf{C}^{[n]}(\mathbf{B}^{[n]}(w))^\top = \overbrace{\mathbf{V}_{[1],\delta_w} \otimes \mathbf{V}_{[1],\delta_w} \otimes \cdots \otimes \mathbf{V}_{[1],\delta_w}}^{n \text{ repetitions}}$$

$$\text{where } \mathbf{V}_{[1],\delta_w} = \begin{pmatrix} p_1(w) & p_2(w) \\ p_2(w) & C \end{pmatrix},$$

$$p_1(w) = \sigma_{11}(1 - w)^2 + \sigma_{22}w^2 - 2\sigma_{12}w(1 - w),$$

$$p_2(w) = (\sigma_{11} + \sigma_{22} - 2\sigma_{12})w + \sigma_{12} - \sigma_{11},$$

$$C = \sigma_{11} + \sigma_{22} - 2\sigma_{12}.$$

Using Eq. (13) (note $2^{[n]}$ is ordered w.r.t. the binary ordering), by induction, one can show the $(A, B)$-th entry of $\mathbf{V}_{[n],\delta_w}$ is $p_1(w)^{|[n]-(A\cup B)|}p_2(w)^{|(A\backslash B)\cup(B\backslash A)|}C^{|A\cap B|}$ for every $A, B \subseteq [n]$. Therefore, for every off-diagonal entry of $\mathbf{V}_{[n],\delta_w}$, it must contain the term $p_2(w)^k$ where $k$ is some positive integer. Notice that $p_2(w^*) = 0$, and thus $\mathbf{V}_{[n],\delta_{w^*}}$ is diagonal.

Conversely, suppose that a probability distribution $P$ on $[0, 1]$ is given such that $\mathbf{V}_{[n],P}$ is diagonal. Then,

$$\mathbf{V}_{[n],P} = \left(\int_0^1 \mathbf{B}^{[n]}(t)\mathrm{d}P(t)\right)\mathbf{C}^{[n]}\left(\int_0^1 \mathbf{B}^{[n]}(t)\mathrm{d}P(t)\right)^\top.$$

Observe that $\mathbf{V}_{[n],P}$ equals to $\int_0^1 \mathbf{B}^{[n]}(t)\mathbf{C}^{[n]}(\mathbf{B}^{[n]}(t))^\top\mathrm{d}P(t)$ along the last column as we can prove by induction that the last row of $\mathbf{B}_n(t)$ contains only constants. Since $\mathbf{B}^{[n]}(t)\mathbf{C}^{[n]}(\mathbf{B}^{[n]}(t))^\top$ can be computed recursively, one can show by induction that $\mathbf{V}_{[n],P}(\emptyset, [n]) = \int_0^1 p_2(t)^n\mathrm{d}P(t)$ and $\mathbf{V}_{n,P}(\{1\}, [n]) = \int_0^1 Cp_2(t)^{n-1}\mathrm{d}P(t)$.

Set $g(w) = p_2(w)^n$ when $n$ is even and $g(w) = Cp_2(w)^{n-1}$ otherwise. Note that $g(w) \geq 0$ and attains its minimum 0 only at the point $w^*$. By Lemma 6, it implies $P = \delta_{w^*}$. $\qquad\square$

## D    Experiment Settings

The datasets we use in the main paper are summarized in Table 4. The logistic model used is

$$\underset{\mathbf{w},b}{\mathrm{argmin}}\,\frac{1}{n}\sum_{(\mathbf{x},y)\in\mathcal{D}_{tr}}\mathrm{CrossEntropy}(\mathbf{w}\cdot\mathbf{x}+b,y)$$

In the experiment of noisy label detection, the F1-score is computed by using

$$\frac{2\cdot|\{z\in\mathcal{D}_{tr}\mid z\text{ is flipped and }z\text{ is detected noisy}\}|}{|\{z\in\mathcal{D}_{tr}\mid z\text{ is flipped}\}|+|\{z\in\mathcal{D}_{tr}\mid z\text{ is detected noisy}\}|},$$

which is the harmonic mean of precision and recall.

### D.1    Fitting Kronecker Noises

We independently sample 128 sequence $\emptyset = S_0 \subsetneq S_1 \subsetneq \cdots \subsetneq S_n = [n]$ in a way that $|S_{i+1}\backslash S_i| = 1$ for every $0 \leq i < n$. For each subset $S_i$ ($0 \leq i \leq n$) in a sampled sequence, $v(S_i)$ is computed using 128 random seeds. All these utility evaluations are then employed to generate an estimated covariance matrix $\hat{D} \in \mathbb{R}^{(n+1)\times(n+1)}$, which is an estimate for a sub-matrix of the ground-truth covariance matrix of the stochasticity. Note that under Assumption 1, for every $0 \leq i, j \leq n$,

$$\mathrm{Cov}(v_n(S_i), v_n(S_j)) = \sigma_{11}^{n-\max(i,j)}\cdot\sigma_{12}^{\max(i,j)-\min(i,j)}\cdot\sigma_{22}^{\min(i,j)}.$$

### D.2    Sampling Lift

Basically, sampling lift, which is designed for all semi-values, refers to any estimators based on

$$\phi_i(v; P) = \mathbb{E}_S[v_n(S\cup i) - v_n(S)]\text{ for every }v_n\in\mathcal{G}\text{ and }i\in[n]$$

where $S$ is sampled from $[n]\backslash i$ with $P(S) = p_{s+1}^n$.

There are two different ways to sample $S$. One is derived from Eq. (2). For example, to estimate the value for $n \in [n]$, the procedure is i) sample $t \in [0, 1]$ according to the underlying probability distribution $P \in \mathcal{P}$; ii) let $X$ be a Bernoulli random variable such that $P(X = 1) = t$, and sample a vector $\mathbf{b} \in \mathbb{R}^{n-1}$ whose entries are independently following $X$; and iii) define $S \subseteq [n]\backslash n$ by letting $i \in S$ if and only if $b_i = 1$. The other way is by noticing that

$$\phi_i(v; P) = \sum_{k=0}^{n-1}q_{k+1}\sum_{\substack{S\subseteq[n]\backslash i\\|S|=k}}\binom{n-1}{k}^{-1}(v(S\cup i) - v(S)) = \mathbb{E}_k\mathbb{E}_{S:|S|=k}[v(S\cup i) - v(S)]].$$

Table 4: A summary of datasets used in this work.

| Dataset | Source |
|---|---|
| MNIST (LeCun et al. 1998) | PyTorch |
| FMNIST (Xiao et al. 2017) | PyTorch |
| 2dplanes | `https://www.openml.org/d/727` |
| bank-marketing (Moro et al. 2011) | `https://www.openml.org/d/1461` |
| bioresponse | `https://www.openml.org/d/4134` |
| covertype | `https://www.openml.org/d/150` |
| cpu | `https://www.openml.org/d/761` |
| credit | `https://www.openml.org/d/31` |
| default (Yeh and Lien 2009) | `https://www.openml.org/d/42477` |
| diabetes | `https://www.openml.org/d/37` |
| fraud (Dal Pozzolo et al. 2014) | `https://www.openml.org/d/42175` |
| gas (Vergara et al. 2012) | `https://www.openml.org/d/1476` |
| har (Anguita et al. 2013) | `https://www.openml.org/d/1478` |
| iris | `https://www.openml.org/d/61` |
| letter (Frey and Slate 1991) | `https://www.openml.org/d/6` |
| optdigits | `https://www.openml.org/d/28` |
| pendigits | `https://www.openml.org/d/32` |
| phoneme | `https://www.openml.org/d/1489` |
| pol | `https://www.openml.org/d/722` |
| satimage | `https://www.openml.org/d/182` |
| segment | `https://www.openml.org/d/36` |
| spambase | `https://www.openml.org/d/44` |
| texture | `https://www.openml.org/d/40499` |
| wind | `https://www.openml.org/d/847` |

where $q_k = p_k^n \cdot \binom{n-1}{k-1}$ for every $1 \le k \le n$. The corresponding sampling strategy is i) sample $k \sim \{q_k\}_{1 \le k \le n}$, and then ii) sample $S$ uniformly from $2^{[n] \setminus i}$ subject to $|S| = k - 1$.

We comment that for the first way we do not have to calculate the distribution $\{q_k\}_{1 \le k \le n}$. Note the implementation for $\{q_k\}_{1 \le k \le n}$ might suffer from numerical blowup induced by $\binom{n-1}{k-1}$, and it is why Wang and Jia (2023) did not report the results of the Beta Shapley values if $n > 500$.

## E    More Experiment Results

In this section, we provide detailed figures, see Figures 4 to 10, to supplement the results reported in Tables 3 and 2. In each figure, $lr$ stands for the learning rate used for that dataset. In addition, we also run experiments while setting $|\mathcal{D}_{tr}| = 200$, 10 percent of which is flipped. The corresponding results are presented in Figures 11 to 17.

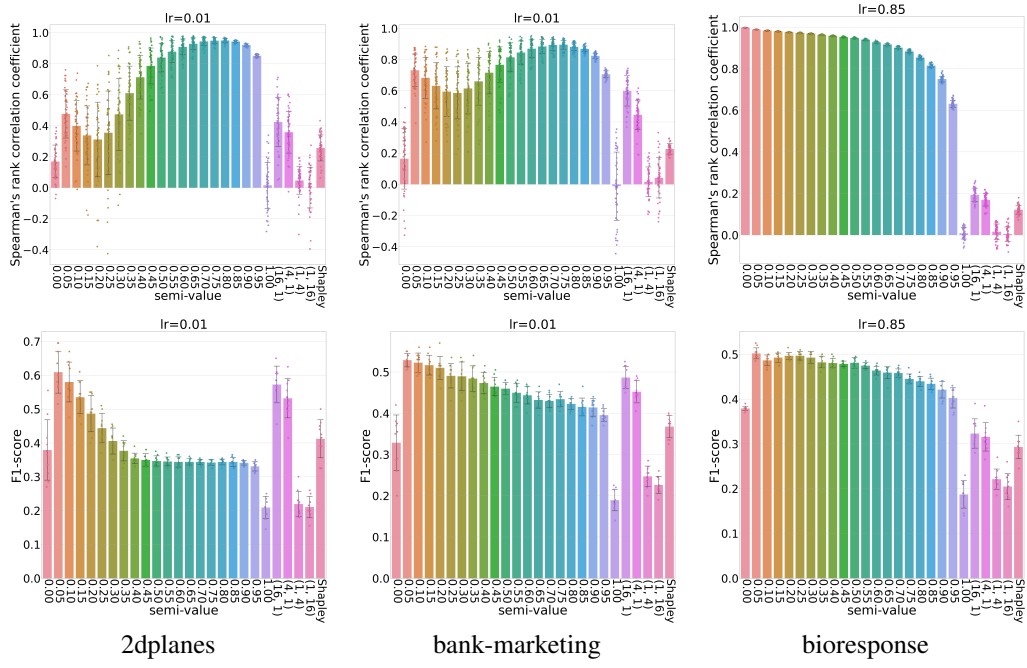

Figure 4: The first row exhibits results from the experiment of ranking consistency, while the second row is for the experiment of noisy label detection. Each result is reported using mean and standard deviation. For noisy label detection, the F1-score is reported over 10 random seeds. For ranking consistency, the Spearman's rank correlation coefficient is reported over all possible pairs of 10 random seeds, which is 45 in total.

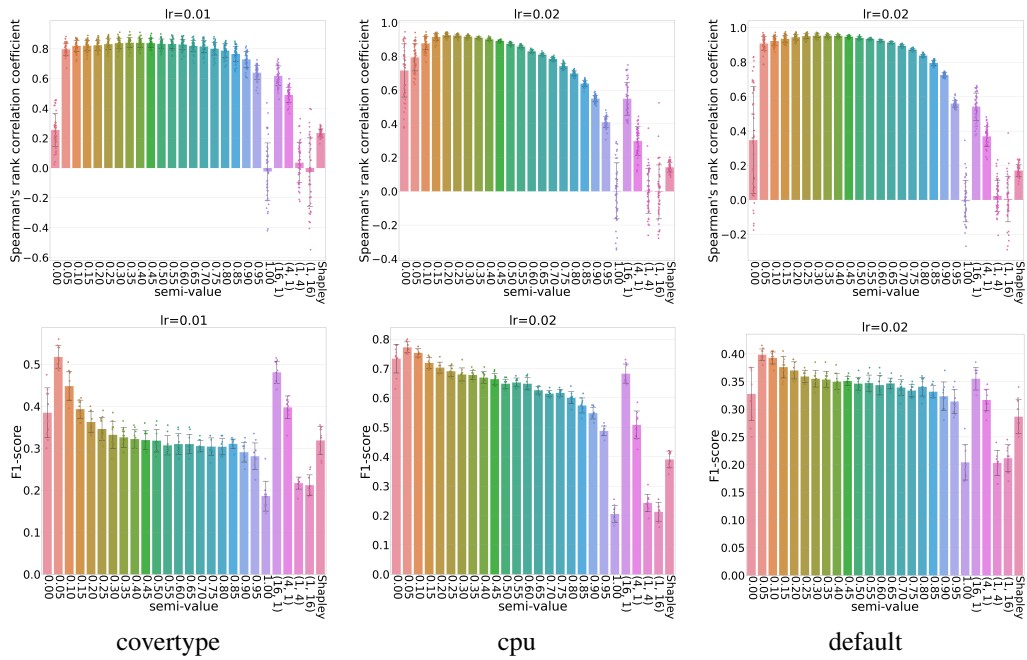

Figure 5: The first row exhibits results from the experiment of ranking consistency, while the second row is for the experiment of noisy label detection. Each result is reported using mean and standard deviation. For noisy label detection, the F1-score is reported over 10 random seeds. For ranking consistency, the Spearman's rank correlation coefficient is reported over all possible pairs of 10 random seeds, which is 45 in total.

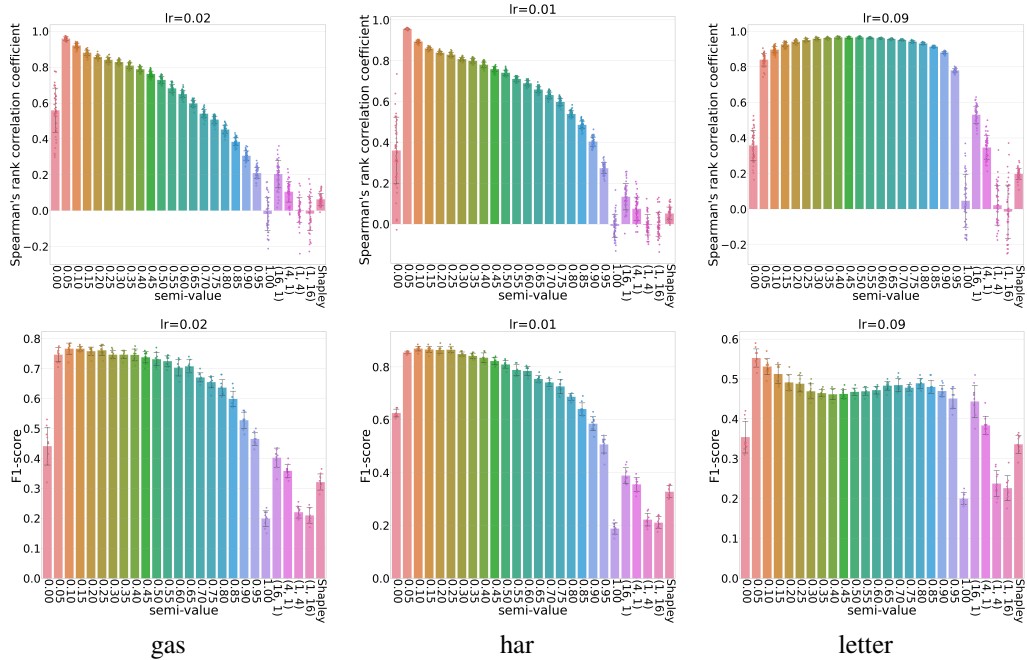

Figure 6: The first row exhibits results from the experiment of ranking consistency, while the second row is for the experiment of noisy label detection. Each result is reported using mean and standard deviation. For noisy label detection, the F1-score is reported over 10 random seeds. For ranking consistency, the Spearman's rank correlation coefficient is reported over all possible pairs of 10 random seeds, which is 45 in total.

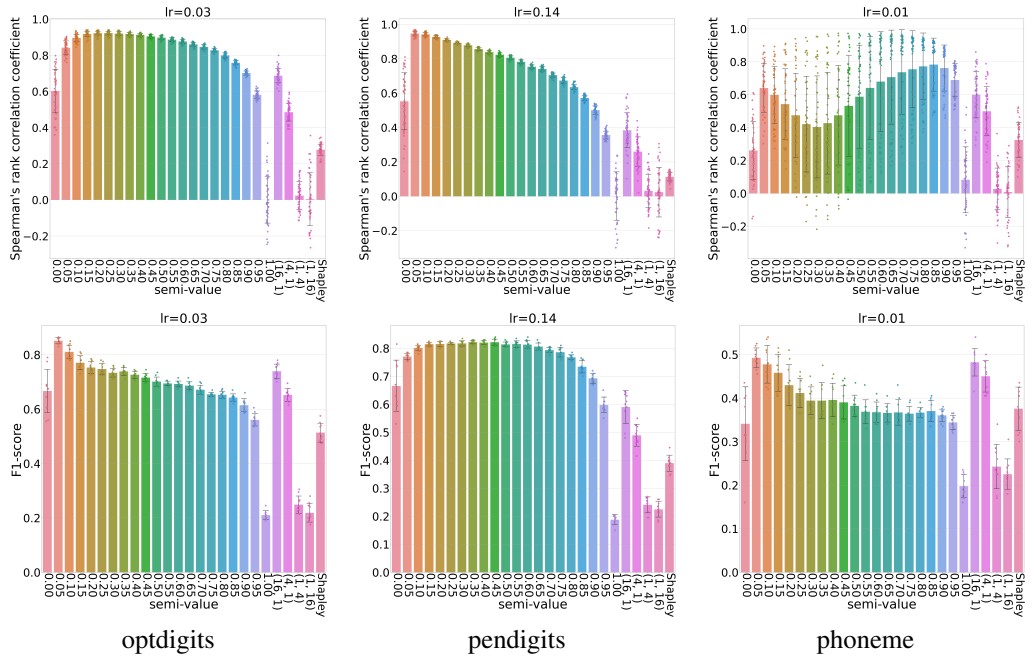

Figure 7: The first row exhibits results from the experiment of ranking consistency, while the second row is for the experiment of noisy label detection. Each result is reported using mean and standard deviation. For noisy label detection, the F1-score is reported over 10 random seeds. For ranking consistency, the Spearman's rank correlation coefficient is reported over all possible pairs of 10 random seeds, which is 45 in total.

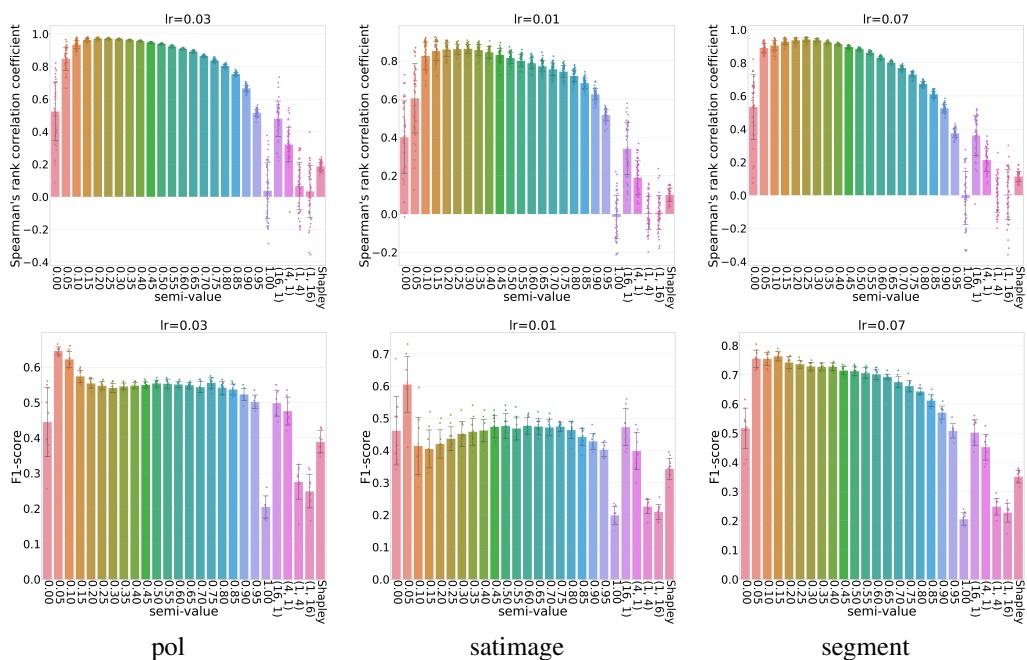

Figure 8: The first row exhibits results from the experiment of ranking consistency, while the second row is for the experiment of noisy label detection. Each result is reported using mean and standard deviation. For noisy label detection, the F1-score is reported over 10 random seeds. For ranking consistency, the Spearman's rank correlation coefficient is reported over all possible pairs of 10 random seeds, which is 45 in total.

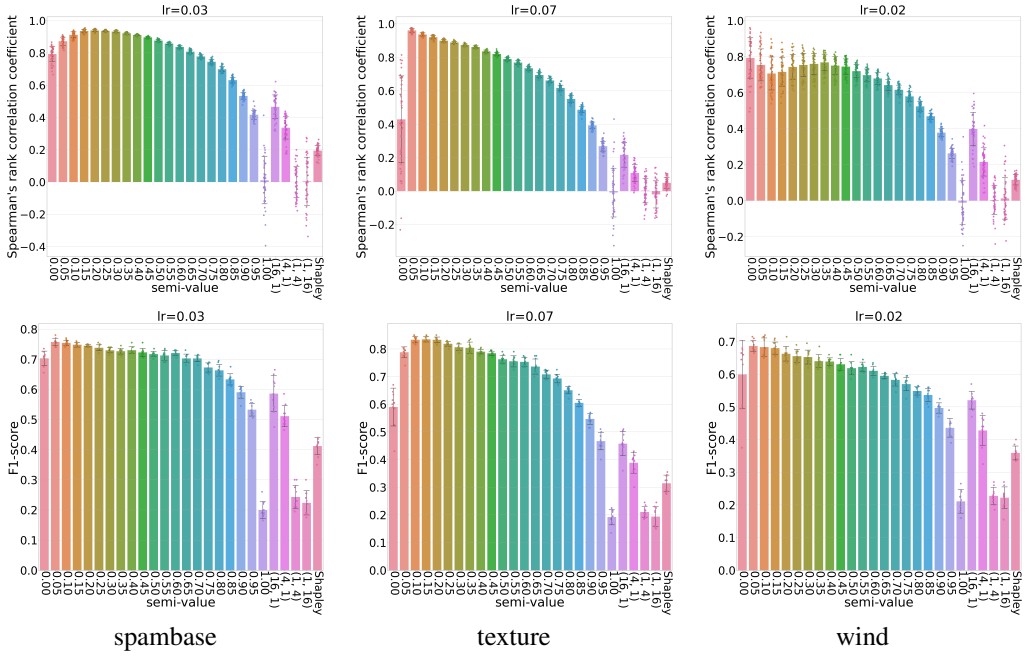

Figure 9: The first row exhibits results from the experiment of ranking consistency, while the second row is for the experiment of noisy label detection. Each result is reported using mean and standard deviation. For noisy label detection, the F1-score is reported over 10 random seeds. For ranking consistency, the Spearman's rank correlation coefficient is reported over all possible pairs of 10 random seeds, which is 45 in total.

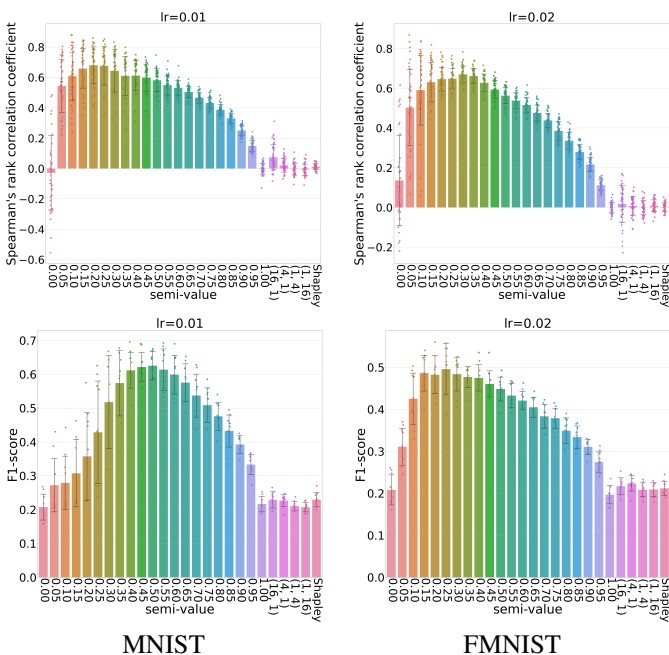

Figure 10: The first row exhibits results from the experiment of ranking consistency, while the second row is for the experiment of noisy label detection. Each result is reported using mean and standard deviation. For noisy label detection, the F1-score is reported over 10 random seeds. For ranking consistency, the Spearman's rank correlation coefficient is reported over all possible pairs of 10 random seeds, which is 45 in total.

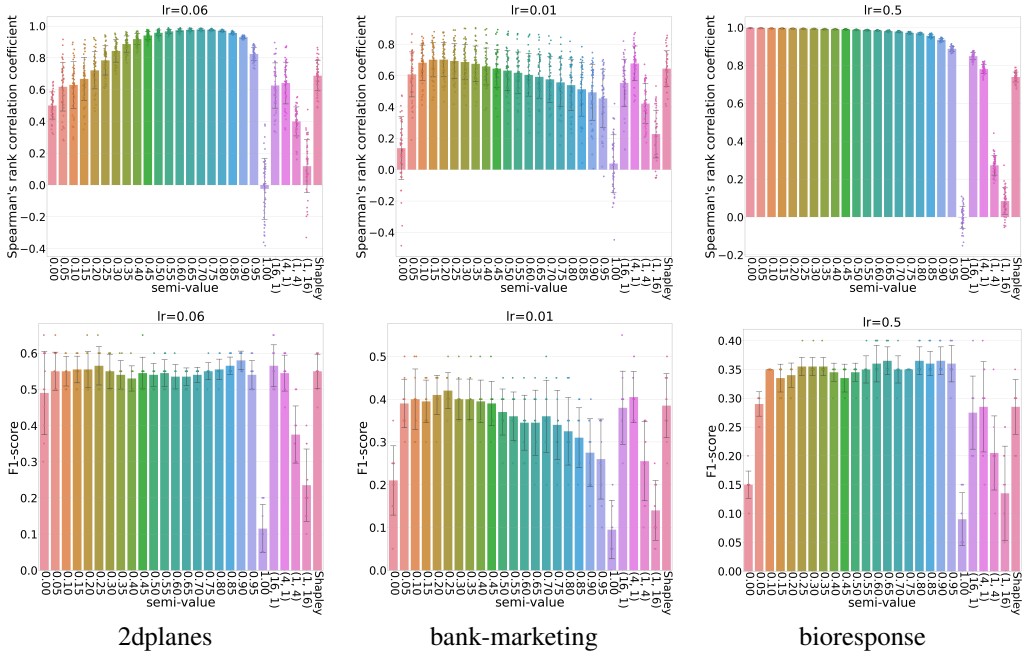

Figure 11: The first row exhibits results from the experiment of ranking consistency, while the second row is for the experiment of noisy label detection. Each result is reported using mean and standard deviation. For noisy label detection, the F1-score is reported over 10 random seeds. For ranking consistency, the Spearman's rank correlation coefficient is reported over all possible pairs of 10 random seeds, which is 45 in total.

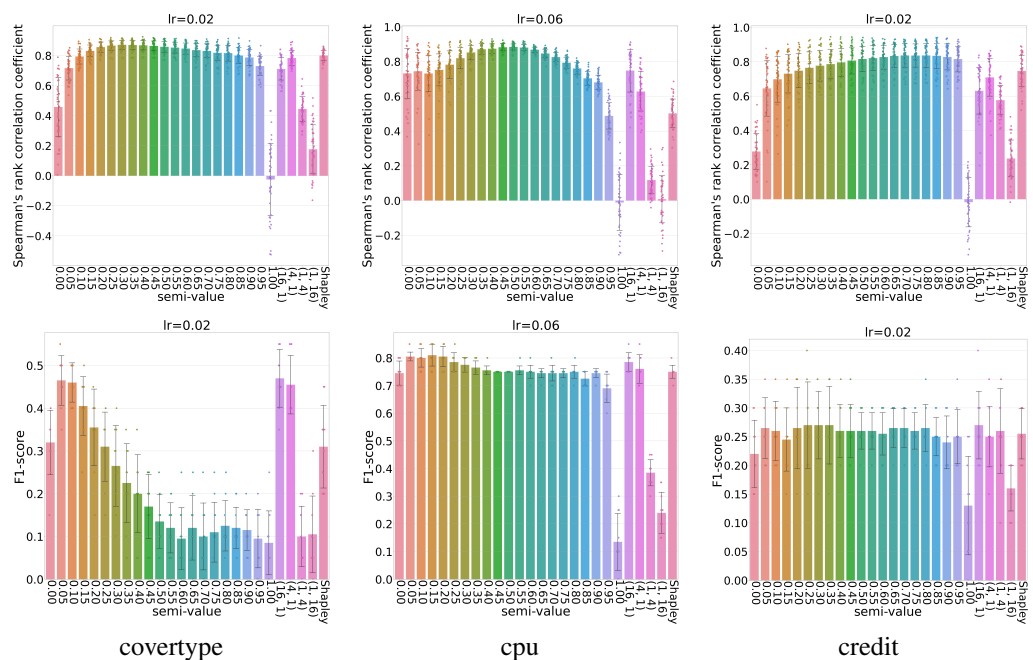

Figure 12: The first row exhibits results from the experiment of ranking consistency, while the second row is for the experiment of noisy label detection. Each result is reported using mean and standard deviation. For noisy label detection, the F1-score is reported over 10 random seeds. For ranking consistency, the Spearman's rank correlation coefficient is reported over all possible pairs of 10 random seeds, which is 45 in total.

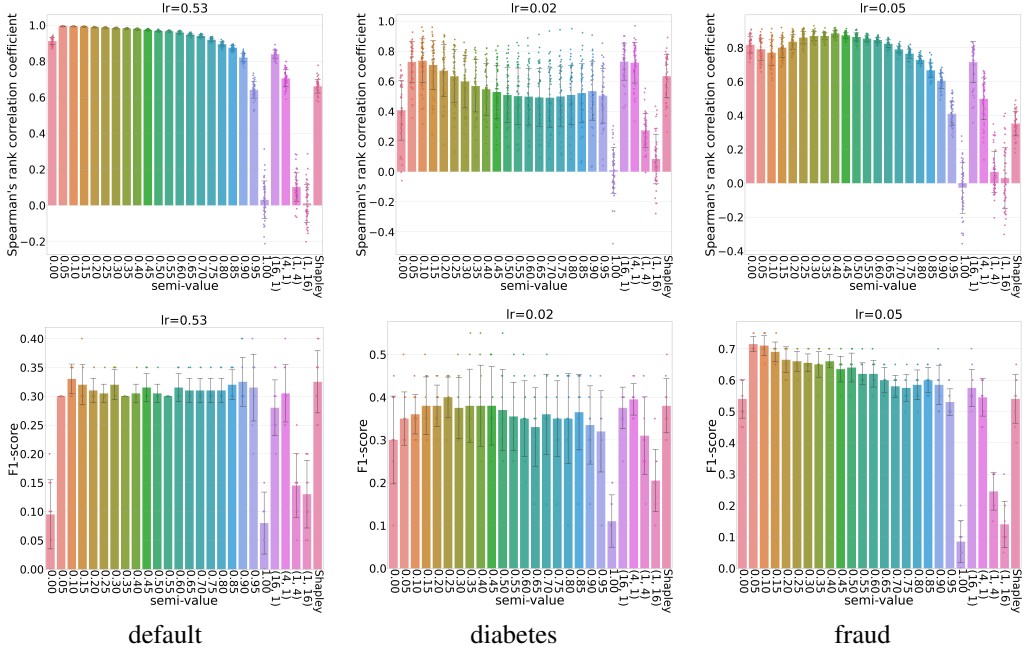

Figure 13: The first row exhibits results from the experiment of ranking consistency, while the second row is for the experiment of noisy label detection. Each result is reported using mean and standard deviation. For noisy label detection, the F1-score is reported over 10 random seeds. For ranking consistency, the Spearman's rank correlation coefficient is reported over all possible pairs of 10 random seeds, which is 45 in total.

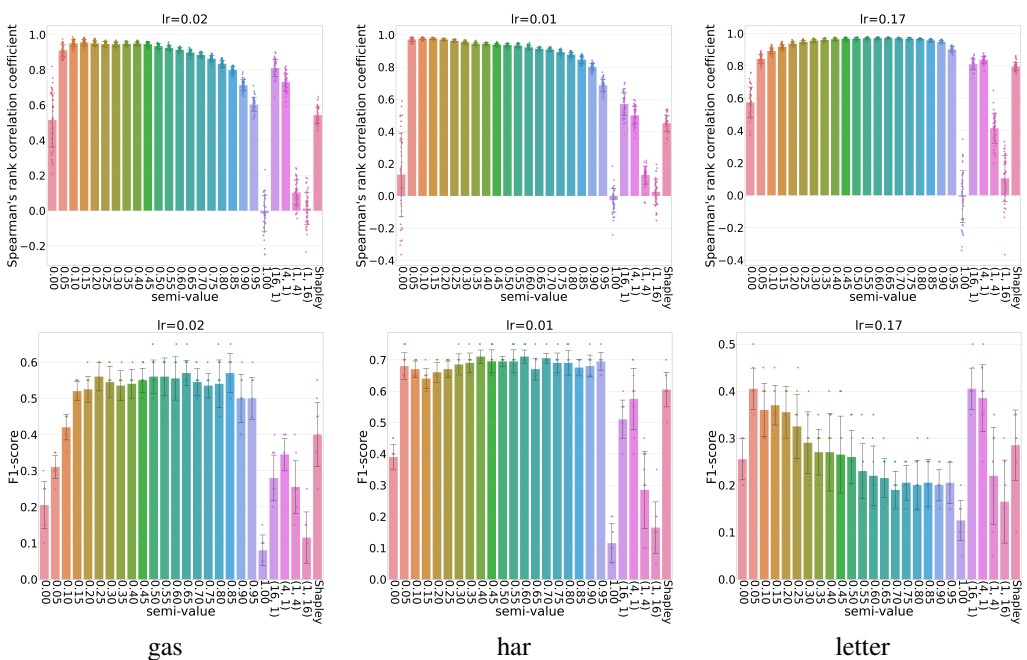

Figure 14: The first row exhibits results from the experiment of ranking consistency, while the second row is for the experiment of noisy label detection. Each result is reported using mean and standard deviation. For noisy label detection, the F1-score is reported over 10 random seeds. For ranking consistency, the Spearman's rank correlation coefficient is reported over all possible pairs of 10 random seeds, which is 45 in total.

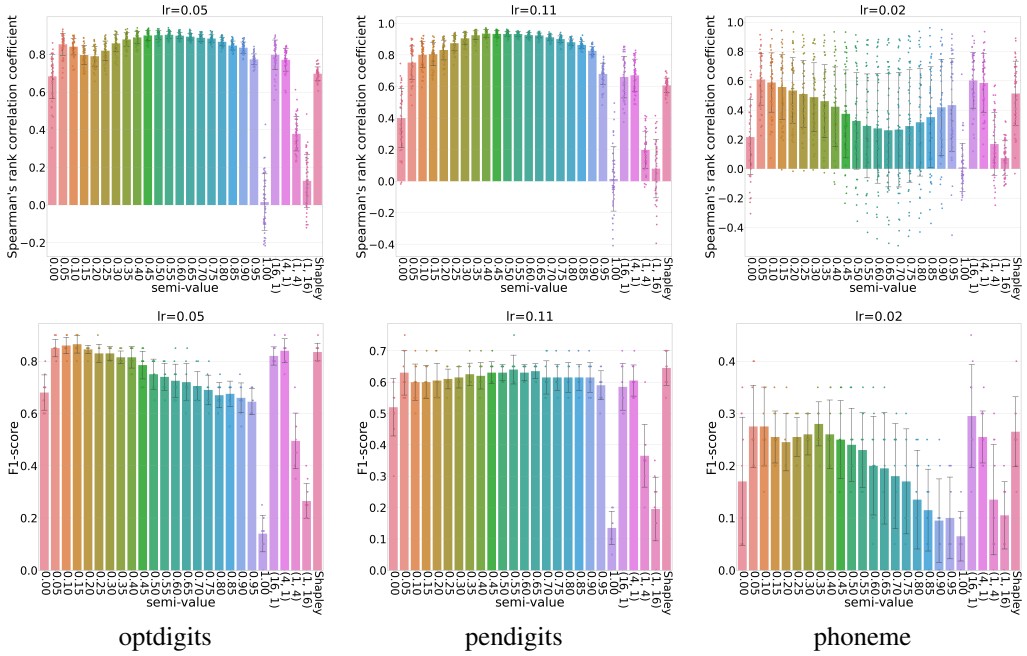

Figure 15: The first row exhibits results from the experiment of ranking consistency, while the second row is for the experiment of noisy label detection. Each result is reported using mean and standard deviation. For noisy label detection, the F1-score is reported over 10 random seeds. For ranking consistency, the Spearman's rank correlation coefficient is reported over all possible pairs of 10 random seeds, which is 45 in total.

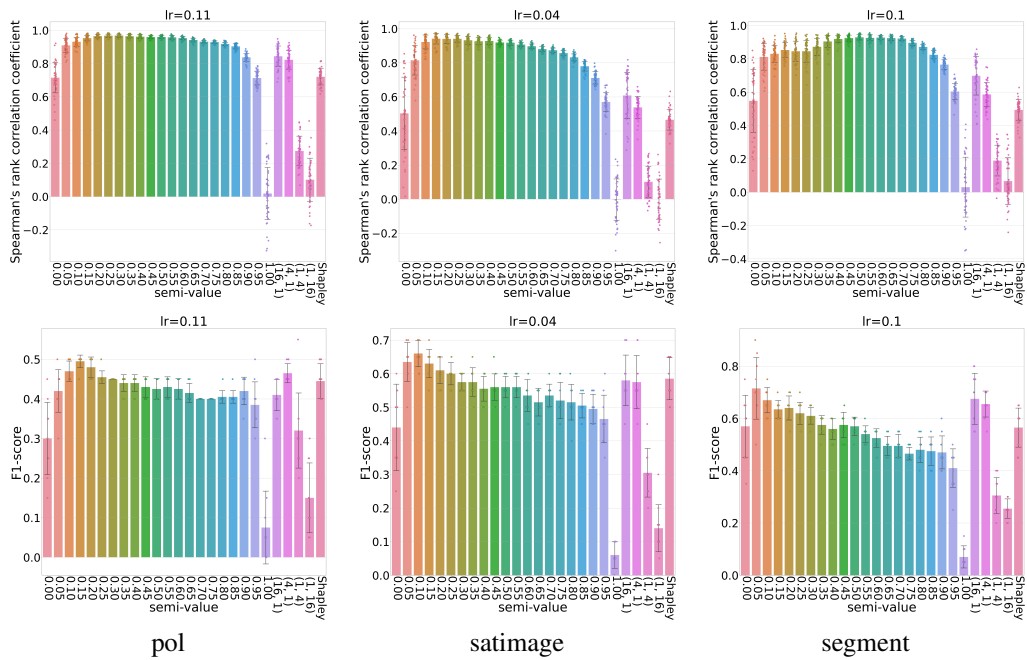

Figure 16: The first row exhibits results from the experiment of ranking consistency, while the second row is for the experiment of noisy label detection. Each result is reported using mean and standard deviation. For noisy label detection, the F1-score is reported over 10 random seeds. For ranking consistency, the Spearman's rank correlation coefficient is reported over all possible pairs of 10 random seeds, which is 45 in total.

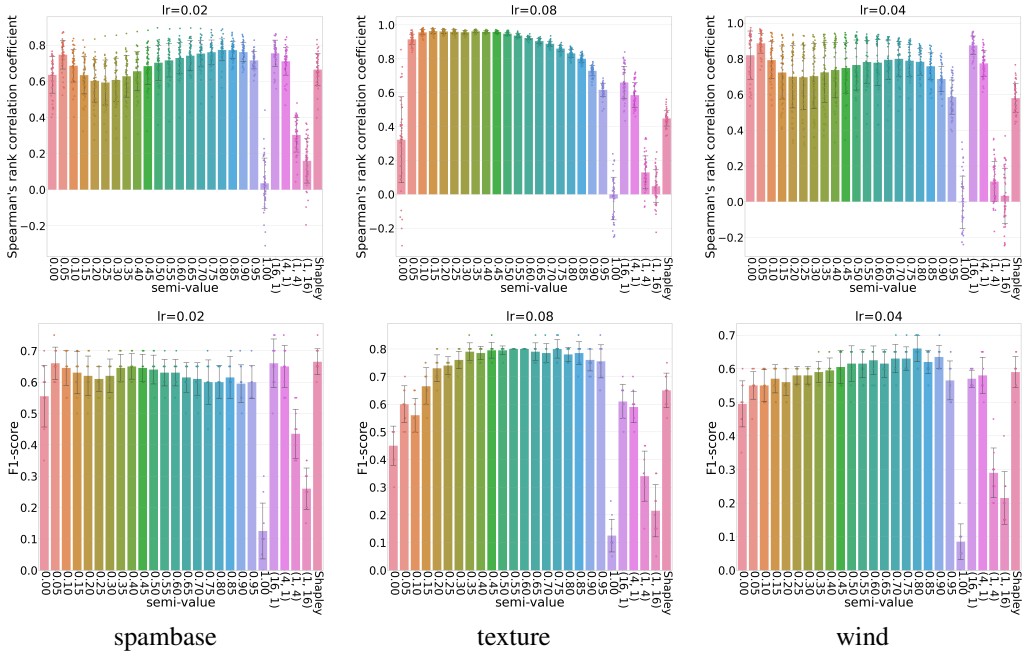

Figure 17: The first row exhibits results from the experiment of ranking consistency, while the second row is for the experiment of noisy label detection. Each result is reported using mean and standard deviation. For noisy label detection, the F1-score is reported over 10 random seeds. For ranking consistency, the Spearman's rank correlation coefficient is reported over all possible pairs of 10 random seeds, which is 45 in total.

# Extra References

Anguita, D., A. Ghio, L. Oneto, X. Parra, J. L. Reyes-Ortiz, et al. (2013). "A Public Domain Dataset for Human Activity Recognition Using Smartphones". In: *Esann*. Vol. 3, p. 3.

Dal Pozzolo, A., O. Caelen, Y.-A. Le Borgne, S. Waterschoot, and G. Bontempi (2014). "Learned Lessons in Credit Card Fraud Detection From a Practitioner Perspective". *Expert Systems with Applications*, vol. 41, no. 10, pp. 4915–4928.

Denneberg, D. and M. Grabisch (1999). "Interaction Transform of Set Functions Over a Finite Set". *Information Sciences*, vol. 121, no. 1-2, pp. 149–170.

Durrett, R. (2019). "Probability: Theory and Examples". Vol. 49. Cambridge University Press.

Frey, P. W. and D. J. Slate (1991). "Letter Recognition Using Holland-Style Adaptive Classifiers". *Machine Learning*, vol. 6, pp. 161–182.

Fujimoto, K., I. Kojadinovic, and J.-L. Marichal (2006). "Axiomatic Characterizations of Probabilistic and Cardinal-Probabilistic Interaction Indices". *Games and Economic Behavior*, vol. 55, no. 1, pp. 72–99.

Grabisch, M., J.-L. Marichal, and M. Roubens (2000). "Equivalent Representations of Set Functions". *Mathematics of Operations Research*, vol. 25, no. 2, pp. 157–178.

Grabisch, M. and M. Roubens (1999). "An Axiomatic Approach to the Concept of Interaction Among Players in Cooperative Games". *International Journal of Game Theory*, vol. 28, no. 4, pp. 547–565.

Horn, R. A. and C. R. Johnson (2012). "Matrix Analysis". Cambridge University Press.

Moro, S., R. Laureano, and P. Cortez (2011). "Using Data Mining for Bank Direct Marketing: An Application of the Crisp-Dm Methodology".

Sundararajan, M., K. Dhamdhere, and A. Agarwal (2020). "The Shapley Taylor Interaction Index". In: *International Conference on Machine Learning*. PMLR, pp. 9259–9268.

Vergara, A., S. Vembu, T. Ayhan, M. A. Ryan, M. L. Homer, and R. Huerta (2012). "Chemical Gas Sensor Drift Compensation Using Classifier Ensembles". *Sensors and Actuators B: Chemical*, vol. 166, pp. 320–329.

Xiao, H., K. Rasul, and R. Vollgraf (2017). "Fashion-Mnist: A Novel Image Dataset for Benchmarking Machine Learning Algorithms". *arXiv preprint arXiv:1708.07747*.

Yeh, I.-C. and C.-h. Lien (2009). "The Comparisons of Data Mining Techniques for the Predictive Accuracy of Probability of Default of Credit Card Clients". *Expert Systems with Applications*, vol. 36, no. 2, pp. 2473–2480.

Zhang, H., Y. Xie, L. Zheng, D. Zhang, and Q. Zhang (2021). "Interpreting Multivariate Shapley Interactions in DNNs". In: *Proceedings of the AAAI Conference on Artificial Intelligence*. Vol. 35. 12, pp. 10877–10886.

