# OpenReview forum: "Robust Data Valuation with Weighted Banzhaf Values"
_NeurIPS.cc/2023/Conference — NeurIPS 2023 poster_

### Official Review · Reviewer_WZtW · 2023-07-03

**Soundness:** 3 good
**Presentation:** 2 fair
**Contribution:** 3 good
**Rating:** 6
**Confidence:** 4

**Summary:**

The paper proposes that 1) the semi-values obtained from fitting Kronecker noise to the data robustly rank data more consistently across runs and 2) we can efficiently estimate these weighted Banzhaf values with the maximum sample reuse principle. The extensive experiments illustrated the advantages of the contributions: higher sample efficiency and performance in noisy label detection and rank consistency.

**Strengths:**

1. It is novel to fit noise to the data to determine the semivalue weights and important to expand beyond and compare against common data valuation methods (Shapley and Banzhaf)
2. Extensive experiments on many datasets were performed.

**Weaknesses:**

The definition of robustness (and especially Sec 3.2), assumptions and limitations of the work are not sufficiently and clearly explained. More details are given in the questions and suggestions below. Incorporating the suggestions will improve the contextualization relative to prior work and make the claims more convincing.

**Questions:**

Q1. Is the definition of robustness in this paper the same as Data Banzhaf?

Q2. Does the estimation according to Line 134/equation 1 get significantly less inaccurate as $w$ tends to 0 or 1? One of the terms will have much fewer samples.

Q3. Does Kronecker Noise mean Gaussian noise with Kronecker covariance?
What is an intuitive interpretation of each entry in $X_i$ and $\Sigma$? How is $X_i$ related to $v$? Provide further explanation/citations on the suitability of Kronecker noise (less parameters to fit?) and why $\sigma_{22} < \sigma_{11}$.

My guess is that $X_i \sim \mathcal{N}(0, \Sigma =\begin{pmatrix} \sigma_{11}\  \sigma_{12}\\\\ \sigma_{21}\   \sigma_{22} \end{pmatrix})$ . Instead of observing the utility function $\bar{u} = \begin{pmatrix} u(\emptyset) \\\\ u(i) \end{pmatrix}$ directly, $\bar{u} + X_i$ is observed.

In equation 5, what is $\phi_P$ and why is its outer product $\Sigma$?

Q4. Can you explain why Theorem 1/Remark 1 contradict Data Banzhaf? Is it due to a different notion of robustness / assumption of noise structure? Data Banzhaf approach is noise-structure agnostic as they consider users may have difficulties estimating the noise.

Q5. Line 229-231 mentions that exact values are intractable on larger datasets and approximation would “draw the real noises away from following Kronecker noises”. What does the quoted part mean? Is this a problem with using Kronecker noise that limits the applicability in real use cases?

**Suggestions**
* The robustness concept should be better introduced in the introduction’s 3rd paragraph. Later, a mathematical/formal definition of the robustness concept should be included.
For example, the Data Banzhaf paper explains that the stochasticity in the rankings/utilities is due to stochastic training methods. Such inconsistencies may interfere with identifying the usefulness of data. The Data Banzhaf paper gives an in-depth definition of Robustness in Sec 4. These details are missing from this paper.

* The “weighted Banzhaf” and “consistency” concepts introduced here have alternative names in CGT literature: binomial semivalues and null player exclusion/hereditary property. See Domenech, M., Giménez, J. M., & Puente, M. A. (2016). Some properties for probabilistic and multinomial (probabilistic) values on cooperative games. Optimization, 65(7), 1377-1395.

* In the experiments, Shapley and the Beta Shapley values are underperforming as they place large weights on the smaller/larger coalitions. I would expect Beta$(k, k)$ for large $k$ or Beta$(kp^*, k(1-p^*))$ where p* s the robust Banzhaf version to perform as well.
Thus, an alternative conclusion (instead of using binomial semivalues) is that the semivalue weights should be learnt/adaptive.

* It might be possible to empirically verify that in real dataset/SGD training, training on more data leads to more or less noise in the utilities.




Minor suggestions (not affecting score)
* coeffient misspelling in Figure 1 caption
* the axis tick labels in all figures are too small
* in equation 1, s is not defined to be the size of S

**Limitations:**

The limitation mentioned is that there is no universally best and most robust value for all datasets. However, other limitations (e.g. the Kronecker noise assumption or complexity) should be discussed.

---

> ### Author Rebuttal · Authors · 2023-08-09
>
> Thank you for pointing out our insufficient presentation for the context as well as your helpful suggestions. The suggested reference is useful, and will be added in our revision. Please view our global response for clarification, and below we address other comments, and we will polish our paper accordingly.
>
> **Q: Does the estimation according to Line 134/equation 1 get significantly less inaccurate as $w$ tends to 0 or 1?**
>
> A: No, it is always accurate in terms of almost-sure convergence. Precisely, let $\hat{\phi}^m$ (omitting the argument $v$ as the below statement holds for every utility function $v$) be the estimate for $\phi$ using $m$ samples from the proposed sampling scheme. Specifically, $\hat{\phi}^1$ can be viewed as a random vector with its underlying distribution induced by the sampling scheme, and proposition 2 proves that $\mathbb{E}[\hat{\phi}^1]=\phi$. Therefore, by the law of large numbers, $\hat{\phi}^m\rightarrow\phi$ almost surely as $m\rightarrow\infty$. Heuristically, when $w$ tends to $0$ or $1$, the weights $p^{n-1}_s$ in Eq. (1) for those terms that receive fewer examples decreases to $0$, which means the error induced by ignoring them tends to be negligible. We will polish this part in the final version.
>
> **Q: Why $\sigma_{11}>\sigma_{22}$?**
>
> A: We are not sure if we understand this question correctly. Basically, We mention $\sigma_{11}>\sigma_{22}$ in Line 168 just to discuss a possible scenario, as supported by proposition 3, where the Kronecker noise allows for modeling decreasing variance of $v(S)$ as $|S|$ increases. We do not impose this condition in any of our results or experiments. Empirically, provided that $v$ is defined by prediction accuracy, such a decreasing variance was mostly observed (it might lift a little bit for some $|S|$ and then keep decreasing) in our experiments.
>
> **Q: What is $ \phi_{P} $ and why is its outer product?**
>
> A: Let $\phi_P(v,i)$ denotes the $i$-th entry of $\phi_P(v)$,  substituting Eq (2) in Eq (1) yields
> $$
> \phi_P(v, i) = \sum_{S \subseteq [n]\backslash i} \left( \int_{0}^{1} t^{s}(1-t)^{n-1-s} \mathrm{d}t \right) \left( v(S\cup i) - v(S) \right) .
> $$
> The outer product in theorem 1 was a typo while we wanted to express $ \det(\mathrm{Cov}(\phi_{P}(v))) $ where $ \mathrm{Cov} $ stands for covariance matrix. We had remedied this typo in the supplementary where we proved theorem 1.
>
> **Q: What it means by Lines 229-231?**
>
> A: Thanks for pointing out that our description on this part is not that proper, and we will modify in the paper. The context is that the randomness of each noisy utility function $v$ is induced by stochastic training. Empirically, such randomness comes from varying the underlying random seed controlling the training phase. We model such randomness by assuming that $ \boldsymbol\epsilon = v - \mathbb{E}[v] $ follows a Kronecker noise, by which the most robust semi-value can be derived. In other words, the randomness of $ v $ in theorem 1 is restricted to be from stochastic training. On large datasets, for each fixed random seed, $ \phi_{P}(v) $ can only be approximated, which means there is another randomness onto $\phi_P(v)$ brought in by the approximation procedure. In this case, these two noises of different origins get entangled on $\phi_P(v)$ and that makes it complicated to analyze the randomness of $\phi_P(v)$.
> Therefore, to fairly examine our theory, we focus on small datasets where each semi-value can be computed exactly so that there is *only one* expected randomness that is induced from stochastic training.
>
> **Q:  I would expect Beta$(k,k)$ for large $k$ or Beta$(kp^\*,k(1-p^\*))$ where $p^\*$ is the robust Banzhaf version to perform as well.**
>
> A: We experiment on all $(\alpha,\beta)\in[16]\times[16]$ (256 combinations in total) for Beta Shapley, and report the best for Beta Shapley on as many datasets as time permits. All results are reported with 10% of data in $\mathcal{D}_{tr}$ flipped, and the other settings remain the same. Note that when they achieve the same performance in noisy label detection, weighted Banzhaf values have the smallest variance across different runs (i.e., different random seeds used for training).
> #### Ranking consistency
> Dataset|Weighted Banzhaf|Beta Shapley
> ---|---|---
> 2dplanes|0.50:**0.868**$\pm$0.035|(8,8):0.851$\pm$0.038
> bank-marketing|0.35:**0.924**$\pm$0.030|(1,1):0.918$\pm$0.025
> bioresponse|0.10:**0.969**$\pm$0.008|(10,2):0.944$\pm$0.019
> covertype|0.50:**0.811**$\pm$0.082|(1,1):0.806$\pm$0.074
> cpu|0.00:0.874$\pm$0.065|(16,1):**0.883**$\pm$0.078
> default|0.05:**0.986**$\pm$0.002|(16,1):0.978$\pm$0.004
> gas|0.05:**0.952**$\pm$0.008|(16,2):0.886$\pm$0.023
> letter|0.35:**0.711**$\pm$0.058|(5,3):0.700$\pm$0.051
> fraud|0.00:0.910$\pm$0.019|(16,1):**0.923**$\pm$0.011
> pol|0.10:**0.949**$\pm$0.013|(16,3):0.945$\pm$0.010
> #### Noisy label detection
> Dataset|Weighted Banzhaf|Beta Shapley
> ---|---|---
> 2dplanes|0.10:0.517$\pm$0.047|(13,3):**0.542**$\pm$0.098
> bank-marketing|0.20:**0.325**$\pm$**0.025**|(4,1):**0.325**$\pm$0.056
> bioresponse|0.55:**0.358**$\pm$**0.034**|(3,4):**0.358**$\pm$0.053
> covertype|0.05:**0.483**$\pm$0.055|(13,1):0.442$\pm$0.034
> cpu|0.05:**0.667**$\pm$**0.047**|(10,1):**0.667**$\pm$0.055
> default|0.10:**0.342**$\pm$**0.019**|(7,2):**0.342**$\pm$0.034
> gas|0.15:**0.550**$\pm$0.065|(2,1):0.525$\pm$0.038
> letter|0.20:0.383$\pm$0.047|(13,5):**0.392**$\pm$0.053
> fraud|0.05:**0.792**$\pm$0.019|(9,2):0.783$\pm$0.055
> pol|0.10:**0.558**$\pm$0.034|(12,1):0.525$\pm$0.069
>
> **Q: It might be possible to empirically verify that in real dataset/SGD training, training on more data leads to more or less noise in the utilities.**
>
> A: Thanks for this suggest, we will add this type of experiment in the revision. Nevertheless, all our results are reported with standard deviation across different random seeds determining the initialization of trainable models as well as the order of feeding data, which demonstrates the randomness contained in $\phi_P(v)$.

---

> > ### Comment · Reviewer_WZtW · 2023-08-15
> >
> > I thank the authors for the detailed response, helpful clarifications and follow-up experiments. I acknowledge that I have read the global and individual responses.
> >
> > The additional discussion has improved my opinion of my work though like the other reviewers, I am still concerned about the justification/choice of the Kronecker noise model and robustness. I will wait to the discussion period to re-evaluate my score.
> >
> > In the global response, the authors mention "$X_i(1)$ represents the randomness brought in by the absence of datum $i$". Why is there additional randomness in this case?

---

> > > ### Author Response · Authors · 2023-08-17
> > >
> > > Thank you for your reply! We apologize for the confusion and let us clarify our explanation of the Kronecker noise.
> > >
> > > The utility function $v$ maps each subset $A$ of players/data to a number. We want a *succinct* noise model that will perturb $v(A)$ and $v(B)$ in a correlated manner, depending on how much they overlap (see Proposition 3). The straightforward approach is to let $\tilde v = v + \epsilon$ where $\mathrm{Cov}(\epsilon) = \Lambda$. The downside is that the matrix $\Lambda$ is of size $2^n \times 2^n$, too large to fit. Instead, we let $(\epsilon_i, \bar \epsilon_i)$ be independent copies sampled from a $2\times 2$ covariance matrix $\Sigma$, and then we perturb $\tilde v(A) = v(A) + \prod_{i\in A} \epsilon_i \prod_{j\not\in A} \bar\epsilon_j$. This product form of noise is inspired by Owen's multilinear extension of utility functions, and only requires modeling a $2\times 2$ covariance matrix. Intuitively, it can be memorized as multiplying $\epsilon_i$ if $i\in A$ and multiplying $\bar\epsilon_i$ if $i\not\in A$. This is just a convenient form of prescribing correlated noise; it does not mean a player/datum still incurs noise when not present in a coalition. We also note that if we use instead $\tilde v(A) = v(A) + \prod_{i\in A} \epsilon_i$, then the noise perturbation will be biased against larger sets $A$ (i.e., more terms in the product).
> > >
> > > Our theory (including the Kronecker noise) is mainly proposed to explain *the adaptive phenomenon that no single value dominates in all experimental settings* (see Tables 2 and 3). Our experiments were performed with real, agnostic noises, i.e., *they do not necessarily follow any assumption or theory.* Besides, we validated the criteria of differential entropy using empirical results summarized in Figure 3, which confirmed that Eq. (6) leads to the most consistent semivalue in empirical rankings. Intuitively, a random vector tends to be deterministic (which means the resulting random ranking is also deterministic) as its differential entropy decreases to 0.

---

### Official Review · Reviewer_CJa5 · 2023-07-07

**Soundness:** 3 good
**Presentation:** 4 excellent
**Contribution:** 2 fair
**Rating:** 6
**Confidence:** 4

**Summary:**


This paper extends the notion of Banzhaf values to weighted Banzhaf values to improve the robustness of data ranking. The authors show under Kronecker noises, when minimizing the worse-case entropy, the most robust parameters belong to the family of weighted Banzhaf values. Similarly, as implemented in Data Banzhaf, authors use a maximum sample reuse strategy to improve sampling efficiency, as shown by juxtaposing it with other sampling techniques. The authors demonstrate the robustness of weighted Banzhaf values by comparing them with other data valuation methods over multiple datasets. Weighted Banzhaf values produce consistent data ranking. Further, the performance is presented through noisy label detection experiments.

**Strengths:**

+ The authors show the exact case when Data Banzhaf achieves highest robustness and cases when weighted Banzhaf values can achieve better robustness.

+ The weighted Banzhaf values are successfully generalized from Data Banzhaf and demonstrates largest safe margin among compared data valuation methods.



**Weaknesses:**

+ All utility functions are set to be accuracy of simple models. Specifically, only simple models are considered, such as logistic regression models or LeNet.

+ Only simple tabular and image datasets are considered.

+ The sample size is also minimal, 2000 samples at most.

It seems that even though efficient sampling is employed, the current data valuation method cannot be applied in more practical settings, larger models, datasets, and sample sizes, which is due to the intractable computation of model re-trainings.

**Questions:**

+ How can you ensure exact Shapley value, if each evaluation is noisy?

+ Since weighted Banzhaf results are based on the best Banzhaf weights, can you also provide results for the best parameters for Beta Shapley?

+ What is the time complexity and actual runtime for weighted Banzhaf value at each step?

+ It would be insightful to compare weighted Banzhaf values with KNN Shapley, which does not require model training. Is the data ranking, in that case, inherently robust to noise?


+ Typo Line 242: descent
+ Typo Line 266: flipped

---

> ### Author Rebuttal · Authors · 2023-08-09
>
> Thanks for the detailed review and feedback! Please refer to our global response for clarification on the context. Below we address other specific comments.
>
> **Q: How can you ensure exact Shapley value, if each evaluation is noisy?**
>
> A: To clarify, each noisy utility function $v$ can be written as $v(\cdot,U)$ where $U$ represents the random seed determining the initialization of trainable models and the order of feeding data in the training phase. Our setting *only* considers such randomness introduced by varying $U$. On small datasets, the Shapley value of each $v(\cdot,U)$ is exactly computed using Eq (1).
>
> **Q: What is the time complexity and actual runtime for weighted Banzhaf value at each step?**
>
> A: To answer this question, we generalize the convergence results in (Wang and Jia, 2023) by assuming that the given utility function $v$ satisfies $ |v|\leq r $.
> Wang and Jia (2023) analyzed the setting $v\in[0,1]$ where $v$ was taken to be prediction performance.
> To describe the complexity, we need the concept of $(\epsilon,\delta)$-approximation for the considered semi-value $\phi$, which is $P[||\hat{\phi}-\phi||\_\infty\geq\epsilon]\leq\delta$. Take Banzhaf value as an example, Wang and Jia (Theorem 4.9, 2023) proved it requires $ \frac{32r^2}{\epsilon^2}\log(\frac{5n}{\delta})$ model evaluations to achieve an $(\epsilon,\delta)$-approximation based on maximum sample reuse principle.
> For $w$-weighted Banzhaf value, mimicking their analysis by replacing Eq (52) therein by $\tilde{\phi}=\frac{1}{mw}\sum_{S\in\mathcal{S}\_{\ni i}}v(S)-\frac{1}{m(1-w)}\sum_{S\in\mathcal{S}_{\not\ni i}}v(S)$, we can prove that it requires $ \frac{(2|w-0.5|+2)^2r^2}{2\epsilon^2w^2(1-w)^2}\log(\frac{5n}{\delta})$, or equivalently $O(\frac{1}{\epsilon^2}\log(\frac{n}{\delta}))$, model evaluations to achieve an $(\epsilon,\delta)$-approximation. On the other hand, theorem 4.8 therein says that (though they only asserted for Banzhaf value, it applies to all semi-values) for sampling lift it requires $ \frac{4nr^2}{\epsilon^2}\log(\frac{2n}{\delta})$, or equivalently $O(\frac{n}{\epsilon^2}\log(\frac{n}{\delta}))$, model evaluations to achieve an $(\epsilon,\delta)$-approximation. These two results somewhat demonstrate why maximum sample reuse principle is better than sampling lift. For each model evaluation $v(S)$, the corresponding complexity relies on how $v$ is designed. In our experiments, the running time for $v(S)$ is $\Theta(|S|)$ as we employed one-epoch learning with SGD. Therefore, for $w$-weighted Banzhaf value, our proposed approximation runs faster as $w$ gets smaller, because it samples more frequently small-size subsets. We will add this discussion in the supplementary.
>
> **Q: Compare weighted Banzhaf values with KNN Shapley.**
>
> A: For value-based data valuation methods, there are two key components: i) the design of utility functions and ii) which semi-value to aggregate marginal contributions.
> KNN Shapley is composed of i) using the performance of KNN as utility functions, which are *deterministic* as no training is required, and ii) use the Shapley value to do aggregation.
> In our context, each utility function $v$ measures the performance of models trained using SGD. Thus, $v$ is *stochastic* and can be represented by $v(\cdot,U)$ where $U$ is random seed determining the training procedure. The Kronecker noise is to model $\epsilon=v-E[v] $ where the randomness is from varying $U$.  To conclude, KNN does not lie in our scope as the produced $v$ is *deterministic.*
>
> **Q: The best results and parameters for Beta Shapley.**
>
> A: For Beta$(\alpha,\beta)$, the range of $\alpha$ or $\beta$ is $(0,\infty)$, and thus it is impossible to have an extensive search. The parameters we used are all the ones reported by the original paper. Nevertheless, we experimented over all $(\alpha,\beta)\in[16]\times[16]$, which leads to 256 combinations in total. By contrast, the best weighted Banzhaf value is selected among 21 candidates. We report on as many datasets as time permits. Specifically, all the results are reported with 10% of data in $\mathcal{D}_{tr}$ flipped, and the other settings remain the same. Note that when they achieve the same performance in noisy label detection, weighted Banzhaf values have the smallest variance across different runs (i.e., different random seeds used for training).
> #### Ranking consistency
> Dataset|Weighted Banzhaf|Beta Shapley
> ---|---|---
> 2dplanes|0.50:**0.868**$\pm$0.035|(8,8):0.851$\pm$0.038
> bank-marketing|0.35:**0.924**$\pm$0.030|(1,1):0.918$\pm$0.025
> bioresponse|0.10:**0.969**$\pm$0.008|(10,2):0.944$\pm$0.019
> covertype|0.50:**0.811**$\pm$0.082|(1,1):0.806$\pm$0.074
> cpu|0.00:0.874$\pm$0.065|(16,1):**0.883**$\pm$0.078
> default|0.05:**0.986**$\pm$0.002|(16,1):0.978$\pm$0.004
> gas|0.05:**0.952**$\pm$0.008|(16,2):0.886$\pm$0.023
> letter|0.35:**0.711**$\pm$0.058|(5,3):0.700$\pm$0.051
> fraud|0.00:0.910$\pm$0.019|(16,1):**0.923**$\pm$0.011
> pol|0.10:**0.949**$\pm$0.013|(16,3):0.945$\pm$0.010
> #### Noisy label detection
> Dataset|Weighted Banzhaf|Beta Shapley
> ---|---|---
> 2dplanes|0.10:0.517$\pm$0.047|(13,3):**0.542**$\pm$0.098
> bank-marketing|0.20:**0.325**$\pm$**0.025**|(4,1):**0.325**$\pm$0.056
> bioresponse|0.55:**0.358**$\pm$**0.034**|(3,4):**0.358**$\pm$0.053
> covertype|0.05:**0.483**$\pm$0.055|(13,1):0.442$\pm$0.034
> cpu|0.05:**0.667**$\pm$**0.047**|(10,1):**0.667**$\pm$0.055
> default|0.10:**0.342**$\pm$**0.019**|(7,2):**0.342**$\pm$0.034
> gas|0.15:**0.550**$\pm$0.065|(2,1):0.525$\pm$0.038
> letter|0.20:0.383$\pm$0.047|(13,5):**0.392**$\pm$0.053
> fraud|0.05:**0.792**$\pm$0.019|(9,2):0.783$\pm$0.055
> pol|0.10:**0.558**$\pm$0.034|(12,1):0.525$\pm$0.069

---

> > ### Comment · Reviewer_CJa5 · 2023-08-13
> >
> > I appreciate authors for their great responses.
> >
> > Similar to other reviewers, I also share concerns on the choice of Kronecker noise over other noises (a comparison with other noises would be helpful) and only choosing a specific kind of robustness as the key metrics.

---

> > > ### Author Response · Authors · 2023-08-17
> > >
> > > Thanks for your efforts! This response is to address your concerns.
> > >
> > > **Q: Comparison with other noises.**
> > >
> > > A: We are not sure if we understand this question correctly. Our experiments (Tables 2 and 3) were done with real noises, i.e., the underlying noises (that are due to stochastic training) did not necessarily follow any noise models.  As far as we know, Wang and Jia (2023) is the first to address the randomness from stochastic training, and our work further pushes the frontier of this direction. Wang and Jia (2023) adopted a noise-structure-agnostic notion (i.e., the safe margin) to conclude that the Banzhaf value is the most robust *in a universal sense*. However, as shown in Tables 2 and 3, there is often not a single semivalue that dominates for all experimental settings. In contrast, we propose the Kronecker noise to model the randomness, and our theory is an attempt to give a possible explanation to *the observed adaptive phenomenon.* We are not aware of other noise models analyzed in the data valuation literature and would appreciate any concrete suggestions.
> > >
> > > **Q: Concern on only choosing a specific kind of robustness as the key metrics.**
> > >
> > > A: We'd like to clarify that for our experiments we use Spearman's rank correlation coefficient and F1-score as performance metrics to evaluate different semivalues. The differential entropy criteria is employed for two purposes: (1) explain the adaptive phenomenon that no single semivalue dominates for all experimental settings; (2) determine which weighted Banzhaf value would achieve the most consistent ranking. We do not intend to claim our robustness criteria as the only useful one (e.g., the safe margin criteria of Wang and Jia (2023) is equally interesting), but rather as a mean to explain our experimental results.

---

> > > > ### Comment · Reviewer_CJa5 · 2023-08-21
> > > >
> > > > Thank you for your clarifications! I am happy to keep the positive score!

---

### Official Review · Reviewer_Zq9F · 2023-07-11

**Soundness:** 3 good
**Presentation:** 3 good
**Contribution:** 3 good
**Rating:** 6
**Confidence:** 4

**Summary:**

This work focuses on data valuation with weighted Banzhaf values, which seems to be effective, particularly in cases where the dataset or the data valuation process is noisy. Toward that, the authors introduce and utilize a Kronecker noise model to calculate robust values*  and moreover to do it in an efficient way utilizing the maximum sample reuse principle. They show that the weighted Banzhaf value with the Kronecker noise model is optimal in minimizing worst-case entropy. They performed experiments to show the efficiency of the method with maximum sample reuse, validate their theoretical finding on optimality (to a certain extend), and evaluate the performance of weighted Banzhaf in capturing noisy labels and data ranking. Their results indicate that weighted Banzhaf shows consistently good performance across different tasks and datasets.

*following the calculation of semi-values used for averaging contribution over subsets


**Strengths:**

**Clarity:** Overall, this paper is quite well written with clear expectations for the reader throughout. The logical flow of the paper is well-structured, although there were a few intuitive aspects that I found lacking. Nonetheless, I believe that the presentation of the work is solid.

**Originality:** The concept of the Kronecker noise model, and moreover utilization of it for learning semi-values and evaluating noisy datasets are original, as far as my knowledge extends.

**Quality:** I think this paper meets the quality standards starting from the introduction of the noise model and weighted version of Banzhaf indexing, to the argumentation of their optimality, and the extensive experimental evaluations.

**Significance:** I believe it is crucial to align game-theoretical data valuation approaches and compare their strengths and weaknesses across different scenarios, including noisy settings. As emphasized by the authors, no universal approach can really be deemed optimal for all datasets and noise settings. However, the flexibility offered by weighted Banzhaf (or data-driven? semi-values) is promising, opening up avenues for future research in understanding the optimality of data-valuation approaches across different regimes. I find the contribution of this work to be significant in this regard.

**Weaknesses:**

I believe there is room for improvement regarding the justification of the Kronecker noise model and its implications for data valuation. It would be helpful to provide further explanations and clarifications on the following points:
- Intuition behind the Kronecker model and the scenarios where it is (not) applicable.
- In addition to discussing the universality of an approach or lack thereof, I believe it is important to draw conclusions or insights from the comparisons. For example, for a reader like me, it is not immediately clear why the proposed method is challenged by other approaches in particular in Table 2, and how this performance comparison is affected by the intrinsic dynamics in the data (such as higher-order interactions, which is generally captured best by Shapley value).


**Questions:**

The points presented under the Weaknesses above can be considered as my questions here. I have the following additional questions:
- In Table 3, are some portions of labels still flipped? My understanding was that they are not. If it's indeed so, I am curious to see F1 scores in addition to the correlation coefficients. Clarification would be appreciated.
- How does the Kronecker noise model behave for large amounts of classes in data?
- Also, the reproducibility is checked as n/a but I think it applies here.
- Do you think overfitting may be an issue with this approach as semi-values are learned from data in the end?

**Limitations:**

As the authors also addressed in the supplementary material, the concept of the Kronecker noise model poses certain limitations where it does not align well with the actual data noise and is empirically justified only on small datasets. I have not observed an additional limitation beyond this.

---

> ### Author Rebuttal · Authors · 2023-08-09
>
> Thanks for your detailed review and feedback. Please refer to our global response for clarification on our context. We will release our code after acceptance to ease the replication of all our results. Below we address other specific comments.
>
> **Q: How does the Kronecker noise model behave for large amounts of classes in data?**
>
> A: Intuitively, more classes requires more training data to produce non-trivial trained models that can distinguish noisy and clean data well. We do not think the Kronecker noise would be a good fit for all possible induced noises on large datasets. Specifically,
> if we plot $\det(Cov(\phi_{\delta_w}(v)))$ (the correct objective in theorem 1) along $w$-axis (parameter for weighted banzhaf values), the curve is always a U-shape, which somewhat implies it is an upside-down U-shape (peaking at one position and gradually decreasing sideways) for the curve of correlation in ranking. But empirically it is not always this case even on a 2-class dataset, see for example the 1st-row-2nd-column plot (200 data from diabetes dataset) in Figure 6 in the supplementary. Despite this, empirical evidence still supports that weighted Banzhaf values are more likely to capture the most consistent one in ranking, as shown in Table 3.
>
> **Q: Is overfitting an issue with this approach as semi-values are learned from data in the end?**
>
> A: In our context, each utility function $v$ can be written by $v(\cdot,U)$ where $U$ represents random seed that controls the training phase. The considered randomness on $v$ is from varying $U$, and we model $\epsilon =v-E[v]$ using the proposed Kronecker noise.
> The empirical covariance matrix $\hat{D}$ to which a Kronecker noise is fitted is approximated by evaluating $v(\cdot,U)$ while varying $U$, see Line 675 in the supplementary for more details.
> *If $\hat{D}$ is approximated well*, we do not think overfitting would be an issue as perfect match indicates that the real noise exactly follows some Kronecker noise. Concretely, for the last two column in Figure 3, the KL divergences for the learned Kronecker noises are $0.5954$ and $0.5490$ for datasets 2dplanes and cpu, respectively.
>
> **Q: In Table 3, are some portions of labels still flipped? My understanding was that they are not. If it's indeed so, I am
> curious to see F1 scores in addition to the correlation coefficients.**
>
> A: We did not flip any labels for Table 3, but we can provide the correlation for all the flipped datasets used in Table 2, and then report the corresponding F1-scores. In the one-page pdf attached to our global response, Table A lists all the results of ranking consistency given that 10% of data in $\mathcal{D}_{tr}$ are flipped, and then Table B reports all the corresponding F1-scores in noisy label detection.
> These results show that ranking consistency does not correlate strictly with the performance of noisy label detection. A sign of this can be observed in Figure 1 as the peaking position is different in each column. Despite that, weighted Banzhaf values still perform well in noisy label detection, as shown in Table 2.

---

> > ### Comment · Reviewer_Zq9F · 2023-08-13
> > **Response to the authors**
> >
> > Dear Authors,
> >
> > Thank you for your clarifications. I acknowledge that I read your individual rebuttal as well as your global response, and have no further questions.
> >
> > Regards

---

### Official Review · Reviewer_SBfP · 2023-07-14

**Soundness:** 3 good
**Presentation:** 3 good
**Contribution:** 3 good
**Rating:** 6
**Confidence:** 3

**Summary:**

The paper looks at the standard data valuation problem, in case of noisy estimation of the value of a coalition. It proposes a model of noise, Kronecker noise, and shows that under this noise a weighted Banzhaf value (with weight that depends on parameters of the noise) is semi-value that maximizes a notion of robustness. If the weight is 0.5, it is equivalent to the Banzhaf value, but it is different otherwise. The paper shows a number of experiments that illustrate this result and the efficiency of computing the weighted Banzhaf value.

Note: There is a very related paper from Wang and Jia at AISTATS '23 that proposes the Banzhaf value and shows it is the most robust semi-value under a certain definition of robustness (and also proposed an efficient estimator based on the so-called maximum sample reused principle); so this submission's novelty and significance is obviously to be judged with respect to that AISTATS '23 paper.

**Strengths:**

The data valuation problem is a hot topic and moving away from the Shapley value adds interesting insights to the existing literature. In particular, considering robustness to noisy evaluation of the value function (as the paper does) is interesting.

In my opinion, the main contribution of the paper is the result of Theorem 1 that states the following: If the noise is Kronecker noise (as defined in Def 1), then the semi-value that minimizes the determinant of the semi-values covariance corresponds to a weighted Banzhaf value with weight given by (6), which depends on individual parameters of the noise.
- To me this results is more of the type of identifying a particular kind of noise and a particular measure of robustness such that weighted Banzhaf maximizes robustness (rather than having a naturally relevant notion of noise and then showing the result), but this is already quite interesting. The notion of Kronecker noise used is somewhat general and flexible.
- The proof is non-trivial (and very long). I was not able to follow all details. I wish the paper had included a sketch of proof to give meaningful intuition.
- I am wondering whether this result is really specific to the data valuation problem and even if data valuation is the most relevant application. Is there any other that one could think of?

The paper shows the application of this result to the data valuation problem based on extensive experiments on synthetic and real datasets (although the real data sets are still somewhat forced to fit the Kronecker noise framework, see below).

The paper also contains another result, Proposition 2, which shows that the maximum sample reuse principle can be used to efficiently estimate the weighted Banzhaf value, but this is very incremental compared to [Wang and Jia, AISTATS '23].

**Weaknesses:**

I did not find the comparison to [Wang and Jia, AISTATS '23] to be perfectly clear in the paper. As the paper mentions, [Wang and Jia, AISTATS '23] have a result that states that Banzhaf maximizes robustness amongst all semi-values, which includes weighted Banzhaf. How can it be reconciled with the result of this paper that shows that in some cases a non-0.5 weighted Banzhaf is more robust? I do not believe that [Wang and Jia, AISTATS '23] assume isotopic noise (correct me if I am wrong). Is that not rather due to a different definition of robustness? It would be better if the paper was clearer about that.

The noise model introduced lacks motivation. Specifically, the noise is applied directly on the value function, but there is no attempt made to link this noise model to noise in the data. In fact, l. 169, the paper says that the covariance of $v(S)$ for a subset $S$ of size $s$ decreases like $1/C^{s}$ with $C=\sigma_{11}/\sigma_{22}>1$ if $\sigma_{11} > \sigma_{22}$. This brings two questions:
- why is $\sigma_{11} > \sigma_{22}$ linked to a decreasing variance with the size of the subset?
- why is this a good model? Sure, we expect that the variance would decrease, but no as fast as $1/C^{s}$, rather something classical like $1/\sqrt{s}$. All that is to say that it is not clear (and even less justified in the paper) that the Kronecker noise model is relevant for the data valuation application.

The paper also lacks motivation for the specific robustness measure used (the determinant of the covariance of the values). There is a somewhat vague paragraph that explains that gaussian distributions maximize entropy for fixed covariance but why entropy and why fix covariance? Also, is it possible that the values are gaussian?
(Side note: in this paragraph, the covariance $\Sigma$ is generic whereas earlier it is specifically the individual one for Kronecker noise, that is confusing.)

Some parts of the paper are a bit fast and unclear. For instance Proposition 2 and the comment below does not really explain what the convergence is, what the sample lift strategy is and why it is so bad here. This is not too crucial since this result is anecdotical but is frustrating for the reader.

**Questions:**

- In the experimental section: my understanding is that the paper fits the covariance matrix of the Kronecker noise to the data and then adds noise to subset with this fitted matrix. Is that correct? I wonder: is the fit good at all? This is related to my earlier comment that I do not feel that Kronecker noise is necessarily a good model here (e.g., because of the too fast decrease of the variance, see above).

- More generally, in many places I was not able to know what the value fonction used is how exactly noise is added in the experiments. Maybe I missed the information?

- The introduction mentions robustness to human errors and attacks. I do not believe that what is proposed here can reasonably be robust to attacks, at least not to worst-case unconstrained attacks. Can the authors comment on the robustness to attacks?

- Understanding Fig 1 is basically not possible because it is not written what the value is and what noise is added. Even after reading the experimental section I was not sure what is plotted on Fig 1.

- Minor: In Remark 1: \sigma_11 = \sigma_22 is more general than isotropic.

**Limitations:**

Limitations are discussed in Appendix E, which is kind of hidden after 14 pages of proofs and is not referenced in the text. It'd be better to at least put a pointer in the main text.

---

> ### Author Rebuttal · Authors · 2023-08-09
>
> Thank you for the detailed review and helpful feedback. Some comments are addressed in our global response, and below we address with the remaining ones. We have elaborated more on our proofs to ease the verification from readers. At the very beginning, we first noticed weighted Banzhaf values are special as stated in lemma 7 (while forging the framework of semi-indices using materials scattered in many cooperative game theory references), which guided us to have a guess  for theorem 1 and 2. Then, we verified them partially by running some experiments before working out their rigorous proofs. The Kronecker structure naturally suggests using induction to prove the results.
>
> **Q: Whether we add noises to subset with the fitted matrix and how we add noises?**
>
> A: Except for the first column of Figure 3 where we artificially added non-isotropic Gaussian noises to *deterministic* utility functions (with a fixed random seed 2023), we *did not intentionally* add noises elsewhere.
> Precisely, each stochastic utility function $v(\cdot)$, e.g., those used in *Figure 1*, and Tables 2 and 3, can be written as $ v(\cdot,U) $ where $ U $ is the random seed controlling the initialization of trainable models, as well as the order of feeding data in the training phase. In other words, the randomness of stochastic utility functions *only* comes from varying $ U $, and say, $ 6 $ independent runs means we set $ U=0,1,\dots,5 $ for each run.
> Line 675 in the supplementary demonstrates how we generate $\hat{\mathbf{D}} $, to which an individual covariance matrix $ \hat{\boldsymbol\Sigma} $ is fitted.
> Then, the fitted $ \hat{\boldsymbol\Sigma} $ is *only* used to generate the supposedly most robust weighted Banzhaf value parameterized by $ w^{*} $ according to Eq. (6), which is tagged by ''robust'' in Figure 3.
>
> **Q: Is the Kronecker noise a good fit?**
>
> A: We admit that the Kronecker noise is not capable of capturing all possible real noises.
> Nevertheless, as shown by the last two columns in Figure 3, each $w^{*}$ obtained from the Kronecker noise fitted to the real noise is the most consistent semi-value in empirical ranking.
> Therefore, in these two experiment settings, it is safe to say that the Kronecker noise is a good-fit model for the underlying noises induced by stochastic training. Though the used datasets are small, the conclusions (that weighted Banzhaf values tend to be the most consistent in ranking) can still be observed empirically on larger datasets, as shown in Table 3.
>
> **Q: Robustness to human errors and attacks.**
>
> A: We did not consider this setting in this work as we only model the randomness from the stochasticity during training. This direction would be an interesting future work.
>
> **Q: Why is $\sigma_{11} > \sigma_{22}$ linked to decreasing variance with the size of the subset?**
>
> A: This is the implication from proposition 3 that gives the analytical formula for $\mathrm{Cov}(\boldsymbol\epsilon) = \mathrm{Cov}(v)$ (the covariance matrix) given that $\boldsymbol\epsilon=v-\mathbb{E}[v]$ follows some Kronecker noise. In other words, $\mathrm{Var}[v(S)] = \sigma_{11}^{n-s}\sigma_{22}^s$. As $s$ increases,  $\mathrm{Var}[v(S)]$ decreases provided that $\sigma_{11}>\sigma_{22}$.
>
> **Q: Details on the sampling lift.**
>
> A: As dubbed by Moehle et al. (2022), sampling lift refers to any approximation based on $\phi_i(v) = \underset{S\subseteq[n]\backslash i}{E} [v(S\cup i)-v(S)]$. Precisely, Eq (1) is equal to$$\phi_i(v)=\sum_{k=0}^{n-1}q_k \sum_{S \subseteq[n]\backslash i,|S|=k}\binom{n-1}{k}^{-1}(v(S\cup i)-v(S))=\underset{k}{E}\underset{S:|S|=k,i\not\in S}{E}[v(S\cup i)-v(S)].$$ where $q_k=p^{n-1}_{k}\binom{n-1}{k} $. The sampling strategy used in (Moehle et al. 2022) is i) sample $k\sim(q_k)$, and then ii) sample $S$ uniformly subject to $|S|=k$ and $i\not\in S$.
> For the reweighted sampling lift used by Kwon and Zou (2022), they consider reweighted marginal contributions $q_s n(v(S\cup i)-v(S))$ instead. A drawback is that $q_k$ has to be calculated cleverly to avoid the numerical blowup induced by $\binom{n-1}{k}$, which is why Wang and Jia (2023) did not provide results for Beta Shapley on >500-data datasets.
> Nevertheless, we noted that there is a better way briefly mentioned by Dubey et al. (1981). Substituting Eq (2) in Eq (1) yields$$\phi_i(v)=\int_0^1\underset{S\subseteq[n]\backslash i}{\sum}t^s(1-t)^{n-1-s}(v(S\cup i)-v(S))dP(t).$$ Subsequently, take $\phi_n(v)$ as an example: i) sample $t\in[0,1]$ according to the provided $P$; ii) let $X$ be a Bernoulli random variable such that $P(X=1)=t $, and sample a vector $b\in R^{n-1} $ whose entries are independently following $X$;
> and iii) define $S\subseteq[n]\backslash n$ by letting $i\in S$ if and only if $b_i=1$.
> For Beta Shapley, $ t\propto t^{\beta-1}(1-t)^{\alpha-1}$, as shown in Table 1.
>
> **Q: Why sampling lift is bad?**
>
> A: To answer, we generalize the convergence results in (Wang and Jia, 2023) by assuming that the given utility function $v$ satisfies $ |v|\leq r $. Wang and Jia (2023) analyzed the setting $v\in[0,1]$ where $v$ was taken to be prediction performance.
> To describe rate of convergence, we need the concept of $(\epsilon,\delta)$-approximation for the considered semi-value $\phi$, which is $P[||\hat{\phi}-\phi||\_\infty\geq\epsilon]\leq\delta$. For maximum sample reuse principle employed for all $w$-wighted Banzhaf value, adapting their proofs for theorem 4.9 gives that it requires $ \frac{(2|w-0.5|+2)^2r^2}{2\epsilon^2w^2(1-w)^2}\log(\frac{5n}{\delta})$, or $O(\log(n/\delta)/\epsilon^2)$, model evaluations to achieve an $(\epsilon,\delta)$-approximation. Besides, theorem 4.8 therein (though they only asserted for the Banzhaf, it applies to all semivalues) proves that it requires $ \frac{4nr^2}{\epsilon^2}\log(\frac{2n}{\delta})$ model evaluations, or $O(n\log(n/\delta)/\epsilon^2)$, for sampling lift.
> These two results somewhat show why maximum sample reuse principle is better than sampling lift.

---

> > ### Comment · Reviewer_SBfP · 2023-08-11
> >
> > I thank the authors for their response. It answers many of my questions.
> >
> > There are two key points that remain in my opinion:
> > - the Kronecker noise model is somewhat arbitrary. It's an interesting noise model but not necessarily one that is a clearly good model of noise at training.
> > - the measure of robustness is also a bit arbitrary (I have not see a clear intuition for it).
> >
> > These aren't really questions, these are more the main negative points that I see in the contribution. All the positive points of course still remain. So all in all I am favorable to accepting the paper but I feel my score of 6 is a good reflection of my overall opinion (the paper has interesting theoretical results but maybe not of significance that justifies a higher score).

---

> > > ### Author Response · Authors · 2023-08-17
> > >
> > > Thank you for the thoughtful comments. This response is to acknowledge the key points and add some further clarifications.
> > > - One point of our work is to bring weighted Banzhaf values into view in data valuation. The empirical effectiveness of this family can be observed in our experiments *where the underlying noises do not necessarily follow any assumption or theory* (see Tables 2 and 3).
> > > In these experiments, we only tuned the hyperparameters (e.g., the learning rate) for each dataset, and the induced noises are strictly due to stochastic training (e.g., random initialization). Our empirical results demonstrate that *there is probably no universally the most robust semivalue.*
> > > - Our theorem 1 aims to give a *possible* theoretical explanation for the adaptive phenomenon (i.e., no single semi-value dominates) observed in Tables 2 and 3. To validate the criteria of differential entropy,  we showed in the first column of Figure 3 that the derived Eq. (6) leads to the most consistent semivalue in empirical rankings. Similar results are obtained in Figure 3 (the last two columns) where the noises are real and agnostic. Intuitively, a random vector tends to be deterministic (which means the resulting random ranking is also deterministic) as its differential entropy decreases to 0.
> > >
> > > Overall, our work is to emphasize two points: i) weighted Banzhaf values are *already empirically promising*; and ii) there is no universally the most robust/effective semivalue for all experimental settings, and our theory is an attempt to explain *such an adaptive phenomenon* observed in our experiments.

---

### Author Rebuttal · Authors · 2023-08-08

We thank all the reviewers for the constructive feedback. Below we address the questions most reviewers are concerned with, which will be included in our revision.

**Q: Clarify the setting for theorem 1**

A: For each noisy utility function $v$, *its randomness is due to stochasticity during training.* Our assumption for this randomness is that $\epsilon=v-E[v]$ follows a Kronecker noise, i.e., $Cov(\epsilon)=\Sigma^{[n]}=\Sigma\otimes\cdots\otimes\Sigma$ (n repetitions) for some $\Sigma\in R^{2\times2}$ where $Cov$ stands for covariance matrix. Recall that $v$ is *ordered* as a vector in $R^{2^n}$ w.r.t. the binary ordering. This ordering for all subsets of $[n]$ aligns well with the Kronecker product, which is why our modeling for the noise is based on the Kronecker product. By definition 1, $\epsilon=X_1\otimes X_2\otimes\cdots\otimes X_n$ where $Cov(X_i)=\Sigma$ for each $i$ and all *continuous* random variables $X_i$ are independent. Then, $\epsilon(S)=\prod_{i\in S}X_i(2)\prod_{i\not\in S}X_i(1)$ for every $S$ (note $X_i\in R^2$). It means that $X_i(1)$ represents the randomness brought in by the absence of datum $i$, while $X_i(2)$ is the one induced by its presence. Note that $Cov(\epsilon)=Cov(v)$, and the optimization problem in theorem 1 starts from $$\underset{P\in\mathcal{P}}{\arg\min}\sup_{v\in\mathcal{G}:Cov(v)=\Sigma^{[n]}}h(\phi_P(v))$$ where $ \mathcal{P} $ contains all distributions on the interval $ [0,1] $, $ h $ is the differential entropy that measures the uncertainty of continuous random vectors, and the $ i$-th entry of $\phi_P(v)\in R^n$ is, obtained by substituting Eq (2) in Eq (1), $$\sum_{S\subseteq [n]\backslash i}(\int_0^1 t^s(1-t)^{n-1-s} dP(t))\cdot(v(S\cup i)-v(S)).$$ In a nutshell, we found a semivalue defined by $P$ that can best tolerate the largest uncertainty brought by a noise having covariance matrix $\Sigma^{[n]}$.
Since $\phi_P$ is linear, $Cov(\phi_P(v))$ is the same, denoted by $\Phi$, given any $v$ satisfying $Cov(v)=\Sigma^{[n]} $. Let $Y=\phi_P(v)$, it is known that $\sup_{Y:Cov(Y)=\Phi}h(Y)=\frac{n}{2}(1+\log(2\pi))+\frac{1}{2}\log(\det(\Phi))$, and the maximum is achieved if $\phi_P(v) $ is Gaussian. Since $\phi_P$ is linear, $\phi_P(v) $ is Gaussian if $v$ is Gaussian. Thus, we have the equivalent problem in theorem 1 (the outer product was our typo)$$\underset{P\in\mathcal{P}}{\arg\min}\det(Cov(\phi_P(v)))\text{ s.t. }Cov(v)=\Sigma^{[n]}.$$
**Q: Clarify between the concepts of Kronecker noise and safe margin.**

A: They are distinct modelings for the inherent noises. For every $p^{n-1}\in R^n$ defining a semivalue, and every $\tau>0$ (the choice of it is irrelevant to the resulting ranking for semivalues), the safe margin is$$\text{Safe}(\tau;p^{n-1})=\min_{i,j\in[n]:i\not=j}\min_{v\in\mathcal{G}:\Delta_{i,j}^{(k)}(v)\geq\tau\ \forall1\leq k\leq n-1}\min_{\hat{v}\in\mathcal{G}:D_{i,j}(v;p^{n-1})D_{i,j}(\hat{v};p^{n-1})\leq0}||v-\hat{v}||_F$$

$$\text{where }\Delta_{i,j}^{(k)}(v)=\binom{n-2}{k-1}^{-1}\sum_{|S|=k-1,S\subseteq[n]\backslash ij}[v(S\cup i)-v(S\cup j)]\text{ and }D_{i,j}(v;p^{n-1})=n(\phi_i(v;p^{n-1})-\phi_j(v;p^{n-1})),$$which means, as claimed, the safe margin is the largest noise that can be tolerated by the semivalue $\phi(\cdot;p^{n-1})$ without changing the ranking for data. It is said that $D_{i,j}(v;p^{n-1})D_{i,j}(\hat{v};p^{n-1})\leq0$ is equivalent to that $v$ and $\hat{v}$ produce different orders for data $i$ and $j$. A merit of the safe margin is that the problem is independent of any noise. In contrast, the Kronecker noise allows one to exploit the individual covariance matrix $\Sigma$, which makes it possible to derive different conclusions. In the synthetic experiments of Wang and Jia (2023), they added isotropic Gaussian noises to *deterministic* utility $v$ to support their theory (see the setting for Figure 7 therein), which also aligns with our theory. Instead, we also added non-isotropic Gaussian noises (the first column of Figure 3) to verify our theory (which reveals the Banzhaf value is not necessarily the most robust in this case). In a nutshell, more flexible modeling leads to finer results.

---

### Decision · Program_Chairs · 2023-09-21

**Decision:**

Accept (poster)

**Comment:**

this paper considers the problem of data valuation, with the Banzhaf value concept instead of the (more classical) Shapley one. This has been done in the past already, but formalised here with the presence of noise.

This is a clear, interesting paper that all reviewers enjoyed (even though they all have comments that should be taken care of for the revised version). I recommend acceptance